# A Globally Convergent Algorithm for Neural Network Parameter Optimization Based on Difference-of-Convex Functions

**Daniel Tschernutter**                                                    *dtschernutter@ethz.ch*
*ETH Zurich*

**Mathias Kraus**                                                         *mathias.kraus@fau.de*
*FAU Erlangen-Nuremberg & ETH Zurich*

**Stefan Feuerriegel**                                                      *feuerriegel@lmu.de*
*Munich Center for Machine Learning & LMU Munich*

**Reviewed on OpenReview:** *https://openreview.net/forum?id=EDqCY6ihbr*

## Abstract

We propose an algorithm for optimizing the parameters of single hidden layer neural networks. Specifically, we derive a blockwise difference-of-convex (DC) functions representation of the objective function. Based on the latter, we propose a block coordinate descent (BCD) approach that we combine with a tailored difference-of-convex functions algorithm (DCA). We prove global convergence of the proposed algorithm. Furthermore, we mathematically analyze the convergence rate of parameters and the convergence rate in value (i. e., the training loss). We give conditions under which our algorithm converges linearly or even faster depending on the local shape of the loss function. We confirm our theoretical derivations numerically and compare our algorithm against state-of-the-art gradient-based solvers in terms of both training loss and test loss.

## 1 Introduction

Neural networks have emerged as powerful machine learning models for applications in operations research and management science. A particular class of neural networks are single hidden layer feedforward neural networks (SLFNs). This class offers large flexibility in modeling relationships between input and output and, therefore, is used in several domains such as transportation (Celikoglu & Silgu, 2016), risk analysis (Sirignano & Giesecke, 2019), pricing (Haugh & Kogan, 2004), and healthcare (Lee et al., 2013).

SLFNs have several properties that make them relevant for both machine learning practice and theory. In machine learning practice, they have been found to be effective in tasks with mid-sized datasets (i. e., hundreds or thousands of samples). In machine learning theory, SLFNs entail favorable properties. For instance, SLFNs have been shown to be universal approximators. As such, SLFNs are able to approximate continuous functions arbitrarily well on compact sets given a sufficient number of hidden neurons (Hornik et al., 1989). Only recently, the expressiveness of SLFNs in regression tasks has been investigated further, and it was shown that there always exists a set of parameters such that the empirical mean squared error is zero, as long as the number of hidden neurons is larger than a finite bound depending solely on the number of training examples and the number of covariates (Zhang et al., 2021). The simple structure of SLFNs oftentimes also allows to derive theoretical properties in downstream tasks. For instance, in option pricing, SLFNs can be used for approximating value functions in approximate dynamic programming to derive tight bounds for option prices (Haugh & Kogan, 2004). In financial risk analysis, SLFNs can be used to predict conditional transition probabilities in discrete-time models, where SLFNs allow one to derive a law of large numbers and a central limit theorem for pool-level risks (Sirignano & Giesecke, 2019).

This work considers the parameter optimization in single hidden layer feedforward neural networks with activation function $\sigma(\cdot) = \max(\cdot, 0)$ and $N$ hidden units to predict target variables $y_j \in \mathbb{R}$ from samples $x_j \in \mathbb{R}^n$, $j = 1, \ldots, m$. The parameters are determined via the optimization problem

$$\inf_{\alpha, W, b} \frac{1}{m} \sum_{j=1}^{m} (\langle \alpha, \sigma(W x_j + b) \rangle - y_j)^2 + \gamma \operatorname{Reg}(\alpha, W, b), \tag{PNN}$$

where the objective is to find optimal parameters $\alpha \in \mathbb{R}^N$, $W \in \mathbb{R}^{N \times n}$, and $b \in \mathbb{R}^N$. The first term in (PNN) represents the mean squared error, while Reg is a regularization term with regularization parameter $\gamma > 0$. In machine learning, a common regularization technique is weight decay, i.e., Reg favors smaller weights. The activation function allows for feature selection by activating or deactivating neurons in the hidden layer. In particular, the rectified linear unit (ReLU), as defined by $\sigma$, results in sparse feature representations (Glorot et al., 2011), which has shown to improve predictive performance, making it a common choice in machine learning practice (LeCun et al., 2015). From a theoretical point of view, the above optimization problem entails the following characteristics: (i) The objective function is highly non-convex; (ii) the objective function is non-differentiable; and (iii) the optimization problem is high-dimensional. These characteristics make the optimization problem difficult to solve.

Previous optimization methods for the above task have primarily been gradient-based; see Section 2 for an overview. Foremost, stochastic gradient descent (SGD) and variants thereof (e.g., using momentum) are applied to optimize neural networks. However, gradient-based methods do not have general convergence guarantees, and require expert knowledge during training due to many hyperparameters (e.g., initial learning rate, momentum). Only recently, methods that do not rely on gradient information have been proposed (e.g., Lau et al., 2018; Zeng et al., 2019; Zhang & Brand, 2017). Here, optimization methods based on block coordinate descent (BCD) and alternating direction method of multipliers (ADMM) have been proposed for optimizing parameters in neural networks. However, previous approaches merely optimize over surrogate losses instead of the original loss or lack convergence guarantees.

In this paper, we propose a globally-convergent algorithm for parameter optimization in SLFNs. Our algorithm named DCON builds upon difference-of-convex (DC) functions optimization and further follows a block coordinate descent approach, where, on each block, the objective function is decomposed as a difference of convex functions. Optimizing over these blocks results in subproblems, which are either already convex or can be approached with a tailored difference-of-convex functions algorithm (DCA). For an introduction to DCA, see Le An & Tao (2005). We provide a theoretical analysis of our algorithm. First, we prove global convergence in value and to limiting-critical points (under additional assumptions). Second, we derive theoretical convergence rates, and, third, we give conditions under which our algorithm DCON converges against global minima. We want to point out that **global convergence** does **not** mean **convergence to global minima**. Global convergence is the convergence independent of starting points and, hence, in our case, independent of weight initialization. See also Lanckriet & Sriperumbudur (2009) for the definition of global convergence.

Our work contributes to machine learning theory in the following ways:

1. We show that our algorithm converges globally (i.e., independently of weight initializations) in value and, under additional assumptions, to limiting-critical points.
2. We give conditions under which the training loss converges with order $q \in \mathbb{N}$. That is, the training loss as defined in (PNN) can achieve very fast convergence depending on the local shape of the loss function.
3. We compare DCON against Adam (Kingma & Ba, 2014) as a state-of-the-art gradient-based optimizer. Our evaluation on nine datasets from the UCI machine learning repository and the MNIST dataset shows that DCON achieves a superior prediction performance.

DCON offers a key benefit for machine learning practice. DCON works without any hyperparameters during training. In particular, hyperparameters that are otherwise common in gradient-based solvers (e.g., learning

rates, number of training epochs, momentum) are absent. Hence, given the number of hidden neurons and the regularization parameter in the SLFN, one can perform the training task in a completely automated manner. Nevertheless, future work is needed to scale DCON to larger datasets in practice. We point to potential research directions in our discussion.

The rest of the paper is structured as follows. In Section 2, we discuss previous research on optimization methods for neural networks. Section 3 analyzes the optimization problem in (PNN) with regard to necessary optimality conditions. Section 4 introduces our novel training algorithm DCON, while, in Section 5, we derive global convergence results and theoretical properties of DCON. In Section 6, we perform numerical experiments to compare DCON against state-of-the-art optimization methods and demonstrate the convergence behavior. Finally, Section 7 concludes.

## 2 Background

We contextualize our contribution within the literature on optimization methods for (PNN). Previous research can be loosely grouped into methods for (i) direct loss optimization and (ii) surrogate loss optimization.

***Direct loss optimization:*** Algorithms that directly optimize the loss function can be divided into gradient-based and gradient-free methods.

Gradient-based methods make use of backpropagation (Rumelhart et al., 1986) to compute gradients of the loss function. The underlying basis is given by stochastic gradient descent (SGD) proposed by Robbins & Monro (1951). Several adaptive variants of vanilla SGD have been developed in recent years. Examples are Adam (Kingma & Ba, 2014), AdaGrad (Duchi et al., 2011), RMSProp (Tijmen & Hinton, 2012), and AMSGrad (Reddi et al., 2018). Despite the tremendous success of SGD in optimizing neural network parameters, a general convergence theory is still lacking. A broad stream of literature provides theoretical guarantees for SGD or variants thereof under fairly restrictive assumptions (Chen et al., 2020; Chizat et al., 2019; Du et al., 2019a;b; Li & Liang, 2018; Liang et al., 2021; Zeyuan et al., 2019; Zou & Gu, 2019; Zou et al., 2020). Therein, the authors foremost assume some kind of over-parametrization, i.e., the number of hidden units $N$ has to increase in linear (Liang et al., 2021), polynomial (Du et al., 2019a;b; Li & Liang, 2018; Zeyuan et al., 2019; Zou & Gu, 2019; Zou et al., 2020) or poly-logarithmic (Chen et al., 2020) order of the number of training samples $m$. For instance, in Du et al. (2019b), convergence is guaranteed if, among other assumptions, $N = \Omega(m^6)$.[1] Other works make assumptions on the distribution of the input data or rely on differentiability assumptions, i.e., smooth neural networks and smooth losses (Chizat et al., 2019). Of note, the analyses in Chen et al. (2020); Li & Liang (2018); Liang et al. (2021), and Zou et al. (2020) are restricted to classification tasks, while we are interested in a regression task. The aim of the aforementioned theoretical works is to answer the question of why neural networks are so successful in non-convex high-dimensional problems, rather than developing a unified convergence theory. Noteworthy, the analysis in Davis et al. (2020) establishes the subsequence convergence of SGD on tame functions. In contrast, our algorithm establishes the convergence of the whole sequence of parameters under certain assumptions. According to Zeng et al. (2019), this gap between the subsequence convergence and the convergence of the whole sequence comes from the fact that SGD can only achieve a descent property (see Assumption F in Davis et al. (2020)), while we prove that our algorithm achieves a sufficient descent property.

Gradient-free methods refrain from using first-order information and deal differently with the undesired characteristics of the objective function. One stream of research focuses on so-called extreme learning machines (Huang et al., 2006). Their idea is to merely sample the weights and intercepts associated with the input layer from a given probability distribution and then solve a linear least squares problem to derive the weights associated with the hidden layer. Extreme learning machines are used in various applications, yet they do not solve (PNN) but instead only act as a heuristic. Recently, Pilanci & Ergen (2020) showed that there exists a convex problem with identical optimal values as (PNN), from which optimal solutions to (PNN) can be derived. Consequently, this would give a global optimum. However, this work should be seen

---

[1]Note that over-parametrization would require neural networks with a very large number of neurons, e.g., thousands of neurons for datasets with only a thousand training examples.

as a purely theoretical contribution as the computational complexity of their algorithm scales exponentially in the number of inputs $n$.

***Surrogate loss optimization:*** Another stream of research does not solve (PNN) directly, but uses a BCD or ADMM approach after replacing the objective with a surrogate loss. Both BCD and ADMM rely on a so-called two-splitting (e.g., Carreira-Perpinan & Wang, 2014; Zhang & Brand, 2017) or three-splitting formulation (e.g., Lau et al., 2018; Taylor et al., 2016). The underlying idea is to introduce auxiliary coordinates for each datapoint and each hidden unit. These auxiliary coordinates appear as equality constraints in the optimization problem of the neural network. However, as these equality constraints lead to an intractable optimization problem, these approaches involve an alternative formulation of (PNN) in which an additional penalty term is required to enforce the equality constraints. The resulting formulation then allows one to optimize iteratively over different variable blocks, resulting in easier (and often convex) subproblems (Askari et al., 2018; Lau et al., 2018; Zhang & Brand, 2017). As a result, it is oftentimes possible to prove global convergence for these approaches (Zeng et al., 2019). However, such surrogate losses for (PNN) lead to a decreased performance in comparison to gradient-based solvers (Askari et al., 2018; Lau et al., 2018) or neural networks that cannot be evaluated at test time (Zhang & Brand, 2017).

Other works derive efficient training algorithms for upper bounds of the loss function; see, e.g., Berrada et al. (2017). The latter also relies on a DC representation of the objective function. However, while the algorithm converges in value, a thorough convergence analysis of parameters is missing. Only recently, Mishkin et al. (2022) proposed a training algorithm for SLFNs with ReLU activation using convex optimization. However, their approach uses a convex reformulation (C-ReLU) of (PNN) that is only equivalent to (PNN) under additional assumption (e.g., among others again a sufficiently large number of hidden neurons), and then solves a surrogate problem (C-GReLU) involving a different activation function, a so-called gated ReLU, that is again only approximately solving (C-ReLU) and thus (PNN).

In sum, there is a large number of research papers that study the training of neural networks. However, to the best of our knowledge, we are the first that present an algorithm that (i) *directly* solves (PNN), (ii) is scalable to mid-sized datasets, and (iii) has general global convergence guarantees, i.e., does not rely on any kind of over-parametrization.

## 3 Optimization Problem

In this section, we formalize the optimization problem and discuss necessary optimality conditions.

### 3.1 Problem Statement

Given a set of $m$ training samples consisting of covariate vectors $x_j \in \mathbb{R}^n$ and target variables $y_j \in \mathbb{R}$, $j = 1, \ldots, m$, the objective is to optimize the parameters of a SLFN with $N$ neurons in the hidden layer and ReLU activation $\sigma$, as given in (PNN). Let $w_i \in \mathbb{R}^n$ and $b_i \in \mathbb{R}$, $i = 1, \ldots, N$, denote the weights and intercepts associated with the input layer, and let $\alpha_i \in \mathbb{R}$, $i = 1, \ldots, N$, denote the weights associated with the hidden layer. All trainable parameters are given by $\theta = (\alpha, W, b) \in \mathbb{R}^{\mathcal{N}}$, where $\mathcal{N} = (n+2)N$. We denote the objective function in (PNN) by $\mathrm{RegLoss}_\gamma(\theta) = \mathrm{Loss}(\theta) + \gamma \cdot \mathrm{Reg}(\theta)$, where $\mathrm{Loss}(\theta)$ is the mean squared error loss from (PNN) defined as

$$\mathrm{Loss}(\theta) = \frac{1}{m} \sum_{j=1}^{m} \left( y_j - \sum_{i=1}^{N} \alpha_i \sigma(\langle w_i, x_j \rangle + b_i) \right)^2, \tag{1}$$

and $\mathrm{Reg}(\theta)$ is a data-weighted $\ell_2$-regularization defined by

$$\mathrm{Reg}(\theta) = \frac{1}{m} \left( \sum_{j=1}^{m} \sum_{i=1}^{N} (\langle w_i, x_j \rangle + b_i)^2 \right) + \frac{1}{m} \|\alpha\|^2. \tag{2}$$

For convenience, we introduce a matrix $M \in \mathbb{R}^{m \times (n+1)}$, which refers to the matrix with rows formed by the covariate vectors with an additional one for the bias, i.e., $(x_{j,1}, \ldots, x_{j,n}, 1)$ for $j \in \{1, \ldots, m\}$. Using the

above definition of $M$, Equation (2) can be written as $\frac{1}{m} \sum_{i=1}^{N} \left\| M \begin{pmatrix} w_i \\ b_i \end{pmatrix} \right\|^2 + \frac{1}{m} \|\alpha\|^2$. This form of regularization has two main advantages in our derivations later on: (i) it allows to derive a convenient DC structure of the objective function, and (ii) it allows for a closed-form solution of the sub-gradients needed in our DCA routine. Nevertheless, it merely corresponds to a standard $\ell_2$-regularization with an additional weight matrix $M$.[2]

## 3.2 Necessary Optimality Conditions

We first prove the existence of a solution to (PNN).

**Proposition 1** (Existence of a solution). *Let $M^T M > 0$. That is, the smallest singular value of $M$ denoted by $\sigma_{\min}(M)$ is greater than zero, i. e., $\sigma_{\min}(M) > 0$. Then, there exists at least one solution to (PNN).*
*Proof. See Appendix A.* □

Next, we present a necessary optimality condition. Note that the loss function in neural network parameter optimization is, as in our case, usually not differentiable in the classical sense. Hence, to state necessary optimality conditions for solutions to (PNN), we draw upon the concept of limiting subdifferentials. For a detailed introduction, we refer to Penot (2012).

**Definition 1** (Limiting subdifferential). *Let $f : \mathbb{R}^{\mathcal{N}} \to \mathbb{R} \cup \{\infty\}$. The Fréchet subdifferential of $f$ at $x \in \mathrm{dom}(f)$, denoted by $\partial^F f(x)$, is given by the set of vectors $v \in \mathbb{R}^{\mathcal{N}}$ which satisfy*

$$\liminf_{\substack{y \neq x \\ y \to x}} \frac{1}{\|x - y\|} [f(y) - f(x) - \langle v, y - x \rangle] \geq 0. \tag{3}$$

*If $x \notin \mathrm{dom}(f)$, we set $\partial^F f(x) = \emptyset$. Then, the limiting subdifferential of $f$ at $x \in \mathrm{dom}(f)$, denoted by $\partial^L f(x)$, is defined as*

$$\partial^L f(x) = \{v \in \mathbb{R}^{\mathcal{N}} : \exists x^k \to x, f(x^k) \to f(x), \partial^F f(x^k) \ni v^k \to v\}. \tag{4}$$

Definition 1 provides a generalized notion of a critical point. Based on it, we state the following necessary optimality condition.

**Remark 1** (Necessary optimality condition). *A necessary condition for $\theta^* \in \mathbb{R}^{\mathcal{N}}$ to be a solution to (PNN) is $0 \in \partial^L \mathrm{RegLoss}_\gamma(\theta^*)$. For details, see Attouch et al. (2013) and the references therein.*

Such a point $\theta^*$ is called limiting-critical, or simply critical. Besides the need for a generalized notion of critical points, the lack of differentiability is also challenging when it comes to analyzing the local behavior of functions. To derive an algorithm that converges towards critical points, we need a notion of how our objective function behaves near critical points. For this, we make use of the so-called Kurdyka-Łojasiewicz (KŁ) property. The KŁ property is a valuable tool in the context of optimization, as it allows to reparameterize the function locally. The KŁ property is defined as follows (Attouch et al., 2013).

**Definition 2** (Kurdyka-Łojasiewicz property). *A proper lower semicontinuous function $f : \mathbb{R}^n \to \mathbb{R} \cup \{\infty\}$ fulfills the Kurdyka-Łojasiewicz property at a point $x^* \in \mathrm{dom}(\partial^L f)$ if there exists an $\eta \in (0, \infty]$, a neighborhood $U_{x^*}$ of $x^*$, and a continuous concave function $\varphi : [0, \eta] \to \mathbb{R}_+$ such that*

---

[2]Note that the standard $\ell_2$-regularizer is given by a term $\|\theta\|_2^2$, while our regularization term merely uses a different energy-norm. That is, our regularizer can be seen as $\|\theta\|_B^2$ for a certain positive definite matrix $B$ given by

$$B = \frac{1}{m} \begin{pmatrix} M^T M & 0 & \dots & 0 & 0 \\ 0 & M^T M & \dots & 0 & 0 \\ \vdots & \vdots & \ddots & 0 & 0 \\ 0 & 0 & 0 & M^T M & 0 \\ 0 & 0 & 0 & 0 & I \end{pmatrix}.$$

1. $\varphi(0) = 0$,

2. $\varphi$ is $\mathcal{C}^1$ on $(0, \eta)$,

3. $\forall s \in (0, \eta)$: $\varphi'(s) > 0$,

4. $\forall x \in U_{x^*} \cap [f(x^*) < f < f(x^*) + \eta]$: $\varphi'(f(x) - f(x^*))$ $\mathrm{dist}(0, \partial f(x)) \geq 1$ *(KŁ inequality)*.

*A proper lower semicontinuous function $f : \mathbb{R}^n \to \mathbb{R} \cup \{\infty\}$ which fulfills the Kurdyka-Łojasiewicz property at each point $x^* \in \mathrm{dom}(\partial^L f)$ is called KŁ function.*

In the next section, we derive an algorithm for which we later prove that it converges to a limiting-critical point of $\mathrm{RegLoss}_\gamma$.

## 4 DCON Algorithm

The idea behind our DCON algorithm is to show that the loss can be partly decomposed as a DC function. For this, we show that, for certain subsets of the parameters $\theta$, i. e., blocks, $\mathrm{RegLoss}_\gamma$ can be written as the difference of convex functions. Specifically, we use a BCD approach in which we loop over different blocks $B_l$, $l = 1, 2, \ldots, N$. In each block, we yield a **DC subproblem**. The DC subproblem is then (approximately) solved via a tailored DCA. The latter involves a series of convex problems and is thus computationally efficient. After looping over all blocks $B_l$, the remaining variables in $\theta$ form an additional block $B_\alpha$, which is approached in the **alpha subproblem**. The alpha subproblem is already convex and can thus be solved efficiently. For better understanding, we visualized the subproblems in Appendix B.1.

### 4.1 Derivation of Subproblems in BCD

In Proposition 2, we show that $\mathrm{RegLoss}_\gamma$ is a DC function if we consider only weights $w_l$ mapping to the $l$-th hidden neuron and its intercept $b_l$, and derive an explicit DC decomposition that later on allows for efficient subgradient computations.

**Proposition 2.** *For fixed $(w_i)_{i \neq l}$, $(b_i)_{i \neq l}$, and $(\alpha_i)_{i=1,\ldots,N}$, the loss can be written in the form*

$$\mathrm{RegLoss}_\gamma(\alpha, W, b) = \mathrm{RegLoss}_\gamma(w_l, b_l) = c_l + g_l(w_l, b_l) - h_l(w_l, b_l), \tag{5}$$

*where $c_l \in \mathbb{R}$ is a constant and the functions $g_l$ and $h_l$ are convex in $(w_l, b_l)$. Both $g_l$ and $h_l$ are given by*

$$g_l(w_l, b_l) = \sum_{j=1}^m \beta_j^{g_l} \sigma(\langle w_l, x_j \rangle + b_l) + \sum_{j=1}^m \frac{\alpha_l^2}{m} \sigma(\langle w_l, x_j \rangle + b_l)^2 \tag{6}$$

$$+ \sum_{j=1}^m \frac{\gamma}{m}(\langle w_l, x_j \rangle + b_l)^2, \tag{7}$$

$$h_l(w_l, b_l) = \sum_{j=1}^m \beta_j^{h_l} \sigma(\langle w_l, x_j \rangle + b_l), \tag{8}$$

*with non-negative weights $\beta_j^{g_l}$ and $\beta_j^{h_l}$. Furthermore, $\alpha_l = 0$ implies $\beta_j^{g_l} = \beta_j^{h_l} = 0$ for all $j \in \{1, \ldots, m\}$.*
*Proof. See Appendix B.* ☐

With the DC decomposition from Proposition 2, we can use DCA to address the corresponding subproblems. When looping over all $B_l = (w_l, b_l)$, the only weights that are not updated are the weights $\alpha_i$, $i = 1, \ldots, N$ of the hidden layer. Hence, in a last step, we hold all weights constant except for $\alpha$, which results in a regularized linear least squares problem; see Proposition 3.

**Proposition 3.** *For fixed $(w_i)_{i=1,\ldots,N}$ and $(b_i)_{i=1,\ldots,N}$, the loss can be written in the form*

$$\mathrm{RegLoss}_\gamma(\alpha, W, b) = \mathrm{RegLoss}_\gamma(\alpha) = c_\alpha + \frac{1}{m}\|y - \Sigma\alpha\|^2 + \frac{\gamma}{m}\alpha^T I \alpha, \tag{9}$$

where $c_\alpha \in \mathbb{R}$ is a constant, $I \in \mathbb{R}^{N \times N}$ is the identity matrix, $y$ is the vector of target variables, $\alpha$ is the vector of weights associated with the hidden layer, and $\Sigma \in \mathbb{R}^{m \times N}$ is the matrix with entries $\Sigma_{ji} = \sigma(\langle w_i, x_j \rangle + b_i)$ for $j \in \{1, \ldots, m\}$ and $i \in \{1, \ldots, N\}$.

*Proof. See Appendix B.* □

Based on the above propositions, we can now optimize $\text{RegLoss}_\gamma$ via a block coordinate descent approach where the blocks are given by $B_l = (w_l, b_l)$ for $l \in \{1, \ldots, N\}$ with objective function $O_l^{\text{DC}}(w_l, b_l) = g_l(w_l, b_l) - h_l(w_l, b_l)$, and $B_\alpha = \alpha$ with objective function $O^\alpha(\alpha) = \frac{1}{m}\|y - \Sigma\alpha\|^2 + \frac{\gamma}{m}\alpha^T I \alpha$. We thus define the following subproblems:

- **DC subproblems:** The $l$-th DC subproblem is defined as

$$\inf_{w_l, b_l} O_l^{\text{DC}}(w_l, b_l). \tag{DC$_l$}$$

- **alpha subproblem:** The alpha subproblem is defined as

$$\inf_\alpha O^\alpha(\alpha). \tag{A}$$

In the following section, we show how both (DC$_l$) and (A) are (approximately) solved within the BCD approach.

## 4.2 Solution of Subproblems in BCD

We now derive efficient procedures to (approximately) solve the defined subproblems.[3]

### 4.2.1 Approximate Solution of DC Subproblems.

The DC subproblems (DC$_l$) are approached with a tailored DCA. For an introduction to DCA, we refer to Le An & Tao (2005). First, we note that $\alpha_l = 0$ implies that the corresponding DC subproblem reduces to $\inf_{w_l, b_l} \sum_{j=1}^m \frac{\gamma}{m}(\langle w_l, x_j \rangle + b_l)^2$, where a solution is given by $(w_l, b_l) = 0$. Hence, for the rest of the section, we assume that $\alpha_l \neq 0$. Following Le An & Tao (2005), the DCA routine for the $l$-th DC subproblem is

$$y_k^* \in \partial h_l((w_l^k, b_l^k)), \tag{10}$$

$$(w_l^{k+1}, b_l^{k+1}) \in \partial g_l^*(y_k^*), \tag{11}$$

until the norm of two successive iterates is sufficiently small (see Proposition 7 later on), and where $(w_l^0, b_l^0)$ is a given initial solution. Therein, the convex conjugate is defined as $f^*(y^*) = \sup_x\{\langle y^*, x \rangle - f(x)\}$ (see, e.g., Borwein & Lewis (2006)). There are two steps that remain to be shown: (i) How to find an element in the subgradient of $h_l$ in Equation (10)? (ii) How to find an element in the subgradient of the convex conjugate of $g_l$ in Equation (11)? Both are addressed in the following.

For (i), an element in $\partial h_l$ can be derived analytically. This is stated in Proposition 4.

**Proposition 4.** *For any $(w_l, b_l)$, we have that*

$$\sum_{j=1}^m \beta_j^{h_l} \begin{pmatrix} x_j \\ 1 \end{pmatrix} H\left(\left\langle \begin{pmatrix} w_l \\ b_l \end{pmatrix}, \begin{pmatrix} x_j \\ 1 \end{pmatrix} \right\rangle\right) \in \partial h_l(w_l, b_l), \tag{12}$$

*where $H$ is the Heaviside function, i. e., $H(x) = 1$ for $x \geq 0$ and zero otherwise.*

*Proof. See Appendix B.* □

---

[3]Note that we use $y^*$ to denote the subgradient of $h_l$ to adhere to standard notation in the DCA literature. It should not be confused with the target values denoted by $y$.

Notably, Equation (12) gives the unique gradient of $h_l$ if $\langle (w_l, b_l), (x_j, 1) \rangle \neq 0$ for all $j \in \{1, \ldots, m\}$.

For (ii), we first have to derive the convex conjugate of $g_l$. However, computing the convex conjugate involves an optimization problem itself, which makes it sometimes difficult to find a closed-form representation. In our case, it is possible to write $g_l^*$ as the difference of a characteristic function and the value function of a positive semidefinite quadratic program. For notation, let $\chi_\Omega$ denote the characteristic function of a set $\Omega$, i.e., $\chi_\Omega(x) = \infty$ for $x \notin \Omega$ and zero else. It will turn out that this difference is a very convenient representation. All details are stated in Proposition 5.

**Proposition 5.** *Let $\alpha_l \neq 0$. The convex conjugate $g_l^*$ of $g_l$ is given by*

$$g_l^*(y^*) = \chi_\Omega(y^*) - \Xi_l(y^*), \tag{13}$$

*where $\Omega = \ker(M)^\perp$ for $M$ as defined in Section 3.1 and $\Xi_l : \mathbb{R}^{n+1} \mapsto \mathbb{R}$ is the value function of the quadratic program*

$$\inf_v \langle q_{y^*}, v \rangle + \frac{1}{2} v^T Q_l v \quad s.t. \quad Av = 0 \text{ and } v \geq 0 \tag{QP}$$

*for $q_{y^*} \in \mathbb{R}^{2m+2(n+1)}$, a sparse block tridiagonal positive semidefinite matrix $Q_l$ with $Q_l \in \mathbb{R}^{2m+2(n+1) \times 2m+2(n+1)}$, depending solely on $\alpha_l$, and a full rank matrix $A \in \mathbb{R}^{m \times 2m+2(n+1)}$ independent of $y^*$ and $\alpha_l$. Furthermore, we have $\mathrm{dom}(g_l^*) = \Omega$.*
*Proof. See Appendix B.* □

In the finite case, the dependency of $g_l^*$ on $y^*$ is only present in the linear term $\langle q_{y^*}, v \rangle$, and $-\Xi_l$ can be seen as a supremum of convex functions over the general index set $V = \{v \in \mathbb{R}^{2m+2(n+1)} : Av = 0 \text{ and } v \geq 0\}$. That is, a subgradient of $g_l^*$ can be obtained by standard subdifferential calculus techniques (see, e.g., Hiriart-Urruty & Lemaréchal, 2004). For this, we need a mild assumption.

**Assumption 1.** *We assume that for all $p \in \{1, \ldots, n+1\}$, there exists a solution $\Delta_p$ to $M^T \Delta_p = e_p$, where $e_p$ is the $p$-th canonical basis vector. This is equivalent to assuming that $M^T M > 0$ as in Proposition 1.*

Assumption 1 is usually fulfilled if enough data are provided. Given that the above holds, we can now derive a closed-form solution of an element in $\partial g_l^*(y^*)$. This is detailed in Proposition 6.

**Proposition 6.** *Let $\alpha_l \neq 0$ and $y^* \in \mathrm{dom}(g_l^*)$. Furthermore, let $v_{y^*}$ be a corresponding solution to (QP) with $v_{y^*} = (v_{y^*}^1, v_{y^*}^2, v_{y^*}^3, v_{y^*}^4)$, then*

$$-\begin{pmatrix} -\Delta_1^T & \Delta_1^T & 0_{n+1}^T & 0_{n+1}^T \\ -\Delta_2^T & \Delta_2^T & 0_{n+1}^T & 0_{n+1}^T \\ \vdots & \vdots & \vdots & \vdots \\ -\Delta_{n+1}^T & \Delta_{n+1}^T & 0_{n+1}^T & 0_{n+1}^T \end{pmatrix} \begin{pmatrix} v_{y^*}^1 \\ v_{y^*}^2 \\ v_{y^*}^3 \\ v_{y^*}^4 \end{pmatrix} = v_{y^*}^3 - v_{y^*}^4 \in \partial g_l^*(y^*), \tag{14}$$

*where $0_{n+1}$ is the zero vector in $\mathbb{R}^{n+1}$.*
*Proof. See Appendix B.* □

In sum, the first DCA step from Equation (10) can be performed efficiently by evaluating Equation (12). The second DCA step from Equation (11) requires that one solves a positive semi-definite quadratic program, which can be done efficiently by out-of-the-box solvers for convex programming. Later, we also present an ADMM-based approach that leverages the special form of the matrix $Q_l$ to derive a scalable solver for (QP).

In the following, we analyze the convergence behavior of the DCA routine for the DC subproblem; see Proposition 7.

**Proposition 7.** *The DCA routine (see (10) and (11)) with $y_k^*$ as given in (12) and $(w_l^{k+1}, b_l^{k+1})$ as given in (14) converges in finitely many iterations to points $(y^*, (w_l^*, b_l^*)) \in [\partial g_l(w_l^*, b_l^*) \cap \partial h_l(w_l^*, b_l^*)] \times [\partial g_l^*(y^*) \cap \partial h_l^*(y^*)]$. Furthermore, there exists an upper bound on the number of iterations $\mathcal{K}^{max} \in \mathbb{N}$ depending solely on the number of training samples $m$, and $(w_l^*, b_l^*)$ is a local solution of $(\mathsf{DC}_l)$ if $\Xi_l(y^*) \leq \Xi_l(y)$ for all $y \in \partial h_l(w_l^*, b_l^*)$. The latter condition is equivalent to $\partial h_l(w_l^*, b_l^*) \subseteq \partial g_l(w_l^*, b_l^*)$.*
*Proof. See Appendix B.* □

Note that the condition $\Xi_l(y^*) \leq \Xi_l(y)$ for all $y \in \partial h_l(w_l^*, b_l^*)$ holds if $\partial h_l(w_l^*, b_l^*)$ is a singleton. That is, for instance, if $\langle (w_l^*, b_l^*), (x_j, 1) \rangle \neq 0$ for all $j \in \{1, \ldots, m\}$, we have that $(w_l^*, b_l^*)$ is a local solution of $(\mathsf{DC}_l)$. For the rest of this paper, we make the following assumption.

**Assumption 2.** *We assume that our DCA routine always converges to a point $(w_l^*, b_l^*)$ with $\partial h_l(w_l^*, b_l^*) \subseteq \partial g_l(w_l^*, b_l^*)$, i.e., a local solution of $(\mathsf{DC}_l)$.*

Note that this assumption is merely made for convenience, as we can always ensure that $\partial h_l(w_l^*, b_l^*) \subseteq \partial g_l(w_l^*, b_l^*)$ holds by a simple restart procedure following Tao & An (1998). We provide such a procedure in Appendix B.8.

### 4.2.2 Solution of Alpha Subproblem.

To derive a solution of $(\mathsf{A})$, we introduce the matrix $H_\gamma = \Sigma^T \Sigma + \gamma I$. Then, by ignoring constant terms, $(\mathsf{A})$ is equivalent to the quadratic program $\inf_\alpha \ \alpha^T H_\gamma \alpha - 2 \langle \Sigma^T y, \alpha \rangle$. Note that the objective is strictly convex. Hence, the unique solution of $(\mathsf{A})$ is given by the solution of the linear system $H_\gamma \alpha = \Sigma^T y$.

### 4.3 Pseudocode

We now combine our derivations into the DCON algorithm for optimizing single hidden layer neural network parameters (see Algorithm 1). In the pseudocode, let $\mathrm{DCA}(\beta^{g_l}, \beta^{h_l}, (w, b), \mathcal{K})$ refer to the DCA subroutine for the $l$-th DC subproblem $(\mathsf{DC}_l)$ with a given initial solution $(w, b)$, weights $\beta^{g_l}$ and $\beta^{h_l}$, and a maximum number of $\mathcal{K}$ iterations. Further, let $\mathrm{LS}(\Sigma)$ refer to the solver of $(\mathsf{A})$ with system matrix $\Sigma$.

The algorithm proceeds as follows. In line 1, the neural network parameters are initialized via the Xavier initialization (Glorot & Bengio, 2010). The idea is then to approach all DC subproblems in a randomized order starting from the current weights (see lines 3–10) and, afterward, solve the alpha subproblem in each outer iteration (see lines 11–13). After each subproblem solution, the new weights are inserted into the parameter vector $\theta$. This results in a new parameter vector $\theta$ after each outer iteration (line 14). Randomization is accomplished by sampling random permutations $\pi$ of the set $\{1, \ldots, N\}$ from the set of all permutations denoted by $S_N$; see line 3.

---

**Algorithm 1:** DCON

**Input:** Number of iterations $\mathcal{M}$, maximum number of DCA iterations $\mathcal{K}$
**Output:** Neural network parameters $\theta^*$
1   Initialize weights $\theta$ via Xavier initialization
2   **for** $k = 0, \ldots, \mathcal{M}$ **do**
3      Choose random permutation $\pi \in S_N$
4      **for** $j = 1, \ldots, N$ **do**
       /* Construct $l$-th DC subproblem                                        */
5        Set $l \leftarrow \pi(j)$
6        Get initial weights and intercept for subproblem $l$ from parameter vector $(w_l^k, b_l^k) \leftarrow \mathrm{GetWeights}(\theta, l)$
7        Compute $\beta^{g_l}, \beta^{h_l} \leftarrow \mathrm{ComputeBetas}(\theta, l)$
       /* DCA for DC subproblem                                            */
8        $(w_l^{k+1}, b_l^{k+1}) \leftarrow \mathrm{DCA}(\beta^{g_l}, \beta^{h_l}, (w_l^k, b_l^k), \mathcal{K})$
9        Update parameter vector $\theta \leftarrow \mathrm{InsertWeights}(w_l^{k+1}, b_l^{k+1}, \theta)$
10      **end**
     /* Construct alpha subproblem                                                  */
11      Get weights and intercepts for alpha subproblem from parameter vector $(w_1^{k+1}, \ldots, w_N^{k+1}, b_1^{k+1}, \ldots, b_N^{k+1}) \leftarrow \mathrm{GetWeights}(\theta)$
12      Compute system matrix $\Sigma \leftarrow \mathrm{BuildSigma}(w_1^{k+1}, \ldots, w_N^{k+1}, b_1^{k+1}, \ldots, b_N^{k+1})$
     /* Solve alpha subproblem                                              */
13      $\alpha^{k+1} \leftarrow \mathrm{LS}(\Sigma)$
14      Update parameter vector $\theta \leftarrow \mathrm{InsertWeights}(\alpha^{k+1}, \theta)$
15   **end**
16   **return** $\theta$

---

### 4.4 Computational Complexity

The computational complexity of Algorithm 1 is mainly driven by the cost for solving the quadratic program from Proposition 5. The quadratic program is solved at most $\mathcal{K}$ times for each of the $N$ hidden neurons in each outer iteration. While state-of-the-art solvers for convex programming are able to exploit the sparsity

pattern in $Q_l$, the worst case complexity is still $\mathcal{O}(m^3)$ assuming that $m \gg n$. As a remedy, we derive an algorithm based on an ADMM approach in Appendix C. Our ADMM approach leverages the block form of $Q_l$ and reduces the computational complexity to $\mathcal{O}(\mathcal{L}m^2)$, where $\mathcal{L}$ is the maximum number of ADMM iterations. In addition, it relies only on basic linear algebra operations that can be implemented efficiently (e. g., using BLAS/LAPACK libraries for CPUs or cuBLAS for GPUs).

# 5 Convergence Analysis

In the following, we provide a convergence analysis for DCON. For this, we first list a set of convergence conditions (Section 5.1) and show that these are fulfilled (Section 5.2). Afterward, we prove that our algorithm converges globally and give conditions under which it even yields a global solution (Section 5.3). Finally, we analyze the convergence rate of our algorithm (Section 5.4).

## 5.1 Convergence Conditions

Our convergence analysis builds upon the framework in Attouch et al. (2013). Therein, the authors show that a sequence $(x^k)_{k \in \mathbb{N}}$ converges to a limiting-critical point of a proper lower semicontinuous function $f : \mathbb{R}^{\mathcal{N}} \to \mathbb{R} \cup \{\infty\}$ if the following four conditions are fulfilled:

- (H0) The function $f$ is a KŁ function.
- (H1) *Sufficient decrease condition*: There exists an $a > 0$ such that, for all $k \in \mathbb{N}$, $f(x^{k+1}) + a \|x^{k+1} - x^k\|^2 \leq f(x^k)$ holds true.
- (H2) *Relative error condition*: There exists a constant $b > 0$ such that, for all $k \in \mathbb{N}$, there exists a $v^{k+1} \in \partial^L f(x^{k+1})$ which satisfies $\|v^{k+1}\| \leq b \|x^{k+1} - x^k\|$.
- (H3) *Continuity condition*: There exists a subsequence $(x^{k_j})_{j \in \mathbb{N}}$ and $\tilde{x}$ such that $x^{k_j} \to \tilde{x}$ and $f(x^{k_j}) \to f(\tilde{x})$ for $j \to \infty$.

Condition (H0) can be relaxed. The function $f$ only has to fulfill the Kurdyka-Łojasiewicz property at $\tilde{x}$ specified in (H3). We further note that, if all of the above conditions are met, the sequence $(x^k)_{k \in \mathbb{N}}$ has finite length, i. e., $\sum_{k=0}^{\infty} \|x^{k+1} - x^k\| < \infty$. Later on, this will be used to guarantee fast convergence.

### 5.1.1 Preliminaries.

For our convergence analysis, we need a notion of sufficient descent for DCA. The following lemma summarizes previous research (Le An & Tao, 1997). Therein, let $\rho(f)$ denote the modulus of strong convexity for a convex function $f$, i. e., $\rho(f) = \sup\{\rho \geq 0 : f(\cdot) - \frac{\rho}{2}\|\cdot\|^2 \text{ is convex}\}$. In particular, $f$ is strongly convex if $\rho(f) > 0$.

**Lemma 1** (Sufficient descent of DCA)**.** *Let $f = g - h$ with convex functions $g$ and $h$. Furthermore, let $(x^k)_{k \in \mathbb{N}}$ be the sequence generated by DCA. If one of the functions $g$ or $h$ is strongly convex, then $f(x^k) - f(x^{k+1}) \geq (\rho(g) + \rho(h)) \|x^{k+1} - x^k\|^2$.*
*Proof. See Le An & Tao (1997).* $\qquad\qquad\square$

In order to prove that all of the above convergence conditions are fulfilled, we make an additional assumption.

**Assumption 3.** *Let $\mathcal{K}$ be the maximum number of DC iterations in Algorithm 1. We assume that all DCA subroutines for solving* (DC$_l$) *with $l \in \{1, \ldots, N\}$ converge within no more than $\mathcal{K}$ iterations.*

Note that Proposition 7 has already derived the finite convergence of DCA for our DC subproblems. As such, Assumption 3 merely guarantees a uniform upper bound of DC iterations across all DC subproblems. Note also that Assumption 3 is always fulfilled for $\mathcal{K} = \mathcal{K}^{\max}$. Even in the case that restarts are necessary to ensure Assumption 2, Assumption 3 is fulfilled if we set $\mathcal{K} = (\mathcal{K}^{\max})^2$. These bounds are merely rough estimates and can get very large. However, they are by no means tight and in practice setting $\mathcal{K}$ to 50 is already sufficient as demonstrated in our numerical experiments later on.

### 5.2 Proof of Convergence Conditions

In the following, we prove first that the loss function fulfills (H0). We then prove that the sequence $(\theta^k)_{k\in\mathbb{N}}$ generated by Algorithm 1 fulfills each of the conditions (H1), (H2), and (H3).

#### 5.2.1 (H0) KŁ Property of the Loss Function.

The next proposition shows that $\mathrm{RegLoss}_\gamma$ belongs to the class of KŁ functions.

**Proposition 8** (KŁ property of the loss function). *The loss function* $\mathrm{RegLoss}_\gamma$ *is a KŁ function with* $\varphi(s) = C_{\mathrm{KL}} s^{1-\xi}$ *for a constant* $C_{\mathrm{KL}} > 0$ *and* $\xi \in [0, 1)$.
*Proof. See Appendix D.* □

The KŁ property is widely used in optimization. The class of functions that satisfy the KŁ property is large. For instance, it includes all continuous subanalytic functions with closed domain (Bolte et al., 2007).

#### 5.2.2 (H1) Sufficient Decrease Condition.

We now prove that each subproblem solution achieves a sufficient local decrease, that is, fulfills (H1). Afterward, we combine the results to prove the sufficient decrease condition for the sequence $(\theta^k)_{k\in\mathbb{N}}$.

We begin with the DC subproblem. Lemma 1 gives a sufficient descent for DCA if one of the involved functions is strongly convex. The following lemma shows that the modulus of strong convexity of $g_l$ is uniformly bounded from below.

**Lemma 2** (Strong convexity of $g_l$). *Let* $g_l(w_l, b_l)$ *be the convex function defined in Proposition 2. Then,* $\rho(g_l) \geq \frac{2\gamma}{m}\,\sigma_{\min}(M)$ *holds true independently of* $l$.
*Proof. See Appendix D.* □

Due to Assumption 1, $\sigma_{\min}(M)$ is positive and, thus, $g_l$ is strongly convex. The latter now allows to derive the sufficient decrease condition for the DC subproblems.

**Lemma 3** (Sufficient decrease condition for DC subproblems). *Let* $z^j$ *denote the $j$-th iterate of DCA for the $l$-th DC subproblem* $(\mathsf{DC}_l)$ *in the $k$-th outer iteration, i.e.,* $z^0 = (w_l^k, b_l^k)$ *is the starting point and* $z^K = (w_l^{k+1}, b_l^{k+1})$ *the endpoint after $K$ iterations. Then,*

$$g_l(z^0) - h_l(z^0) - \left(g_l(z^K) - h_l(z^K)\right) \geq \frac{2\gamma\sigma_{\min}(M)}{\mathcal{K}m}\|z^K - z^0\|^2 \tag{15}$$

*holds, which is equivalent to*

$$\mathrm{O}_l^{\mathrm{DC}}(w_l^k, b_l^k) - \mathrm{O}_l^{\mathrm{DC}}(w_l^{k+1}, b_l^{k+1}) \geq \frac{2\gamma\sigma_{\min}(M)}{\mathcal{K}m}\left\|(w_l^{k+1}, b_l^{k+1}) - (w_l^k, b_l^k)\right\|^2. \tag{16}$$

*Proof. See Appendix D.* □

Next, we prove the sufficient decrease condition for the alpha subproblem.

**Lemma 4** (Sufficient decrease condition for alpha subproblem). *Let* $\Sigma$ *be given as in Proposition 3 and* $\alpha^{k+1}$ *be the solution of (A). Then,*

$$\mathrm{O}^\alpha(\alpha^k) - \mathrm{O}^\alpha(\alpha^{k+1}) \geq \frac{\gamma}{2}\|\alpha^k - \alpha^{k+1}\|^2. \tag{17}$$

*Proof. See Appendix D.* □

Finally, we combine the results from above to ensure a sufficient decrease in the loss function, as stated in Proposition 9.

**Proposition 9** (Sufficient decrease condition). *Let $(\theta^k)_{k \in \mathbb{N}}$ be the sequence generated by Algorithm 1. Then, $(\theta^k)_{k \in \mathbb{N}}$ satisfies the sufficient decrease condition (H1). That is, there exists an $a > 0$ such that*

$$\text{RegLoss}_\gamma(\theta^{k+1}) + a \, \|\theta^{k+1} - \theta^k\|^2 \leq \text{RegLoss}_\gamma(\theta^k). \tag{18}$$

*Proof. See Appendix D.* □

### 5.2.3 (H2) Relative Error Condition.

We now prove that $(\theta^k)_{k \in \mathbb{N}}$ satisfies (H2). For this, we present some additional intermediate results as follows. First, Lemma 5 proves that $(\theta^k)_{k \in \mathbb{N}}$ stays uniformly bounded during optimization.

**Lemma 5** (Boundedness of $(\theta^k)_{k \in \mathbb{N}}$). *The sequence $(\theta^k)_{k \in \mathbb{N}}$ generated by Algorithm 1 is uniformly bounded by a constant $\Gamma > 0$.*
*Proof. See Appendix D.* □

Second, we show that the terms $\beta_j^{g_l}$ and $\beta_j^{h_l}$ are Lipschitz continuous functions in $\theta$.

**Lemma 6.** *The functions*

$$\beta_j^{g_l}(\theta) = \frac{1}{m} \left( \xi(2y_j\alpha_l) + \sum_{i=l+1}^{N} 2\sigma(\alpha_i\alpha_l)\sigma(\langle w_i, x_j \rangle + b_i) + \sum_{k=1}^{l-1} 2\sigma(\alpha_l\alpha_k)\sigma(\langle w_k, x_j \rangle + b_k) \right), \tag{19}$$

$$\beta_j^{h_l}(\theta) = \frac{1}{m} \left( \sigma(2y_j\alpha_l) + \sum_{i=l+1}^{N} 2\xi(\alpha_i\alpha_l)\sigma(\langle w_i, x_j \rangle + b_i) + \sum_{k=1}^{l-1} 2\xi(\alpha_l\alpha_k)\sigma(\langle w_k, x_j \rangle + b_k) \right), \tag{20}$$

*where $\xi(x) = \max(-x, 0)$ are Lipschitz in $B_\Gamma(0) \subseteq \mathbb{R}^{\mathcal{N}}$.*
*Proof. See Appendix D.* □

Third, Proposition 10 gives a closed-form representation of elements in the limiting subdifferential of the loss function.

**Proposition 10** (Limiting subdifferential of the loss function). *Let $\theta$ be given. Furthermore, let $\epsilon_g = (\epsilon^{g_1}, \ldots, \epsilon^{g_N})$ and $\epsilon_h = (\epsilon^{h_1}, \ldots, \epsilon^{h_N})$ with $\epsilon^{g_l}, \epsilon^{h_l} \in [0,1]^m$ for all $l \in \{1, \ldots, N\}$, and let*

$$\forall j \in \{1, \ldots, m\}: \ \epsilon_j^{g_l}, \epsilon_j^{h_l} \in \begin{cases} \{1\}, & \text{if } \langle w_l, x_j \rangle + b_l \neq 0, \\ [0,1], & \text{else} \end{cases} \tag{21}$$

*and let the condition*

$$0 \leq \beta_j^{g_l}\epsilon_j^{g_l} - \beta_j^{h_l}\epsilon_j^{h_l} \leq \beta_j^{g_l} - \beta_j^{h_l} \text{ for } j \in \{1, \ldots, m\} \text{ with } \langle w_l, x_j \rangle + b_l = 0, \tag{22}$$

*hold true. Then, the vectors*

$$v(\theta, \epsilon_g, \epsilon_h) = \left( (v_{l,t}(\theta, \epsilon^{g_l}, \epsilon^{h_l}))_{\substack{l=1,\ldots,N \\ t=1,\ldots,n}}, (v_l(\theta, \epsilon^{g_l}, \epsilon^{h_l}))_{l=1,\ldots,N}, (v_{\alpha,l}(\theta))_{l=1,\ldots,N} \right), \tag{23}$$

*with entries*

$$v_{l,t}(\theta, \epsilon^{g_l}, \epsilon^{h_l}) = \sum_{j=1}^{m} \beta_j^{g_l} H\left(\langle w_l, x_j \rangle + b_l\right) x_{j,t} \epsilon_j^{g_l} \tag{24}$$

$$+ \sum_{j=1}^{m} 2\frac{\alpha_l^2}{m} H\left(\langle w_l, x_j \rangle + b_l\right) x_{j,t} \left(\langle w_l, x_j \rangle + b_l\right) \tag{25}$$

$$+ \sum_{j=1}^{m} 2\frac{\gamma}{m} x_{j,t} \left(\langle w_l, x_j \rangle + b_l\right) - \sum_{j=1}^{m} \beta_j^{h_l} H\left(\langle w_l, x_j \rangle + b_l\right) x_{j,t} \epsilon_j^{h_l}, \tag{26}$$

$$v_l(\theta, \epsilon^{g_l}, \epsilon^{h_l}) = \sum_{j=1}^{m} \beta_j^{g_l} H\left(\langle w_l, x_j \rangle + b_l\right) \epsilon_j^{g_l} \tag{27}$$

$$+ \sum_{j=1}^{m} 2\frac{\alpha_l^2}{m} H\left(\langle w_l, x_j \rangle + b_l\right) \left(\langle w_l, x_j \rangle + b_l\right) \tag{28}$$

$$+ \sum_{j=1}^{m} 2\frac{\gamma}{m} \left(\langle w_l, x_j \rangle + b_l\right) - \sum_{j=1}^{m} \beta_j^{h_l} H\left(\langle w_l, x_j \rangle + b_l\right) \epsilon_j^{h_l}, \tag{29}$$

$$v_{\alpha,l}(\theta) = \frac{1}{m} \sum_{j=1}^{m} \left[ 2\left(y_j - \sum_{i=1}^{N} \alpha_i \sigma(\langle w_i, x_j \rangle + b_i)\right) \left(-\sigma(\langle w_l, x_j \rangle + b_l)\right) \right] \tag{30}$$

$$+ 2\frac{\gamma}{m} \alpha_l, \tag{31}$$

*are elements of the limiting subdifferential of the loss function, i.e.,*

$$\left\{ v(\theta, \epsilon_g, \epsilon_h) : \epsilon^{g_l}, \epsilon^{h_l} \in [0,1]^m fulfilling~(21)~and~(22)~for~all~l \in \{1, \dots, N\} \right\} \tag{32}$$

*is a subset of $\partial^L \mathrm{RegLoss}_\gamma(\theta)$. Here, $H(x)$ denotes again the Heaviside function, i.e., $H(x) = 1$ if $x \geq 0$ and zero else.*
*Proof. See Appendix D.*

From the proof of Proposition 10, it follows that, for $l \in \{1, \dots, N\}$, the elements

$$\left( (v_{l,t}(\theta, \epsilon^{g_l}, \epsilon^{h_l}))_{t \in \{1, \dots, n\}}, v_l(\theta, \epsilon^{g_l}, \epsilon^{h_l}) \right) = y_g(w_l, b_l, \epsilon^{g_l}) - y_h(w_l, b_l, \epsilon^{h_l}), \tag{33}$$

where $y_g(w_l, b_l, \epsilon^{g_l}) \in \partial g_l(w_l, b_l)$ and $y_h(w_l, b_l, \epsilon^{h_l}) \in \partial h_l(w_l, b_l)$. Now, let $\theta_l^{\mathrm{DC},k}$ denote the parameter vector after the $l$-th DC subproblem when starting with $\theta^k$. From Proposition 7 and Assumption 3, we know that for all $\epsilon^{h_l,k}$ fulfilling (21) for $\theta_l^{\mathrm{DC},k}$ there exists an $\epsilon^{g_l,k}$ fulfilling (21) for $\theta_l^{\mathrm{DC},k}$ such that $y_g(w_l^{k+1}, b_l^{k+1}, \epsilon^{g_l,k}) = y_h(w_l^{k+1}, b_l^{k+1}, \epsilon^{h_l,k})$. When solving additional DC subproblems and the alpha subproblem to reach $\theta^{k+1}$, the terms $\beta_j^{g_l}(\theta_l^{\mathrm{DC},k})$ and $\beta_j^{h_l}(\theta_l^{\mathrm{DC},k})$ change to $\beta_j^{g_l}(\theta^{k+1})$ and $\beta_j^{h_l}(\theta^{k+1})$. To prove condition (H2), we need to make sure that $\left((v_{l,t}(\theta^{k+1}, \epsilon^{g_l,k}, \epsilon^{h_l,k}))_{t \in \{1, \dots, n\}}, v_l(\theta^{k+1}, \epsilon^{g_l,k}, \epsilon^{h_l,k})\right) \in \partial_{(w_l,b_l)}^L \mathrm{RegLoss}_\gamma(\theta^{k+1})$ and, hence, need $\epsilon^{g_l,k}, \epsilon^{h_l,k}$ to fulfill (22) at $\theta^{k+1}$. Thus, we make the following technical assumption.

**Assumption 4** (Differentiability assumption)**.** *For each $k \in \mathbb{N}$ and $l \in \{1, \dots, N\}$, we assume there exists an $\epsilon^{h_l,k}$ such that the corresponding $\epsilon^{g_l,k}$ with*

$$\left( (v_{l,t}(\theta_l^{\mathrm{DC},k}, \epsilon^{g_l,k}, \epsilon^{h_l,k}))_{t \in \{1, \dots, n\}}, v_l(\theta_l^{\mathrm{DC},k}, \epsilon^{g_l,k}, \epsilon^{h_l,k}) \right) = 0 \tag{34}$$

*and $\epsilon^{h_l,k}$ fulfill (22) at $\theta^{k+1}$.*

Now, we can use the above to derive a useful technical property in Lemma 7. The lemma is later used to establish the relative error condition for $(\theta^k)_{k \in \mathbb{N}}$.

**Lemma 7** (Technical lemma). *Let $k \in \mathbb{N}$ and Assumption 3 and Assumption 4 hold. For each $l \in \{1, \ldots, N\}$ there exists a $C > 0$ independent of $l \in \{1, \ldots, N\}$ and $v_l^{k+1} \in \partial_{(w_l, b_l)}^L \mathrm{RegLoss}_\gamma(\theta^{k+1})$ such that*

$$\|v_l^{k+1}\|_1 \leq C\|\theta^{k+1} - \theta_l^{\mathrm{DC},k}\|_1 + C|\alpha_l^{k+1} - \alpha_l^k|. \tag{35}$$

*Proof. See Appendix D.*

Finally, we combine the above results. This yields Proposition 11, which establishes (H2) for $(\theta^k)_{k \in \mathbb{N}}$.

**Proposition 11** (Relative error condition). *Let $(\theta^k)_{k \in \mathbb{N}}$ be the sequence generated by Algorithm 1 and Assumption 3 and 4 hold. Then, $(\theta^k)_{k \in \mathbb{N}}$ satisfies the relative error condition (H2), i.e., there exists a $b > 0$ such that, for all $k \in \mathbb{N}$, there exists a $v^{k+1} \in \partial^L \mathrm{RegLoss}_\gamma(\theta^{k+1})$, which satisfies*

$$\|v^{k+1}\| \leq b \, \|\theta^{k+1} - \theta^k\|. \tag{36}$$

*Proof. See Appendix D.*

### 5.2.4 (H3) Continuity Condition.

Finally, we prove the continuity condition (H3). From Lemma 5, we have that $\|\theta^k\|$ is uniformly bounded for all $k \in \mathbb{N}$. Hence, there exists a convergent subsequence, and, therefore, (H3) follows by the continuity of $\mathrm{RegLoss}_\gamma$.

### 5.3 Global Convergence.

The derivations from the last section are now summarized in Theorem 1.

**Theorem 1** (Global convergence of DCON). *Let Assumption 1 to 4 hold. Furthermore, let $(\theta^k)_{k \in \mathbb{N}}$ be the sequence generated by Algorithm 1. Then, DCON converges to a limiting-critical point of the loss function independent of the initial weights, i.e., $\lim_{k \to \infty} \theta^k = \theta^*$ and $0 \in \partial^L \mathrm{RegLoss}_\gamma(\theta^*)$. Furthermore, the sequence $(\theta^k)_{k \in \mathbb{N}}$ has the finite length property, i.e., $\sum_{k=0}^{\infty} \|\theta^{k+1} - \theta^k\| < \infty$.*
*Proof. Follows from (H0), (H1), (H2) and (H3).*

We note that Theorem 1 yields the global convergence to a limiting-critical point, i.e., the convergence to a limiting-critical point independent of the weight initialization. However, this does not necessarily ensure convergence to local minima. In other words, even if the conditions (H0), (H1), (H2) and (H3) are satisfied, the proximity of the starting point $\theta^0$ to a local minimizer $\theta^*$ does, in general, not imply that the limit is near $\theta^*$. This is owed to the fact that the sequence is not generated by a local model of the objective function (Attouch et al., 2013). However, we can show that DCON converges to a global minimum of $\mathrm{RegLoss}_\gamma$ when $\theta^0$ is sufficiently near; see Proposition 12.

**Proposition 12** (Convergence to global minimum). *Under the assumptions of Theorem 1, the following holds true. If $\theta^* \in \arg\min_\theta \mathrm{RegLoss}_\gamma(\theta)$, there exists a neighborhood $U_{\theta^*}$ of $\theta^*$ such that $\theta^0 \in U_{\theta^*} \Rightarrow \lim_{k \to \infty} \theta^k = \theta^*$. See Appendix D.*

While Assumption 1 to 3 can be well justified, Assumption 4 is quite technical. Nevertheless, DCON converges globally in value under much weaker assumptions.

**Theorem 2** (Global convergence of DCON in value). *Let Assumption 1 hold. Furthermore, let $(\theta^k)_{k \in \mathbb{N}}$ be the sequence generated by Algorithm 1. Then, DCON converges in value, i.e., $(\mathrm{RegLoss}_\gamma(\theta^k))_{k \in \mathbb{N}}$ converges to the infimum $\inf_{k \in \mathbb{N}} \mathrm{RegLoss}_\gamma(\theta^k)$.*
*Proof. Follows from monotone convergence.*

### 5.4 Convergence Rates

According to Theorem 1, the generated sequence has the finite length property. This is usually associated with fast convergence. In the following, we derive the convergence order of DCON, depending on the KŁ exponent $\xi$ of the function $\varphi$ as specified in Proposition 8.

**Proposition 13** (Local convergence of the parameters). *Under the assumptions of Theorem 1, let $\xi$ be the KŁ exponent associated with $\theta^*$. Then, the following holds true:*

- *If $\xi = 0$, DCON converges in finitely many iterations.*
- *If $\xi \in (0, \frac{1}{2}]$, DCON converges R-linearly.*
- *If $\xi \in (\frac{1}{2}, 1)$, DCON converges R-sublinearly.*

*Proof. See Appendix D.*

For machine learning practice, the convergence in value is also of interest. Here, one is also interested in how fast the value of the loss decreases. For DCON, Proposition 14 gives conditions under which the loss sequence $\big(\mathrm{RegLoss}_\gamma(\theta^k)\big)_{k \in \mathbb{N}}$ converges with order $q$.

**Proposition 14** (Local convergence of the loss). *Under the assumptions of Theorem 1, let $\xi$ be the KŁ exponent associated with $\theta^*$. Then, the following holds true:*

- *If $\xi \in (\frac{1}{2(q+1)}, \frac{1}{2q}]$, the loss converges with order $q \in \mathbb{N}$.*
- *If $\xi > \frac{1}{2}$, the loss converges Q-sublinearly.*

*Furthermore, if $\xi \in (\frac{1}{2(q+1)}, \frac{1}{2q})$, we even observe super-Q-convergence. Proof. See Appendix D.*

Proposition 14 shows that DCON can converge very fast in value given a small KŁ exponent $\xi \leq 1/2$. For example, if $\xi \in (\frac{1}{6}, \frac{1}{4})$, the loss converges Q-super-quadratically, while, for $\xi = \frac{1}{4}$, the loss converges Q-quadratically. In the following, we give conditions under which the convergence rate in value can be transferred to the parameters.

**Proposition 15** (Convergence of DCON under local convexity assumption). *Under the assumptions of Theorem 1, let $\xi$ be the KŁ exponent associated with $\theta^*$. If $\mathrm{RegLoss}_\gamma$ admits a neighborhood $U^*$ of $\theta^*$ in which $\mathrm{RegLoss}_\gamma$ is strictly convex, the following holds true: If $\xi \leq \frac{1}{2q}$ for $q \in \mathbb{N}_{\geq 2}$, DCON converges with a Q-convergence order of at least $q - \frac{1}{2}$. Proof. See Appendix D.*

Proposition 13 can be seen as a standard result in the KŁ literature, whereas Proposition 14 and Proposition 15 follow from stronger assumptions on the underlying objective function. For a discussion on how our results are linked to the general KŁ literature, we refer to Appendix E.

In summary, the above results show that DCON can achieve fast convergence of parameters and loss values under mild assumptions. For comparison, given optimal assumptions (i.e., continuously differentiable and strongly convex objective function with Lipschitz continuous gradient), first-order methods converge only linearly (van Scoy et al., 2018), while DCON achieves the same if $\xi = 1/2$ but without any additional assumptions. Evidently, the KŁ exponent $\xi$ is crucial in the above convergence analysis. Determining the Kurdyka-Łojasiewicz exponent for general KŁ functions is still an open research problem. There are works that try to derive calculus rules to determine KŁ exponents under various operations on KŁ functions (Li & Pong, 2018), such as, for instance, the composition (see Theorem 3.2 in Li & Pong, 2018) or block separable sums (see Theorem 3.3 in Li & Pong, 2018) of KŁ functions. Other works determine the KŁ exponent for certain classes of functions, often involving some kind of polynomial representation (Li et al., 2015; Bolte et al., 2017). However, most of these results rely on very strong assumptions on the underlying function, e.g., differentiability or convexity. To the best of our knowledge, there are no results that can be directly used in our – in general – non-differentiable and non-convex setting. Nevertheless, the following proposition gives conditions under which a KŁ exponent $\xi = 1/2$ can be achieved.

**Proposition 16** (KŁ exponent of the loss)**.** *Under the assumptions of Theorem 1, let $(\theta^k)_{k\in\mathbb{N}}$ be the sequence generated by DCON converging to some $\theta^* = (\alpha^*, W^*, b^*)$. If $(W^*, b^*)$ is such that $\langle w_i^*, x_j \rangle + b_i^* \neq 0$ for all $i \in \{1, \ldots, N\}$ and $j \in \{1, \ldots, m\}$, and $\nabla^2 \text{RegLoss}_\gamma(\theta^*)$ is invertible, $\text{RegLoss}_\gamma$ fulfills the KŁ property at $\theta^*$ with $\xi = 1/2$. Proof. See Appendix D.*

## 6 Numerical Experiments

### 6.1 Experimental Setup

Our algorithm is evaluated based on nine datasets (named DS1 to DS9 in the following) that originated from a systematic search. Details can be found in Appendix F.2. In short, we draw upon the UCI machine learning repository[4] and set the filter options to pure regression tasks with numerical attribute type and multivariate data with 100–1000 instances in the training set. Each of the datasets is preprocessed using standard techniques (e. g., scaling of covariates), while taking into account the specifics of each dataset. Details are provided in Appendix F.3.

As a baseline, we consider a state-of-the-art variant of stochastic gradient descent, namely Adam (Kingma & Ba, 2014). On each dataset, we train DCON and the baseline on 30 random train-test splits for three different hidden layer sizes $N \in \{10, 20, 30\}$. Besides that, the neural network architecture has one hyperparameter (regularization parameter $\gamma$), which we tune via grid search. We set the maximum number of DCA iterations to $\mathcal{K} = 50$ and stop DCON after $\mathcal{M} = 1000$ iterations. For Adam, we use early stopping with a patience of 10 epochs and a standard $\ell_2$-regularization. In addition, there are further hyperparameters related to the training algorithm for Adam (i. e., learning rate, first moment exponential decay rate, and batch size). These are also tuned via grid search for each of the $9 \cdot 3 \cdot 30 = 810$ training instances. Details are listed in Appendix F.4. In contrast, comparable hyperparameters related to the training algorithm are absent for DCON.

The results of our main experiments are in Section 6.2. The section reports the prediction performance in terms of mean squared error (MSE), which we average over all 30 runs, i. e., different train-test splits. We further provide a numerical analysis demonstrating our theoretical findings: global convergence guarantees and rate of convergence (Section 6.3). Finally, we show the scalability of DCON by applying it to the MNIST benchmark dataset (LeCun et al., 2010) in Section 6.4.

### 6.2 Overall Numerical Performance

Table 1 reports the relative improvements of DCON in the mean squared error for both the training and test set. On average, our approach outperforms Adam across all layer sizes.

**Training loss.** For the training loss, we find large improvements on almost all datasets and layer sizes. When averaging over all 27 combinations of dataset and layer size, we observe an improvement by a factor of 1.54. For 8 out of 9 datasets, we consistently outperform the baseline by a factor of up to 12.02 (DS6). Only for one dataset (DS7), DCON and Adam are on par (here, Adam is slightly better for a layer size of $N = 10$, whereas the performance of both is comparable for all other layer sizes). Table 1 (bottom row) also lists the average performance improvement per layer size. Here, we see consistent and large improvements, ranging between a factor of 1.19 and a factor of 1.90.

**Test loss.** For the test loss, we see an average improvement by a factor of 0.64 when averaging across all combinations of datasets and layer sizes. On 6 out of 9 datasets, DCON is on par with or even outperforms Adam, showing improvements by a factor of up to 11.81. One further observes a clear performance improvement of DCON for smaller layer sizes, i. e., for $N = 10$ neurons in the hidden layer. For $N = 10$, we obtain an average improvement of a factor 1.61 for the test loss. For $N = 20$ and $N = 30$ neurons in the hidden layer, the improvements still amount to 18 % and 13 %, respectively.

In sum, we confirm numerically that DCON is superior in the training task. DCON yields lower mean squared errors than Adam, often by multiple orders of magnitude. This may be attributed to the properties of DCA, namely that DCA often converges to global solutions (Le An & Tao, 2005). Furthermore, our

---

[4]`https://archive.ics.uci.edu/ml/index.php`, last accessed 03/20/20.

results show that DCON can effectively generalize to unseen data. We think that one reason is the superior training performance, as generalization bounds for regression problems show that lower training losses lead to tighter generalization bounds (e. g., Mohri et al., 2018). We offer a detailed discussion in Appendix F.5.

Table 1: Relative performance improvement in mean squared error of DCON over Adam.

|  | Training | | | | | | Test | | | | | |
|---|---|---|---|---|---|---|---|---|---|---|---|---|
|  | $N = 10$ | | $N = 20$ | | $N = 30$ | | $N = 10$ | | $N = 20$ | | $N = 30$ | |
|  | Mean | (Std.) | Mean | (Std.) | Mean | (Std.) | Mean | (Std.) | Mean | (Std.) | Mean | (Std.) |
| DS1 | 3.69 | (9.27) | 1.79 | (1.08) | 1.76 | (0.52) | 0.80 | (3.00) | 0.21 | (0.54) | 0.12 | (0.67) |
| DS2 | 0.26 | (0.06) | 0.43 | (0.07) | 0.60 | (0.08) | −0.20 | (0.15) | −0.29 | (0.17) | −0.26 | (0.12) |
| DS3 | 0.10 | (0.04) | 0.13 | (0.04) | 0.14 | (0.04) | −0.10 | (0.12) | −0.13 | (0.09) | −0.16 | (0.10) |
| DS4 | 2.41 | (2.86) | 1.54 | (0.95) | 2.18 | (0.86) | 1.72 | (2.25) | 1.14 | (1.01) | 1.16 | (0.85) |
| DS5 | 3.11 | (15.04) | 0.60 | (2.67) | 0.25 | (0.71) | 11.81 | (50.77) | 0.35 | (2.00) | −0.11 | (0.62) |
| DS6 | 3.70 | (0.91) | 12.02 | (3.20) | 5.04 | (0.93) | 0.49 | (0.82) | 0.33 | (0.61) | 0.47 | (0.80) |
| DS7 | −0.01 | (0.09) | 0.00 | (0.06) | 0.00 | (0.05) | −0.04 | (0.09) | −0.01 | (0.11) | −0.02 | (0.08) |
| DS8 | 0.06 | (0.07) | 0.06 | (0.04) | 0.09 | (0.05) | 0.00 | (0.08) | 0.00 | (0.07) | 0.00 | (0.07) |
| DS9 | 0.36 | (0.11) | 0.55 | (0.11) | 0.64 | (0.11) | 0.04 | (0.14) | 0.02 | (0.13) | 0.02 | (0.15) |
| Average | 1.52 | (3.16) | 1.90 | (0.91) | 1.19 | (0.37) | 1.61 | (6.38) | 0.18 | (0.53) | 0.13 | (0.38) |

Results are based on 30 runs with different train-test splits. Reported is the mean performance improvement (e. g., 0.1 means 10 %) and the standard deviation (Std.) in parentheses.

Theoretically, an extremely large number of hidden neurons $N$ can guarantee the convergence of SGD to a global minimum (Du et al., 2019a;b; Zeyuan et al., 2019; Zou & Gu, 2019). To see whether DCON is still beneficial in such an over-parameterized setting, we perform additional experiments in Appendix J.1. Evidently, DCON also benefits from large $N$ and remains superior over SGD. This might be due to the fact that over-parameterization allows to avoid unfavorable local minima in the landscape of the training objective (Zeyuan et al., 2019).

## 6.3 Numerical Analysis of Convergence Behavior

In this section, we perform further numerical experiments to study the convergence behavior of DCON. That is, in the following, we assume that Assumption 1 to 4 hold. We demonstrate that DCON converges to a limiting-critical point. Furthermore, we empirically assess the convergence rate in the training loss and compare it to our theoretical findings from Proposition 14. To do so, we draw upon the neural network architecture (i. e., the tuned regularization parameter) from the previous section and let DCON only terminate upon convergence (i. e., if $\|\theta^{k+1} - \theta^k\|_2 < 10^{-6}$). We then repeat the experiments with this stopping criterion and report results from a single run (i. e., train-test split). To facilitate comparability, the exact same initial weights $\theta^0 = (\alpha^0, W^0, b^0)$ are used for both DCON and Adam. For the same reason, we use a full batch size for Adam to ensure accurate computations of the gradients and mean squared errors.

**Convergence to critical points.** In Figure 1, we demonstrate the convergence of DCON to limiting-critical points. The example shows the convergence for dataset DS1 with $N = 30$. Plots for all other datasets and layer sizes can be found in Appendix F.6. Figure 1a shows how the optimization lets the element in the limiting subdifferential (defined in Proposition 11) approach zero. Figure 1b reports the distance between the parameter vectors from two successive iterations, i. e., $\|\theta^{k+1} - \theta^k\|_2$. As expected, we find that the distance between two successive iterates decreases gradually.

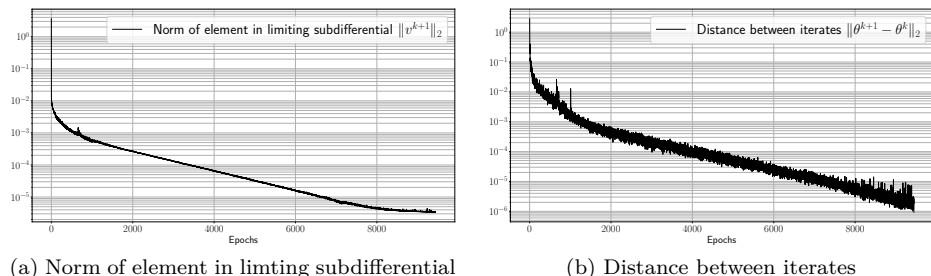

(a) Norm of element in limting subdifferential

(b) Distance between iterates

Figure 1: Convergence to critical points for dataset DS1 ($N = 30$): Results are based on dataset DS1, $N = 30$, and a single train-test split. The term epochs is used to refer to outer iterations. Results for all other datasets and layer sizes are in Appendix F.6. Plot (a) shows the norm of $v(\theta^{k+1}, \epsilon_g, \mathbb{1})$, where $\epsilon_g$ is determined by solving $\min_{\epsilon^{g_l} \in \mathcal{C}} \sum_{t=1}^{n} (v_{l,t}(\theta_l^{\text{DC}}, \epsilon^{g_l}, \mathbb{1}))^2 + (v_l(\theta_l^{\text{DC}}, \epsilon^{g_l}, \mathbb{1}))^2$ after each DC subproblem for each inner iteration. Note that $\mathcal{C}$ decodes the constraints in (21). The rationale is to find the values for $\epsilon^{g_l}$ that set the corresponding entries of $v$ to zero for $\epsilon^{h_l} = \mathbb{1}$, which exist due to Proposition 7. Note that we assume that (22) is fulfilled for $(\epsilon^{g_l}, \mathbb{1})$, i.e., we assume that Assumption 4 holds for $\epsilon^{h_l} = \mathbb{1}$. Plot (b) shows the distance between two successive iterates.

**Rate of convergence.** We now analyze the convergence speed empirically. For this, we compare the mean squared error in the early training phase (here: the first 30 iterations) of DCON and Adam. This is shown in Figure 2a. Evidently, DCON appears to learn faster than Adam in epochs. Here, we adopt the term epoch to report the outer iterations of DCON, as this coincides with the point when each parameter has been updated once. Nevertheless, there is much more optimization involved in an epoch of DCON compared to an epoch of Adam which merely consists of a gradient step. That is, the two curves might not be directly comparable.

We also estimate the convergence order of $\left(\text{RegLoss}_\gamma(\theta^k)\right)_{k \in \mathbb{N}}$ empirically. The results are plotted in Figure 2b. The convergence order is estimated to $q \approx 1.000$, and, hence, we observe linear convergence. In the early phase (first 30 iterations), we observe a faster decay. Similar conclusions can be drawn for most datasets and layer sizes (see Appendix F.6).

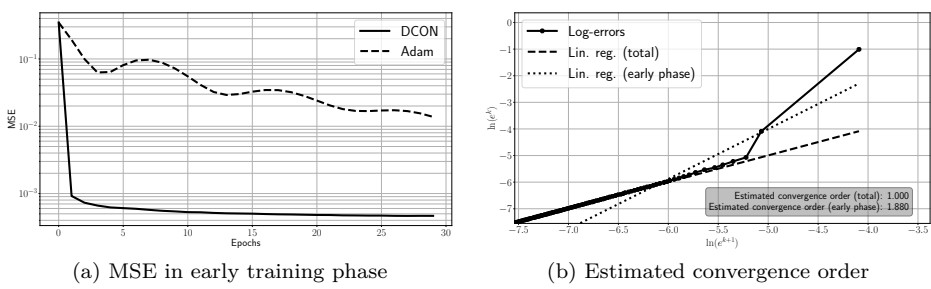

(a) MSE in early training phase

(b) Estimated convergence order

Figure 2: Rate of convergence for dataset DS1 ($N = 30$): Plot (a) shows the mean squared error (MSE) in the early training phase, i.e., the first 30 iterations, of DCON and Adam. In plot (b) we estimate the convergence order of $\left(\text{RegLoss}_\gamma(\theta^k)\right)_{k \in \mathbb{N}}$. That is, we define $e^k = \left|\text{RegLoss}_\gamma(\theta^k) - \text{RegLoss}_\gamma(\theta^*)\right|$ and plot $\ln(e^{k+1})$ against $\ln(e^k)$. Afterward, we fit a linear regression where the slope of the corresponding line gives the estimated order of convergence. We estimate it once in the early training phase and once for all epochs. The estimated convergence order is 1.000.

### 6.4 Scalability

We now demonstrate the scalability of DCON. For this, we leverage the ADMM-based quadratic programing solver proposed in Appendix C. While a quadratic complexity hinders DCON to scale to applications with millions of samples (also due to memory restrictions), it still scales to medium-sized datasets, i. e., with $m < 10,000$. To demonstrate this, we apply DCON to the widely used benchmark dataset MNIST (LeCun et al., 2010). MNIST provides a multi-label classification task, where the inputs are images of handwritten digits (ranging from "0" to "9") based on which the corresponding digit should be predicted. Overall, MNIST comprises of $m = 60,000$ images for training and $m = 10,000$ images for testing.

As MNIST provides a multi-label classification task (while this paper considers a regression task), we train one regression model for each digit, i. e., a one-vs.-all approach, where the correct digit is encoded with a one and the rest with minus one. Implementation details are provided in Appendix G. We then measure the prediction performance on the training set via the mean squared error as we did above. During testing, we combine the predictions from the ten different digit-specific neural networks via an ensemble. The ensemble returns the label corresponding to the neural network for which the prediction is closest to one, yielding a discrete target label. Accordingly, we later report the mean squared error during training (where lower values are better) and the accuracy during testing (where larger values are better).

For our experiments, we use a subset of $m = 10,000$ samples from the MNIST dataset for training. DCON terminates on average within two hours for each digit (compared to 15 minutes for Adam). The runtime drops drastically for smaller datasets, where DCON can also outperform Adam in terms of computing time by several orders of magnitude. To further analyze the limits using current hardware, we also run DCON on the complete MNIST benchmark dataset. Table 2 reports the prediction performances.

In sum, our results show that DCON scales well to medium-sized datasets with $m < 10.000$ observations. For comparison, Lee et al. (2013) use datasets with $m \approx 100$, while datasets in Haugh & Kogan (2004) correspond to $m \approx 4000$.

Table 2: Performance of DCON and Adam for the MNIST benchmark dataset.

| | Training MSE | | | | | | | | | | Test accuracy |
|---|---|---|---|---|---|---|---|---|---|---|---|
| | Digit 0 | Digit 1 | Digit 2 | Digit 3 | Digit 4 | Digit 5 | Digit 6 | Digit 7 | Digit 8 | Digit 9 | Ensemble |
| **MNIST subset** | | | | | | | | | | | |
| Adam | 0.05 | 0.04 | 0.07 | 0.09 | 0.09 | 0.10 | 0.07 | 0.07 | 0.11 | 0.13 | 0.93 |
| DCON | 0.02 | 0.03 | 0.03 | 0.06 | 0.05 | 0.04 | 0.04 | 0.04 | 0.08 | 0.06 | 0.94 |
| Improv. | 1.50 | 0.33 | 1.33 | 0.50 | 0.80 | 1.50 | 0.75 | 0.75 | 0.38 | 1.17 | 0.01 |
| **Complete MNIST dataset** | | | | | | | | | | | |
| Adam | 0.05 | 0.05 | 0.08 | 0.09 | 0.15 | 0.09 | 0.06 | 0.07 | 0.12 | 0.10 | 0.94 |
| DCON | 0.03 | 0.03 | 0.05 | 0.08 | 0.06 | 0.09 | 0.05 | 0.05 | 0.10 | 0.09 | 0.94 |
| Improv. | 0.67 | 0.67 | 0.60 | 0.12 | 1.50 | 0.00 | 0.20 | 0.40 | 0.20 | 0.11 | 0.00 |

Prediction performance is measured via mean squared error (MSE) during training (lower is better) and via accuracy during testing (higher is better). We also report the relative performance improvement of DCON over Adam (e. g., 0.1 means a 10 % improvement of DCON over Adam).

## 7 Conclusions and Future Work

We proposed an algorithm to optimize parameters of single hidden layer feedforward neural networks. Our algorithm is based on a blockwise DC representation of the objective function. The resulting DC subproblems are approached with a tailored difference-of-convex functions algorithm. We proved that DCON converges globally in value and to limiting-critical points under additional assumptions. Furthermore, we analyzed DCON in terms of convergence speed and convergence to global minima.

There are two directions for future work that we think are of particular value. First, Assumption 4 is quite technical and not easy to verify. Here, it might be possible to develop a proof that establishes (H2) without Assumption 4. For this, one might use a more involved analysis using the properties of the limiting subdifferential in Proposition 10 to get rid of condition (22), as by now we are directly working with Fréchet

subdifferentials. Second, research could work on a parallel version of our algorithm and make it scalable to much larger datasets. We provide first theoretical insights in how DCON can be parallelized in Appendix K.1. Our derivations show how the quadratic program from Proposition 5 can be decomposed into a sum of much smaller quadratic programs. A parallel algorithm based on this decomposition can further help counteracting the theoretical computational complexity of $\mathcal{O}(m^2)$.

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

# A    Optimization Problem

## A.1    Proof of Proposition 1

*Proof.* We prove the existence of a solution to (PNN) by showing that the loss function is coercive. Coercivity follows by

$$\text{RegLoss}_\gamma(\theta) \geq \frac{1}{m} \sum_{j=1}^{m} \left( \langle w_l, x_j \rangle + b_l \right)^2 \tag{37}$$

$$= \frac{1}{m} \left\| M \begin{pmatrix} w_l \\ b_l \end{pmatrix} \right\|^2 \geq \frac{\sigma_{\min}(M)}{m} \left\| \begin{pmatrix} w_l \\ b_l \end{pmatrix} \right\|^2, \text{ for all } l \in \{1, \ldots, N\}, \text{ and} \tag{38}$$

$$\text{RegLoss}_\gamma(\theta) \geq \frac{1}{m} \|\alpha\|^2. \tag{39}$$

Using the continuity of $\text{RegLoss}_\gamma(\theta)$, the existence of a solution follows.     □

# B    Derivation of Algorithm

## B.1    Visualization of Subproblems

For better understanding, we visualize the parameters involved in each of the subproblems on a simplified examples with $N = 5$ hidden neurons. Figure 3a shows the parameter notations. Note that the bias terms $b$ are located on the hidden neurons. Figures 3b to 3f show the corresponding DC subproblems, while Figure 3g visualizes the alpha subproblem.

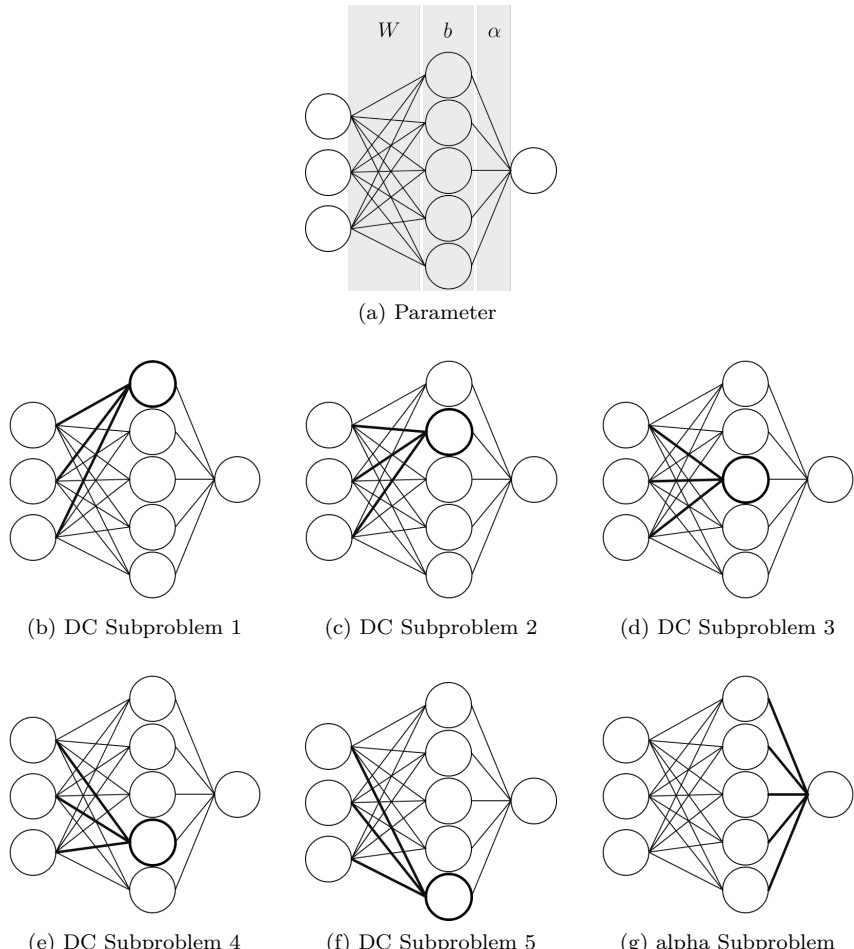

(a) Parameter

(b) DC Subproblem 1            (c) DC Subproblem 2            (d) DC Subproblem 3

(e) DC Subproblem 4            (f) DC Subproblem 5            (g) alpha Subproblem

Figure 3: Visualization of Subproblems.

## B.2 Proof of Proposition 2

*Proof.* To derive the stated decomposition, we proceed as follows:

$$m \, \text{RegLoss}_\gamma(w_l, b_l) = \sum_{j=1}^{m} \left( y_j - \sum_{i=1}^{N} \alpha_i \sigma(\langle w_i, x_j \rangle + b_i) \right)^2 \tag{40}$$

$$+ \gamma \sum_{j=1}^{m} \sum_{i=1}^{N} (\langle w_i, x_j \rangle + b_i)^2 + \gamma \|\alpha\|^2 \tag{41}$$

$$= \sum_{j=1}^{m} \left( y_j^2 - 2y_j \sum_{i=1}^{N} \alpha_i \sigma(\langle w_i, x_j \rangle + b_i) + \left( \sum_{i=1}^{N} \alpha_i \sigma(\langle w_i, x_j \rangle + b_i) \right)^2 \right) \tag{42}$$

$$+ \gamma \sum_{j=1}^{m} \sum_{i=1}^{N} (\langle w_i, x_j \rangle + b_i)^2 + \gamma \|\alpha\|^2 \tag{43}$$

$$= \sum_{j=1}^{m} \left( y_j^2 - 2y_j \sum_{i=1}^{N} \alpha_i \sigma(\langle w_i, x_j \rangle + b_i) + \sum_{i=1}^{N} \alpha_i^2 \sigma(\langle w_i, x_j \rangle + b_i)^2 \right. \tag{44}$$

$$\left. + 2 \sum_{i=1}^{N} \sum_{p=1}^{i-1} \alpha_i \sigma(\langle w_i, x_j \rangle + b_i) \alpha_p \sigma(\langle w_p, x_j \rangle + b_p) \right) \tag{45}$$

$$+ \gamma \sum_{j=1}^{m} \sum_{i=1}^{N} (\langle w_i, x_j \rangle + b_i)^2 + \gamma \|\alpha\|^2 \tag{46}$$

$$= \sum_{j=1}^{m} y_j^2 - \sum_{\substack{i=1 \\ i \neq l}}^{N} \sum_{j=1}^{m} 2y_j \alpha_i \sigma(\langle w_i, x_j \rangle + b_i) - \sum_{j=1}^{m} 2y_j \alpha_l \sigma(\langle w_l, x_j \rangle + b_l) \tag{47}$$

$$+ \sum_{\substack{i=1 \\ i \neq l}}^{N} \sum_{j=1}^{m} \alpha_i^2 \sigma(\langle w_i, x_j \rangle + b_i)^2 + \sum_{j=1}^{m} \alpha_l^2 \sigma(\langle w_l, x_j \rangle + b_l)^2 \tag{48}$$

$$+ \sum_{\substack{i=1 \\ i \neq l}}^{N} \sum_{p=1}^{i-1} \sum_{j=1}^{m} 2\alpha_i \alpha_p \sigma(\langle w_i, x_j \rangle + b_i) \sigma(\langle w_p, x_j \rangle + b_p) \tag{49}$$

$$+ \sum_{p=1}^{l-1} \sum_{j=1}^{m} 2\alpha_l \alpha_p \sigma(\langle w_l, x_j \rangle + b_l) \sigma(\langle w_p, x_j \rangle + b_p) \tag{50}$$

$$+ \gamma \sum_{j=1}^{m} \sum_{\substack{i=1 \\ i \neq l}}^{N} (\langle w_i, x_j \rangle + b_i)^2 + \gamma \sum_{j=1}^{m} (\langle w_l, x_j \rangle + b_l)^2 \tag{51}$$

$$+ \gamma \|\alpha\|^2. \tag{52}$$

Rewriting the term

$$\sum_{\substack{i=1 \\ i \neq l}}^{N} \sum_{p=1}^{i-1} \sum_{j=1}^{m} 2\alpha_i \alpha_p \sigma(\langle w_i, x_j \rangle + b_i) \sigma(\langle w_p, x_j \rangle + b_p) \tag{53}$$

as

$$\sum_{i=1}^{l-1}\sum_{p=1}^{i-1}\sum_{j=1}^{m}2\alpha_i\alpha_p\sigma(\langle w_i,x_j\rangle+b_i)\sigma(\langle w_p,x_j\rangle+b_p) \tag{54}$$

$$+\sum_{i=l+1}^{N}\left(\sum_{\substack{p=1\\p\neq l}}^{i-1}\sum_{j=1}^{m}2\alpha_i\alpha_p\sigma(\langle w_i,x_j\rangle+b_i)\sigma(\langle w_p,x_j\rangle+b_p)\right. \tag{55}$$

$$\left.+\sum_{j=1}^{m}2\alpha_i\alpha_l\sigma(\langle w_i,x_j\rangle+b_i)\sigma(\langle w_l,x_j\rangle+b_l)\right) \tag{56}$$

and defining the constant

$$\tilde{c}_l=\sum_{j=1}^{m}y_j^2-\sum_{\substack{i=1\\i\neq l}}^{N}\sum_{j=1}^{m}2y_j\alpha_i\sigma(\langle w_i,x_j\rangle+b_i) \tag{57}$$

$$+\sum_{\substack{i=1\\i\neq l}}^{N}\sum_{j=1}^{m}\alpha_i^2\sigma(\langle w_i,x_j\rangle+b_i)^2 \tag{58}$$

$$+\sum_{i=1}^{l-1}\sum_{p=1}^{i-1}\sum_{j=1}^{m}2\alpha_i\alpha_p\sigma(\langle w_i,x_j\rangle+b_i)\sigma(\langle w_p,x_j\rangle+b_p) \tag{59}$$

$$+\sum_{i=l+1}^{N}\sum_{\substack{p=1\\p\neq l}}^{i-1}\sum_{j=1}^{m}2\alpha_i\alpha_p\sigma(\langle w_i,x_j\rangle+b_i)\sigma(\langle w_p,x_j\rangle+b_p) \tag{60}$$

$$+\gamma\sum_{j=1}^{m}\sum_{\substack{i=1\\i\neq l}}^{N}(\langle w_i,x_j\rangle+b_i)^2+\gamma\|\alpha\|^2 \tag{61}$$

yields the form

$$\begin{aligned}
m\,\mathrm{RegLoss}_\gamma(w_l,b_l)=\tilde{c}_l&-\sum_{j=1}^{m}2y_j\alpha_l\sigma(\langle w_l,x_j\rangle+b_l)\\
&+\sum_{j=1}^{m}\alpha_l^2\sigma(\langle w_l,x_j\rangle+b_l)^2\\
&+\sum_{i=l+1}^{N}\sum_{j=1}^{m}2\alpha_i\alpha_l\sigma(\langle w_i,x_j\rangle+b_i)\sigma(\langle w_l,x_j\rangle+b_l)\\
&+\sum_{p=1}^{l-1}\sum_{j=1}^{m}2\alpha_l\alpha_p\sigma(\langle w_l,x_j\rangle+b_l)\sigma(\langle w_p,x_j\rangle+b_p)\\
&+\gamma\sum_{j=1}^{m}(\langle w_l,x_j\rangle+b_l)^2\,.
\end{aligned} \tag{62}$$

In Equation (62), we have isolated all terms with $w_l$ and $b_l$. Note also that $\langle w_l,x_j\rangle+b_l$ is linear in $(w_l,b_l)$ and, hence, $\sigma(\langle w_l,x_j\rangle+b_l)$ is convex. Furthermore, the function $\sigma(\langle w_l,x_j\rangle+b_l)^2$ is convex since $\sigma(\langle w_l,x_j\rangle+b_l)$ is non-negative and convex and the function $(\cdot)^2$ is monotonically increasing. To ensure that the linear combinations of those functions are also convex, we have to split the sums into linear combinations involving positive weights and linear combinations involving negative weights. Hence, we define the following index

sets

$$I_{1,l}^+ = \{j \in \{1, \ldots, m\} : 2y_j\alpha_l \geq 0\}, \tag{63}$$

$$I_{1,l}^- = \{1, \ldots, m\} \setminus I_{1,l}^+, \tag{64}$$

$$I_{2,l}^+ = \{(i,j) \in \{l+1, \ldots, N\} \times \{1, \ldots, m\} : 2\alpha_i\alpha_l\sigma(\langle w_i, x_j \rangle + b_i) \geq 0\}, \tag{65}$$

$$I_{2,l}^- = (\{l+1, \ldots, N\} \times \{1, \ldots, m\}) \setminus I_{2,l}^+, \tag{66}$$

$$I_{3,l}^+ = \{(p,j) \in \{1, \ldots, l-1\} \times \{1, \ldots, m\} : 2\alpha_l\alpha_p\sigma(\langle w_p, x_j \rangle + b_p) \geq 0\}, \tag{67}$$

$$I_{3,l}^- = (\{1, \ldots, l-1\} \times \{1, \ldots, m\}) \setminus I_{3,l}^+. \tag{68}$$

By splitting the sums in Equation (62) in the following manner

$$m \, \mathrm{RegLoss}_\gamma(w_l, b_l) = \tilde{c}_l - \sum_{j \in I_{1,l}^+} 2y_j\alpha_l\sigma(\langle w_l, x_j \rangle + b_l) + \sum_{j \in I_{1,l}^-} |2y_j\alpha_l|\sigma(\langle w_l, x_j \rangle + b_l) \tag{69}$$

$$+ \sum_{j=1}^m \alpha_l^2\sigma(\langle w_l, x_j \rangle + b_l)^2 \tag{70}$$

$$+ \sum_{(i,j) \in I_{2,l}^+} 2\alpha_i\alpha_l\sigma(\langle w_i, x_j \rangle + b_i)\sigma(\langle w_l, x_j \rangle + b_l) \tag{71}$$

$$- \sum_{(i,j) \in I_{2,l}^-} |2\alpha_i\alpha_l\sigma(\langle w_i, x_j \rangle + b_i)|\sigma(\langle w_l, x_j \rangle + b_l) \tag{72}$$

$$+ \sum_{(p,j) \in I_{3,l}^+} 2\alpha_l\alpha_p\sigma(\langle w_p, x_j \rangle + b_p)\sigma(\langle w_l, x_j \rangle + b_l) \tag{73}$$

$$- \sum_{(p,j) \in I_{3,l}^-} |2\alpha_l\alpha_p\sigma(\langle w_p, x_j \rangle + b_p)|\sigma(\langle w_l, x_j \rangle + b_l) \tag{74}$$

$$+ \gamma\sum_{j=1}^m (\langle w_l, x_j \rangle + b_l)^2, \tag{75}$$

we yield the form

$$m \, \mathrm{RegLoss}_\gamma(w_l, b_l) = \hat{c}_l + \hat{g}_l(w_l, b_l) - \hat{h}_l(w_l, b_l). \tag{76}$$

Here, the functions $\hat{g}_l$ and $\hat{h}_l$ are defined as

$$\hat{g}_l(w_l, b_l) = \sum_{j \in I_{1,l}^-} |2y_j \alpha_l| \sigma(\langle w_l, x_j \rangle + b_l) \tag{77}$$

$$+ \sum_{j=1}^m \alpha_l^2 \sigma(\langle w_l, x_j \rangle + b_l)^2 \tag{78}$$

$$+ \sum_{(i,j) \in I_{2,l}^+} 2\alpha_i \alpha_l \sigma(\langle w_i, x_j \rangle + b_i) \sigma(\langle w_l, x_j \rangle + b_l) \tag{79}$$

$$+ \sum_{(p,j) \in I_{3,l}^+} 2\alpha_l \alpha_p \sigma(\langle w_p, x_j \rangle + b_p) \sigma(\langle w_l, x_j \rangle + b_l) \tag{80}$$

$$+ \gamma \sum_{j=1}^m (\langle w_l, x_j \rangle + b_l)^2, \tag{81}$$

$$\hat{h}_l(w_l, b_l) = \sum_{j \in I_{1,l}^+} 2y_j \alpha_l \sigma(\langle w_l, x_j \rangle + b_l) \tag{82}$$

$$+ \sum_{(i,j) \in I_{2,l}^-} |2\alpha_i \alpha_l \sigma(\langle w_i, x_j \rangle + b_i)| \sigma(\langle w_l, x_j \rangle + b_l) \tag{83}$$

$$+ \sum_{(p,j) \in I_{3,l}^-} |2\alpha_l \alpha_p \sigma(\langle w_p, x_j \rangle + b_p)| \sigma(\langle w_l, x_j \rangle + b_l), \tag{84}$$

or, in short,

$$\hat{g}_l(w_l, b_l) = \sum_{j=1}^m \hat{\beta}_j^{g_l} \sigma(\langle w_l, x_j \rangle + b_l) + \sum_{j=1}^m \alpha_l^2 \sigma(\langle w_l, x_j \rangle + b_l)^2 + \gamma \sum_{j=1}^m (\langle w_l, x_j \rangle + b_l)^2 \tag{85}$$

$$\hat{h}_l(w_l, b_l) = \sum_{j=1}^m \hat{\beta}_j^{h_l} \sigma(\langle w_l, x_j \rangle + b_l) \tag{86}$$

with coefficients $\hat{\beta}_j^{g_l} \geq 0$ and $\hat{\beta}_j^{h_l} \geq 0$ for all $j \in \{1, \ldots, m\}$. Dividing both sides of Equation (76) by $m$ finally yields the desired form with $\beta_j^{g_l} = \hat{\beta}_j^{g_l}/m$, $\beta_j^{h_l} = \hat{\beta}_j^{h_l}/m$ and $c_l = \hat{c}_l/m$. $\qquad \square$

### B.3 Proof of Proposition 3

*Proof.* By holding all parameters except for $\alpha$ constant in $\text{RegLoss}_\gamma(\alpha, W, b)$, one yields

$$\text{RegLoss}_\gamma(\alpha) = c_\alpha + \frac{1}{m} \|y - \Sigma \alpha\|^2 + \frac{\gamma}{m} \|\alpha\|^2, \tag{87}$$

with

$$c_\alpha = \frac{1}{m} \left( \sum_{j=1}^m \sum_{i=1}^N (\langle w_i, x_j \rangle + b_i)^2 \right). \tag{88}$$

The linear least squares term with system matrix $\Sigma$, as defined in Proposition 3, follows directly by the structure of the mean squared error loss. $\qquad \square$

### B.4 Proof of Proposition 4

*Proof.* Recall that $h_l(w_l, b_l)$ is given by

$$\sum_{j=1}^m \beta_j^{h_l} \sigma \left( \left\langle \begin{pmatrix} w_l \\ b_l \end{pmatrix}, \begin{pmatrix} x_j \\ 1 \end{pmatrix} \right\rangle \right). \tag{89}$$

The proof follows by a straightforward calculation. Let $(\tilde{w}_l, \tilde{b}_l)$ be arbitrary. We have to show that

$$h_l(\tilde{w}_l, \tilde{b}_l) \geq h_l(w_l, b_l) + \left\langle \sum_{j=1}^{m} \beta_j^{h_l} \begin{pmatrix} x_j \\ 1 \end{pmatrix} H\left(\left\langle \begin{pmatrix} w_l \\ b_l \end{pmatrix}, \begin{pmatrix} x_j \\ 1 \end{pmatrix} \right\rangle\right), \begin{pmatrix} \tilde{w}_l \\ \tilde{b}_l \end{pmatrix} - \begin{pmatrix} w_l \\ b_l \end{pmatrix} \right\rangle. \tag{90}$$

The right-hand side of the above inequality can be rewritten as

$$\sum_{j=1}^{m} \beta_j^{h_l} \sigma\left(\left\langle \begin{pmatrix} w_l \\ b_l \end{pmatrix}, \begin{pmatrix} x_j \\ 1 \end{pmatrix} \right\rangle\right) + \sum_{j=1}^{m} \beta_j^{h_l} \left\langle \begin{pmatrix} \tilde{w}_l \\ \tilde{b}_l \end{pmatrix}, \begin{pmatrix} x_j \\ 1 \end{pmatrix} \right\rangle H\left(\left\langle \begin{pmatrix} w_l \\ b_l \end{pmatrix}, \begin{pmatrix} x_j \\ 1 \end{pmatrix} \right\rangle\right) \tag{91}$$

$$- \sum_{j=1}^{m} \beta_j^{h_l} \left\langle \begin{pmatrix} w_l \\ b_l \end{pmatrix}, \begin{pmatrix} x_j \\ 1 \end{pmatrix} \right\rangle H\left(\left\langle \begin{pmatrix} w_l \\ b_l \end{pmatrix}, \begin{pmatrix} x_j \\ 1 \end{pmatrix} \right\rangle\right), \tag{92}$$

which equals

$$\sum_{j=1}^{m} \beta_j^{h_l} \left\langle \begin{pmatrix} \tilde{w}_l \\ \tilde{b}_l \end{pmatrix}, \begin{pmatrix} x_j \\ 1 \end{pmatrix} \right\rangle H\left(\left\langle \begin{pmatrix} w_l \\ b_l \end{pmatrix}, \begin{pmatrix} x_j \\ 1 \end{pmatrix} \right\rangle\right), \tag{93}$$

since

$$\sigma\left(\left\langle \begin{pmatrix} w_l \\ b_l \end{pmatrix}, \begin{pmatrix} x_j \\ 1 \end{pmatrix} \right\rangle\right) = \left\langle \begin{pmatrix} w_l \\ b_l \end{pmatrix}, \begin{pmatrix} x_j \\ 1 \end{pmatrix} \right\rangle H\left(\left\langle \begin{pmatrix} w_l \\ b_l \end{pmatrix}, \begin{pmatrix} x_j \\ 1 \end{pmatrix} \right\rangle\right). \tag{94}$$

Hence, the inequality in Equation (90) is equivalent to

$$\sum_{j=1}^{m} \beta_j^{h_l} \left\langle \begin{pmatrix} \tilde{w}_l \\ \tilde{b}_l \end{pmatrix}, \begin{pmatrix} x_j \\ 1 \end{pmatrix} \right\rangle \left(H\left(\left\langle \begin{pmatrix} \tilde{w}_l \\ \tilde{b}_l \end{pmatrix}, \begin{pmatrix} x_j \\ 1 \end{pmatrix} \right\rangle\right) - H\left(\left\langle \begin{pmatrix} w_l \\ b_l \end{pmatrix}, \begin{pmatrix} x_j \\ 1 \end{pmatrix} \right\rangle\right)\right) \geq 0. \tag{95}$$

Since $\beta_j^{h_l} \geq 0$, this inequality holds if

$$\left\langle \begin{pmatrix} \tilde{w}_l \\ \tilde{b}_l \end{pmatrix}, \begin{pmatrix} x_j \\ 1 \end{pmatrix} \right\rangle \left(H\left(\left\langle \begin{pmatrix} \tilde{w}_l \\ \tilde{b}_l \end{pmatrix}, \begin{pmatrix} x_j \\ 1 \end{pmatrix} \right\rangle\right) - H\left(\left\langle \begin{pmatrix} w_l \\ b_l \end{pmatrix}, \begin{pmatrix} x_j \\ 1 \end{pmatrix} \right\rangle\right)\right) \geq 0, \tag{96}$$

which can be easily verified by a case distinction. $\square$

### B.5  Proof of Proposition 5

*Proof.* For a given $y^* = (y_1^*, y_2^*)$, the convex conjugate of $g_l$ is defined as

$$g_l^*(y_1^*, y_2^*) = \sup_{w_l, b_l} \left\{ \left\langle \begin{pmatrix} y_1^* \\ y_2^* \end{pmatrix}, \begin{pmatrix} w_l \\ b_l \end{pmatrix} \right\rangle - \sum_{j=1}^{m} \beta_j^{g_l} \sigma(\langle w_l, x_j \rangle + b_l) \tag{97}$$

$$- \sum_{j=1}^{m} \frac{\alpha_l^2}{m} \sigma(\langle w_l, x_j \rangle + b_l)^2 \tag{98}$$

$$- \gamma \sum_{j=1}^{m} \frac{1}{m} \left(\langle w_l, x_j \rangle + b_l\right)^2 \right\}. \tag{99}$$

Let the function $F$ be defined as

$$F(w_l, b_l) = \left\langle \begin{pmatrix} y_1^* \\ y_2^* \end{pmatrix}, \begin{pmatrix} w_l \\ b_l \end{pmatrix} \right\rangle - \sum_{j=1}^{m} \beta_j^{g_l} \sigma(\langle w_l, x_j \rangle + b_l) \tag{100}$$

$$- \sum_{j=1}^{m} \frac{\alpha_l^2}{m} \sigma(\langle w_l, x_j \rangle + b_l)^2 \tag{101}$$

$$- \gamma \sum_{j=1}^{m} \frac{1}{m} \left(\langle w_l, x_j \rangle + b_l\right)^2. \tag{102}$$

Using the definition of $M$ and extending the ReLu activation $\sigma$ for vectors $v \in \mathbb{R}^{n+1}$ to

$$\sigma(v) = \begin{pmatrix} \max(v_1, 0) \\ \vdots \\ \max(v_{n+1}, 0) \end{pmatrix}, \tag{103}$$

one can rewrite $F$ in the form

$$F(w_l, b_l) = \left\langle \begin{pmatrix} y_1^* \\ y_2^* \end{pmatrix}, \begin{pmatrix} w_l \\ b_l \end{pmatrix} \right\rangle - \left\langle \beta^{g_l}, \sigma \left( M \begin{pmatrix} w_l \\ b_l \end{pmatrix} \right) \right\rangle \tag{104}$$

$$- \frac{\alpha_l^2}{m} \left\langle \sigma \left( M \begin{pmatrix} w_l \\ b_l \end{pmatrix} \right), \sigma \left( M \begin{pmatrix} w_l \\ b_l \end{pmatrix} \right) \right\rangle \tag{105}$$

$$- \gamma \frac{1}{m} \left\langle M \begin{pmatrix} w_l \\ b_l \end{pmatrix}, M \begin{pmatrix} w_l \\ b_l \end{pmatrix} \right\rangle. \tag{106}$$

For the rest of the proof, we proceed in two cases.

Case 1: $\exists \begin{pmatrix} w_l \\ b_l \end{pmatrix} \in \ker(M) : \left\langle \begin{pmatrix} y_1^* \\ y_2^* \end{pmatrix}, \begin{pmatrix} w_l \\ b_l \end{pmatrix} \right\rangle \neq 0$

In this case, $F$ is unbounded. This can be seen by defining the sequence $(w_l^k, b_l^k) = \lambda k(w_l, b_l)$, where

$$\lambda = \text{sgn}\left( \left\langle \begin{pmatrix} y_1^* \\ y_2^* \end{pmatrix}, \begin{pmatrix} w_l \\ b_l \end{pmatrix} \right\rangle \right). \tag{107}$$

Since $(w_l, b_l) \in \ker(M)$, we have that

$$F(w_l^k, b_l^k) = \lambda k \left\langle \begin{pmatrix} y_1^* \\ y_2^* \end{pmatrix}, \begin{pmatrix} w_l \\ b_l \end{pmatrix} \right\rangle = k \left| \left\langle \begin{pmatrix} y_1^* \\ y_2^* \end{pmatrix}, \begin{pmatrix} w_l \\ b_l \end{pmatrix} \right\rangle \right| \tag{108}$$

and, hence, $F(w_l^k, b_l^k) \to \infty$ for $k \to \infty$. That is, $g_l^*(y_1^*, y_2^*) = \infty$.

Case 2: $\forall \begin{pmatrix} w_l \\ b_l \end{pmatrix} \in \ker(M) : \left\langle \begin{pmatrix} y_1^* \\ y_2^* \end{pmatrix}, \begin{pmatrix} w_l \\ b_l \end{pmatrix} \right\rangle = 0$

In this case, we have that $y^* \in \ker(M)^\perp = \text{Im}(M^T)$. Hence, there exists a $d_{y^*} \in \mathbb{R}^m$ such that $M^T d_{y^*} = y^*$. Leveraging the variable transformation

$$v_{w_l, b_l} = M \begin{pmatrix} w_l \\ b_l \end{pmatrix}, \tag{109}$$

the function $F$ can be rewritten as

$$F(w_l, b_l) = \langle d_{y^*}, v_{w_l, b_l} \rangle - \langle \beta^{g_l}, \sigma(v_{w_l, b_l}) \rangle \tag{110}$$

$$- \frac{\alpha_l^2}{m} \langle \sigma(v_{w_l, b_l}), \sigma(v_{w_l, b_l}) \rangle \tag{111}$$

$$- \gamma \frac{1}{m} \langle v_{w_l, b_l}, v_{w_l, b_l} \rangle. \tag{112}$$

By splitting $v_{w_l, b_l}$ in a positive and negative component, i.e., $v_{w_l, b_l} = v^1 - v^2$ with $0 \leq v^1 \perp v^2 \geq 0$ and using the fact that $\sigma(v_{w_l, b_l}) = v^1$, Equation (110) is given in a quadratic form.[5] However, simply optimiz-

---

[5] The notation $0 \leq v^1 \perp v^2 \geq 0$ is used as a compact expression of a complementary condition on $v^1$ and $v^2$, i.e.,

$$v_i^1 \cdot v_i^2 = 0 \quad \forall i,$$

$$v^1 \geq 0,$$

$$v^2 \geq 0,$$

ing Equation (110) is insufficient, since Equation (109) might not be satisfied. Hence, we add additional constraints in the form

$$\exists\, v^3 \geq 0 \text{ and } v^4 \geq 0 \text{ s.t.: } M\left(v^3 - v^4\right) = v_{w_l, b_l}. \tag{113}$$

Altogether, this yields the following quadratic program

$$\sup \left\langle \begin{pmatrix} d_{y^*} - \beta^{g_l} \\ -d_{y^*} \\ 0 \\ 0 \end{pmatrix}, \begin{pmatrix} v^1 \\ v^2 \\ v^3 \\ v^4 \end{pmatrix} \right\rangle - \frac{1}{m} \begin{pmatrix} v^1 \\ v^2 \\ v^3 \\ v^4 \end{pmatrix}^T \begin{pmatrix} (\alpha_l^2 + \gamma)I & -\gamma I & 0 & 0 \\ -\gamma I & \gamma I & 0 & 0 \\ 0 & 0 & 0 & 0 \\ 0 & 0 & 0 & 0 \end{pmatrix} \begin{pmatrix} v^1 \\ v^2 \\ v^3 \\ v^4 \end{pmatrix}$$

$$\text{s.t.} \quad \begin{pmatrix} -I & I & M & -M \end{pmatrix} \begin{pmatrix} v^1 \\ v^2 \\ v^3 \\ v^4 \end{pmatrix} = 0, \tag{114}$$

$$\begin{pmatrix} v^1 & v^2 & v^3 & v^4 \end{pmatrix} \geq 0.$$

The above quadratic program naturally imposes the complementary conditions of $v^1$ and $v^2$. This can be seen when observing the Karush?Kuhn?Tucker (KKT) conditions. That is, a solution $(v^1, v^2, v^3, v^4)$ of the quadratic program in Equation (114) fulfills

$$\begin{pmatrix} \frac{\alpha_l^2 + \gamma}{m} v^1 - \frac{\gamma}{m} v^2 \\ \frac{\gamma}{m} v^2 - \frac{\gamma}{m} v^1 \\ 0 \\ 0 \end{pmatrix} + \begin{pmatrix} \beta^{g_l} - d_{y^*} \\ d_{y^*} \\ 0 \\ 0 \end{pmatrix} - \begin{pmatrix} \mu_1 \\ \mu_2 \\ \mu_3 \\ \mu_4 \end{pmatrix} + \begin{pmatrix} -\lambda \\ \lambda \\ M^T \lambda \\ -M^T \lambda \end{pmatrix} = \begin{pmatrix} 0 \\ 0 \\ 0 \\ 0 \end{pmatrix}, \tag{115}$$

for some Lagrange multipliers $\mu_1 \geq 0$, $\mu_2 \geq 0$, $\mu_3 \geq 0$, $\mu_4 \geq 0$ and $\lambda$. Furthermore, complementary slackness holds between $v^i$ and $\mu_i$. Solving for $\lambda$ in the first row and inserting it into the second row yields

$$\mu_2 = \frac{\alpha_l^2}{m} v^1 + \beta^{g_l} - \mu_1. \tag{116}$$

Hence, whenever the $i$-th component of $v^1$ is strictly greater than zero, we have

$$(\mu_2)_i = \frac{\alpha_l^2}{m} v_i^1 + \beta_i^{g_l} \geq \frac{\alpha_l^2}{m} v_i^1 > 0, \tag{117}$$

and, therefore, $v_i^2 = 0$ by complementary slackness.

The above-mentioned quadratic program is equivalent to the one stated in Proposition 5 with

$$q_{y^*} = (\beta^{g_l} - d_{y^*}, d_{y^*}, 0, 0), \tag{118}$$

$$Q_l = \frac{2}{m} \begin{pmatrix} (\alpha_l^2 + \gamma)I & -\gamma I & 0 & 0 \\ -\gamma I & \gamma I & 0 & 0 \\ 0 & 0 & 0 & 0 \\ 0 & 0 & 0 & 0 \end{pmatrix}, \tag{119}$$

$$A = \begin{pmatrix} -I & I & M & -M \end{pmatrix}, \tag{120}$$

$$v = (v^1, v^2, v^3, v^4). \tag{121}$$

Hence, $g_l^*(y_1^*, y_2^*)$ is given by the negative optimal objective function value of this quadratic program. The form $g_l^*(y^*) = \chi_\Omega(y^*) - \Xi_l(y^*)$ directly follows from the derivations in case 1 and 2.

---

which says that either $v_i^1$ or $v_i^2$ can be strictly positive.

What remains to be shown is that $\text{dom}(g_l^*) = \Omega$. To do so, we have to show that, for $y^* \in \Omega$, the quadratic program

$$\inf_v \langle q_{y^*}, v \rangle + \frac{1}{2} v^T Q_l v \quad \text{s.t.} \quad Av = 0 \text{ and } v \geq 0, \tag{122}$$

has a finite solution. We prove the claim via contradiction. Suppose there exist $v_k \geq 0$ with $Av_k = 0$ and $\langle q_{y^*}, v_k \rangle + \frac{1}{2} v_k^T Q_l v_k \to -\infty$ for $k \to \infty$. As $\frac{1}{2} v_k^T Q_l v_k \geq 0$ for all $k \in \mathbb{N}$, we yield $\langle q_{y^*}, v_k \rangle \to -\infty$ for $k \to \infty$. Now, we have that

$$\langle q_{y^*}, v_k \rangle = \langle \beta^{g_l} - d_{y^*}, v_k^1 \rangle + \langle d_{y^*}, v_k^2 \rangle \tag{123}$$

$$= \langle \beta^{g_l}, v_k^1 \rangle + \langle d_{y^*}, v_k^2 - v_k^1 \rangle, \tag{124}$$

and, hence, that $\langle d_{y^*}, v_k^2 - v_k^1 \rangle \to -\infty$ as $\langle \beta^{g_l}, v_k^1 \rangle \geq 0$. Then, this yields

$$|\langle d_{y^*}, v_k^2 - v_k^1 \rangle| \geq \frac{1}{2} v_k^T Q_l v_k = \frac{\alpha_l^2}{m} \|v_k^1\|^2 + \frac{\gamma}{m} \|v_k^1 - v_k^2\|^2 \geq \frac{\gamma}{m} \|v_k^1 - v_k^2\|^2, \tag{125}$$

for large $k$, which means

$$\frac{\gamma}{m} \|v_k^1 - v_k^2\|^2 \leq |\langle d_{y^*}, v_k^2 - v_k^1 \rangle| \leq \|d_{y^*}\| \|v_k^1 - v_k^2\|, \tag{126}$$

for large $k$. As $\langle d_{y^*}, v_k^2 - v_k^1 \rangle \to -\infty$, we have that $\|v_k^1 - v_k^2\| \to \infty$, which contradicts Equation (126), as the left-hand side grows quadratically and the right-hand side only linearly. Hence, the quadratic program has a finite solution for all $y^* \in \Omega$. $\qquad\square$

### B.6 Proof of Proposition 6

*Proof.* For $y^* \in \text{dom}(g_l^*) = \Omega$, we have that $\partial g_l^*(y^*) = \partial(-\Xi_l)(y^*)$, where $\Xi_l$ is the value function of (QP). To prove the claim, we make use of Lemma 4.4.1 in Hiriart-Urruty & Lemaréchal (2004). Thus, following Hiriart-Urruty & Lemaréchal (2004), we have

$$(-\Xi_l)(y^*) = \sup\{O_v(y^*) : v \in V\} < \infty, \tag{127}$$

where $O_v(y^*) = -\langle q_{y^*}, v \rangle - \frac{1}{2} v^T Q_l v$ and $V = \{v \in \mathbb{R}^{2m+2(n+1)} : Av = 0 \text{ and } v \geq 0\}$. Note that $O_v$ is convex in $y^*$ as $q_{\lambda y^* + (1-\lambda)z^*} = \lambda q_{y^*} + (1-\lambda) q_{z^*}$ for all $y^*, z^* \in \Omega$ and $\lambda \in [0,1]$. Hence, by Lemma 4.4.1 in Hiriart-Urruty & Lemaréchal (2004), we have

$$\partial(-\Xi_l)(y^*) \supset \overline{\text{co}}\{\cup \partial O_v(y^*) : v \in V(y^*)\}, \tag{128}$$

where $V(y^*) = \{v \in V : O_v(y^*) = (-\Xi_l)(y^*)\}$. Now, we use that

$$y^* + \lambda e_p = M^T d_{y^*} + \lambda e_p = M^T \underbrace{(d_{y^*} + \lambda \Delta_p)}_{=:d_{y^* + \lambda e_p}}, \tag{129}$$

where $\Delta_p$ is a solution of the linear system of equations $M^T \Delta_p = e_p$, which exists due to Assumption 1. We then have

$$q_{y^* + \lambda e_p} = (\beta^{g_l} - d_{y^* + \lambda e_p}, d_{y^* + \lambda e_p}, 0, 0) \tag{130}$$

$$= (\beta^{g_l} - d_{y^*} - \lambda \Delta_p, d_{y^*} + \lambda \Delta_p, 0, 0) \tag{131}$$

$$= q_{y^*} + \lambda(-\Delta_p, \Delta_p, 0, 0) \tag{132}$$

$$= q_{y^*} + \lambda \Delta_{q,p}, \tag{133}$$

where we define $\Delta_{q,p} = (-\Delta_p, \Delta_p, 0, 0)$. Hence, $\nabla_{y^*} O_v(y^*) = (-\langle \Delta_{q,p}, v \rangle)_{p=1,\ldots,n+1}$. As $v_{y^*}$ is a solution to the quadratic program, (128) yields that $\nabla_{y^*} O_{v_{y^*}}(y^*)$ is a valid subgradient, which proves the claim.

Furthermore, one has

$$
-\begin{pmatrix} -\Delta_1^T & \Delta_1^T & 0_{n+1}^T & 0_{n+1}^T \\ -\Delta_2^T & \Delta_2^T & 0_{n+1}^T & 0_{n+1}^T \\ \vdots & \vdots & \vdots & \vdots \\ -\Delta_{n+1}^T & \Delta_{n+1}^T & 0_{n+1}^T & 0_{n+1}^T \end{pmatrix} \begin{pmatrix} v_{y^*}^1 \\ v_{y^*}^2 \\ v_{y^*}^3 \\ v_{y^*}^4 \end{pmatrix} = \begin{pmatrix} \langle \Delta_1, v_{y^*}^1 - v_{y^*}^2 \rangle \\ \vdots \\ \langle \Delta_{n+1}, v_{y^*}^1 - v_{y^*}^2 \rangle \end{pmatrix}
\tag{134}
$$

$$
= \begin{pmatrix} \langle \Delta_1, M(v_{y^*}^3 - v_{y^*}^4) \rangle \\ \vdots \\ \langle \Delta_{n+1}, M(v_{y^*}^3 - v_{y^*}^4) \rangle \end{pmatrix} = \begin{pmatrix} \langle M^T \Delta_1, v_{y^*}^3 - v_{y^*}^4 \rangle \\ \vdots \\ \langle M^T \Delta_{n+1}, v_{y^*}^3 - v_{y^*}^4 \rangle \end{pmatrix}
\tag{135}
$$

$$
= \begin{pmatrix} \langle e_1, v_{y^*}^3 - v_{y^*}^4 \rangle \\ \vdots \\ \langle e_{n+1}, v_{y^*}^3 - v_{y^*}^4 \rangle \end{pmatrix} = v_{y^*}^3 - v_{y^*}^4.
\tag{136}
$$

$\square$

### B.7 Proof of Proposition 7

*Proof.* The analysis is based on the theory of polyhedral DC programming (see, e.g., Le An & Tao, 1997; 2005). A DC program is called polyhedral if either $g$ or $h$ is polyhedral convex. In the following, we summarize important results from polyhedral DC programming in earlier research.

(a) For polyhedral DC programs, DCA converges in finitely many iterations. See (v) in the properties of the simplified DCA in Le An & Tao (2005) or Tao & An (1997).

(b) As we consider a polyhedral DC program, the sequences $(w_l^k, b_l^k)_{k \in \mathbb{N}}$ and $y_k^*$ generated by DCA converge to $((w_l^*, b_l^*), y^*) \in [\partial g_l^*(y^*) \cap \partial h_l^*(y^*)] \times [\partial g_l(w_l^*, b_l^*) \cap \partial h_l(w_l^*, b_l^*)]$. See (iv) in Theorem 6 in Le An & Tao (1997).

(c) If $(w_l^*, b_l^*)$ is a local minimizer of $g_l - h_l$, then $(w_l^*, b_l^*) \in \mathcal{P}_l = \{(w_l, b_l) \in \mathbb{R}^{n+1} : \partial h_l(w_l, b_l) \subset \partial g_l(w_l, b_l)\}$. The converse statement holds true if $h_l$ is a polyhedral convex function. See (ii) in Theorem 1 in Le An & Tao (2005).

(d) $(w_l^*, b_l^*) \in \mathcal{P}_l$ if and only if there exists a $y^* \in \mathcal{S}(w_l^*, b_l^*) = \underset{y \in \partial h_l(w_l^*, b_l^*)}{\arg \min} \{\langle (w_l^*, b_l^*), y \rangle - g_l^*(y)\}$ such that $(w_l^*, b_l^*) \in \partial g_l^*(y^*)$. See (i) in Theorem 3 in Le An & Tao (1997).

Our proof is structured in two parts. In part 1, we first prove that $h_l$ is polyhedral convex. From point (a), it then follows that our DCA routine converges in finitely many iterations. In part 2, we then prove that $((w_l^*, b_l^*), y^*)$ from point (b) fulfills the assumption of point (d), given that $\Xi_l(y^*) \leq \Xi_l(y) \quad \forall y \in \partial h_l(w_l^*, b_l^*)$. By point (c), it then follows that our DCA routine converges to a local solution of ($\mathsf{DC}_l$).

Part 1: To show that $h_l$ is polyhedral convex, we have to show that its epigraph is a polyhedral convex set, i.e., a finite intersection of closed half-spaces (see, e.g., Rockafellar, 1997). To do so, we proceed as follows. Let the sets $A_j^+$, $A_j^-$ for $j \in \{1, \ldots, m\}$ and $\Omega_{\mathcal{J}}$ for $\mathcal{J} \subseteq \{1, \ldots, m\}$ be defined as

$$
A_j^+ = \left\{ \begin{pmatrix} w_l \\ b_l \end{pmatrix} \in \mathbb{R}^{n+1} : \left\langle \begin{pmatrix} x_j \\ 1 \end{pmatrix}, \begin{pmatrix} w_l \\ b_l \end{pmatrix} \right\rangle \geq 0 \right\},
\tag{137}
$$

$$
A_j^- = \left\{ \begin{pmatrix} w_l \\ b_l \end{pmatrix} \in \mathbb{R}^{n+1} : \left\langle \begin{pmatrix} x_j \\ 1 \end{pmatrix}, \begin{pmatrix} w_l \\ b_l \end{pmatrix} \right\rangle \leq 0 \right\},
\tag{138}
$$

$$
\Omega_{\mathcal{J}} = \left\{ \begin{pmatrix} w_l \\ b_l \end{pmatrix} \in \mathbb{R}^{n+1} : \begin{pmatrix} w_l \\ b_l \end{pmatrix} \in A_j^+ \quad \forall j \in \mathcal{J} \text{ and } \begin{pmatrix} w_l \\ b_l \end{pmatrix} \in A_j^- \quad \forall j \in \mathcal{J}^\complement \right\}.
\tag{139}
$$

Furthermore, we denote with $M_{\mathcal{J}}$ the matrix $M$ where all rows corresponding to indices not in $\mathcal{J}$ are set to zero, with $\hat{M}_{\mathcal{J}}$ the matrix

$$\hat{M}_{\mathcal{J}} = \begin{pmatrix} M_{\mathcal{J}} & \vec{0} \\ \vec{0}^T & -1 \end{pmatrix}, \tag{140}$$

and with $\hat{\beta}^{h_l}$ the vector $\hat{\beta}^{h_l} = (\beta^{h_l}, 1)^T$. Now, we have, for $(w_l, b_l) \in \Omega_{\mathcal{J}}$,

$$h_l(w_l, b_l) = \sum_{j=1}^{m} \beta_j^{h_l} \sigma(\langle w_l, x_j \rangle + b_l) = \left\langle \beta^{h_l}, \sigma\left(M\begin{pmatrix} w_l \\ b_l \end{pmatrix}\right)\right\rangle = \left\langle \beta^{h_l}, M_{\mathcal{J}}\begin{pmatrix} w_l \\ b_l \end{pmatrix}\right\rangle, \tag{141}$$

where we used again the vectorized version of $\sigma$ as defined in Equation (103). Thus, for $t \in \mathbb{R}$, it holds

$$h_l(w_l, b_l) \leq t \quad \Leftrightarrow \quad \left\langle \beta^{h_l}, M_{\mathcal{J}}\begin{pmatrix} w_l \\ b_l \end{pmatrix}\right\rangle - t \leq 0 \tag{142}$$

$$\Leftrightarrow \quad \left\langle \hat{\beta}^{h_l}, \hat{M}_{\mathcal{J}}\begin{pmatrix} w_l \\ b_l \\ t \end{pmatrix}\right\rangle \leq 0 \tag{143}$$

$$\Leftrightarrow \quad \left\langle \hat{M}_{\mathcal{J}}^T \hat{\beta}^{h_l}, \begin{pmatrix} w_l \\ b_l \\ t \end{pmatrix}\right\rangle \leq 0. \tag{144}$$

By defining the vector $a_{\mathcal{J}} = \hat{M}_{\mathcal{J}}^T \hat{\beta}^{h_l}$ and using that

$$\mathbb{R}^{n+1} = \bigcup_{\mathcal{J} \in 2^{\{1,\dots,m\}}} \Omega_{\mathcal{J}} \text{ and } \Omega_{\mathcal{J}} = \bigcap_{j \in \mathcal{J}} \bigcap_{i \in \mathcal{J}^{\complement}} A_j^+ \cap A_i^-, \tag{145}$$

where $2^{\{1,\dots,m\}}$ denotes the power set of $\{1,\dots,m\}$, we now have

$$\text{epi}(h_l) = \left\{ \begin{pmatrix} w_l \\ b_l \\ t \end{pmatrix} \in \mathbb{R}^{n+2} : h_l(w_l, b_l) \leq t \right\} \tag{146}$$

$$= \bigcup_{\mathcal{J} \in 2^{\{1,\dots,m\}}} \left\{ \begin{pmatrix} w_l \\ b_l \end{pmatrix} \in \Omega_{\mathcal{J}}, t \in \mathbb{R} : \left\langle a_{\mathcal{J}}, \begin{pmatrix} w_l \\ b_l \\ t \end{pmatrix}\right\rangle \leq 0 \right\} \tag{147}$$

$$= \bigcup_{\mathcal{J} \in 2^{\{1,\dots,m\}}} \bigcap_{j \in \mathcal{J}} \bigcap_{i \in \mathcal{J}^{\complement}} \left\{ \begin{pmatrix} w_l \\ b_l \end{pmatrix} \in A_j^+ \cap A_i^-, t \in \mathbb{R} : \left\langle a_{\mathcal{J}}, \begin{pmatrix} w_l \\ b_l \\ t \end{pmatrix}\right\rangle \leq 0 \right\} \tag{148}$$

$$= \bigcup_{\mathcal{J} \in 2^{\{1,\dots,m\}}} \bigcap_{j \in \mathcal{J}} \bigcap_{i \in \mathcal{J}^{\complement}} \left\{ \begin{pmatrix} w_l \\ b_l \end{pmatrix} \in \mathbb{R}^{n+1}, t \in \mathbb{R} : \left\langle a_{\mathcal{J}}, \begin{pmatrix} w_l \\ b_l \\ t \end{pmatrix}\right\rangle \leq 0, \tag{149}$$

$$\left\langle \begin{pmatrix} -x_j \\ -1 \\ 0 \end{pmatrix}, \begin{pmatrix} w_l \\ b_l \\ t \end{pmatrix}\right\rangle \leq 0, \tag{150}$$

$$\left\langle \begin{pmatrix} x_i \\ 1 \\ 0 \end{pmatrix}, \begin{pmatrix} w_l \\ b_l \\ t \end{pmatrix}\right\rangle \leq 0 \right\} \tag{151}$$

$$= \bigcup_{\mathcal{J} \in 2^{\{1,\dots,m\}}} \bigcap_{j \in \mathcal{J}} \bigcap_{i \in \mathcal{J}^{\complement}} \left\{ \begin{pmatrix} w_l \\ b_l \end{pmatrix} \in \mathbb{R}^{n+1}, t \in \mathbb{R} : \left\langle a_{\mathcal{J}}, \begin{pmatrix} w_l \\ b_l \\ t \end{pmatrix}\right\rangle \leq 0 \right\} \cap \tag{152}$$

$$\left\{ \begin{pmatrix} w_l \\ b_l \end{pmatrix} \in \mathbb{R}^{n+1}, t \in \mathbb{R} : \left\langle \begin{pmatrix} -x_j \\ -1 \\ 0 \end{pmatrix}, \begin{pmatrix} w_l \\ b_l \\ t \end{pmatrix}\right\rangle \leq 0 \right\} \cap \tag{153}$$

$$\left\{ \begin{pmatrix} w_l \\ b_l \end{pmatrix} \in \mathbb{R}^{n+1}, t \in \mathbb{R} : \left\langle \begin{pmatrix} x_i \\ 1 \\ 0 \end{pmatrix}, \begin{pmatrix} w_l \\ b_l \\ t \end{pmatrix}\right\rangle \leq 0 \right\}. \tag{154}$$

Hence, epi($h_l$) is a union of convex polyhedra. From Lemma 1 in Bemporad et al. (2001), it follows that a union of convex polyhedra is convex if and only if it is a convex polyhedron. Since epi($h_l$) is the epigraph of a convex function and thus convex, it follows that epi($h_l$) is a convex polyhedron. This proves the first part.

Note that due to the above derivations one can also see that, independently of $l$ and $(w_l, b_l)$, there are only at most $2^m$ different elements in $\partial h_l(w_l, b_l)$ in the form of Equation (12). Hence, the number of DCA iterations is bounded by $\mathcal{K}^{\max} = 2^m$. See also Theorem 5 in Tao & An (1997) and the arguments therein for fixed respectively natural choices of subgradients.

Part 2: For the second part, we first observe that

$$\partial h_l(w_l, b_l) = \left\{ \sum_{j=1}^{m} \beta_j^{h_l} \begin{pmatrix} x_j \\ 1 \end{pmatrix} H\left( \left\langle \begin{pmatrix} w_l \\ b_l \end{pmatrix}, \begin{pmatrix} x_j \\ 1 \end{pmatrix} \right\rangle \right) \epsilon_j^{h_l} : \ \epsilon_j^{h_l} \in \begin{cases} [0, 1], & \text{if } \left\langle \begin{pmatrix} w_l \\ b_l \end{pmatrix}, \begin{pmatrix} x_j \\ 1 \end{pmatrix} \right\rangle = 0 \\ \{1\}, & \text{else} \end{cases} \right\}. \tag{155}$$

Now, let $((w_l^*, b_l^*), y^*) \in [\partial g_l^*(y^*) \cap \partial h_l^*(y^*)] \times [\partial g_l(w_l^*, b_l^*) \cap \partial h_l(w_l^*, b_l^*)]$ be given as in point (b). By Theorem 6 (i) in Le An & Tao (1997) and $\rho(g_l) + \rho(h_l) > 0$, we know that

$$y^* = \sum_{j=1}^{m} \beta_j^{h_l} \begin{pmatrix} x_j \\ 1 \end{pmatrix} H\left( \left\langle \begin{pmatrix} w_l^* \\ b_l^* \end{pmatrix}, \begin{pmatrix} x_j \\ 1 \end{pmatrix} \right\rangle \right), \tag{156}$$

as DCA converges in finitely many iterations and $(w_l^{k+1}, b_l^{k+1}) = (w_l^k, b_l^k)$. For $y \in \partial h_l(w_l^*, b_l^*)$, we thus have

$$\left\langle y, \begin{pmatrix} w_l^* \\ b_l^* \end{pmatrix} \right\rangle = \sum_{j=1}^{m} \beta_j^{h_l} H\left( \left\langle \begin{pmatrix} w_l^* \\ b_l^* \end{pmatrix}, \begin{pmatrix} x_j \\ 1 \end{pmatrix} \right\rangle \right) \left\langle \begin{pmatrix} w_l^* \\ b_l^* \end{pmatrix}, \begin{pmatrix} x_j \\ 1 \end{pmatrix} \right\rangle \epsilon_j^{h_l} \tag{157}$$

$$= \sum_{j=1}^{m} \beta_j^{h_l} H\left( \left\langle \begin{pmatrix} w_l^* \\ b_l^* \end{pmatrix}, \begin{pmatrix} x_j \\ 1 \end{pmatrix} \right\rangle \right) \left\langle \begin{pmatrix} w_l^* \\ b_l^* \end{pmatrix}, \begin{pmatrix} x_j \\ 1 \end{pmatrix} \right\rangle \tag{158}$$

$$= \left\langle y^*, \begin{pmatrix} w_l^* \\ b_l^* \end{pmatrix} \right\rangle, \tag{159}$$

where the second equality follows by the definition of $\epsilon_j^{h_l}$ in Equation (155). By assumption, we have that $\Xi_l(y^*) \leq \Xi_l(y) \quad \forall y \in \partial h_l(w_l^*, b_l^*)$, which implies that $y^* \in \mathcal{S}(w_l^*, b_l^*)$, as needed in point (d).

In summary, we have that $(w_l^*, b_l^*) \in \partial g_l^*(y^*)$ and $y^* \in \mathcal{S}(w_l^*, b_l^*)$. By point (d), it follows that $(w_l^*, b_l^*) \in \mathcal{P}_l$ and, hence, by point (c), that $(w_l^*, b_l^*)$ is a local minimizer of ($\mathsf{DC}_l$).

$\square$

## B.8 Restart Procedure for our DCA Routine

The point $(w_l^*, b_l^*)$ returned by the DCA routine developed in Section 4.2 is a local solution of ($\mathsf{DC}_l$) if $\partial h_l(w_l^*, b_l^*) \subseteq \partial g_l(w_l^*, b_l^*)$ holds true. If the latter condition is violated, we provide a procedure that allows to further reduce the objective function value by restarting the DCA routine from a new initial point following Tao & An (1998). In that manner, we can ensure that Assumption 2 always holds true.

The main idea is the following. Suppose that there exists a $y^0 \in h_l(w_l^*, b_l^*)$ such that $y^0 \notin g_l(w_l^*, b_l^*)$. Then, Tao & An (1998) show that restarting the DCA routine from the point $((w_l^*, b_l^*), y^0)$ yields a strict decrease in the objective function value in the first iterations, i.e., for $(w_l^1, b_l^1) \in \partial g_l^*(y^0)$ it holds $g(w_l^1, b_l^1) - h(w_l^1, b_l^1) < g(w_l^*, b_l^*) - h(w_l^*, b_l^*)$. Thus, we merely need to provide a procedure to compute $y^0$ for restarting the DCA routine or ensuring that $\partial h_l(w_l^*, b_l^*) \subseteq \partial g_l(w_l^*, b_l^*)$ holds true. We do so in the following.

First, note that

$$\partial g_l(w_l^*, b_l^*) = \left\{ \sum_{j=1}^m \beta_j^{g_l} \begin{pmatrix} x_j \\ 1 \end{pmatrix} H\left(\left\langle \begin{pmatrix} w_l^* \\ b_l^* \end{pmatrix}, \begin{pmatrix} x_j \\ 1 \end{pmatrix} \right\rangle \right) \epsilon_j^{g_l} : \ \epsilon_j^{g_l} \in \begin{cases} [0,1], & \text{if } \left\langle \begin{pmatrix} w_l^* \\ b_l^* \end{pmatrix}, \begin{pmatrix} x_j \\ 1 \end{pmatrix} \right\rangle = 0 \\ \{1\}, & \text{else} \end{cases} \right\} + q, \tag{160}$$

where

$$q = \sum_{j=1}^m 2\frac{\alpha_l^2}{m} H\left(\langle w_l^*, x_j \rangle + b_l^*\right) \begin{pmatrix} x_j \\ 1 \end{pmatrix} \left(\langle w_l^*, x_j \rangle + b_l^*\right) + \sum_{j=1}^m 2\frac{\gamma}{m} \begin{pmatrix} x_j \\ 1 \end{pmatrix} \left(\langle w_l^*, x_j \rangle + b_l^*\right) \tag{161}$$

and

$$\partial h_l(w_l^*, b_l^*) = \left\{ \sum_{j=1}^m \beta_j^{h_l} \begin{pmatrix} x_j \\ 1 \end{pmatrix} H\left(\left\langle \begin{pmatrix} w_l^* \\ b_l^* \end{pmatrix}, \begin{pmatrix} x_j \\ 1 \end{pmatrix} \right\rangle \right) \epsilon_j^{h_l} : \ \epsilon_j^{h_l} \in \begin{cases} [0,1], & \text{if } \left\langle \begin{pmatrix} w_l^* \\ b_l^* \end{pmatrix}, \begin{pmatrix} x_j \\ 1 \end{pmatrix} \right\rangle = 0 \\ \{1\}, & \text{else} \end{cases} \right\}. \tag{162}$$

Now, $\partial h_l(w_l^*, b_l^*) \subseteq \partial g_l(w_l^*, b_l^*)$ always holds true if $\partial h_l(w_l^*, b_l^*)$ is a singleton, i. e., if $(\langle w_l^*, x_j \rangle + b_l^*) \neq 0$ for all $j \in \{1, \dots, m\}$. Hence, we assume that there exists a non-empty subset $\mathcal{J} \subseteq \{1, \dots, m\}$ with $\langle w_l^*, x_j \rangle + b_l^* = 0$ for all $j \in \mathcal{J}$ and $\langle w_l^*, x_j \rangle + b_l^* \neq 0$ for all $j \notin \mathcal{J}$ and yield

$$\partial g_l(w_l^*, b_l^*) = \left\{ \sum_{j \in \mathcal{J}} \beta_j^{g_l} \begin{pmatrix} x_j \\ 1 \end{pmatrix} \epsilon_j^{g_l} : \ \epsilon^{g_l} = \left(\epsilon_j^{g_l}\right)_{j \in \mathcal{J}} \in [0,1]^{|\mathcal{J}|} \right\} + q_g + q, \tag{163}$$

with

$$q_g = \sum_{j \notin \mathcal{J}} \beta_j^{g_l} \begin{pmatrix} x_j \\ 1 \end{pmatrix} H\left(\left\langle \begin{pmatrix} w_l^* \\ b_l^* \end{pmatrix}, \begin{pmatrix} x_j \\ 1 \end{pmatrix} \right\rangle \right), \tag{164}$$

and

$$\partial h_l(w_l^*, b_l^*) = \left\{ \sum_{j \in \mathcal{J}} \beta_j^{h_l} \begin{pmatrix} x_j \\ 1 \end{pmatrix} \epsilon_j^{h_l} : \ \epsilon^{h_l} = \left(\epsilon_j^{h_l}\right)_{j \in \mathcal{J}} \in [0,1]^{|\mathcal{J}|} \right\} + q_h, \tag{165}$$

with

$$q_h = \sum_{j \notin \mathcal{J}} \beta_j^{h_l} \begin{pmatrix} x_j \\ 1 \end{pmatrix} H\left(\left\langle \begin{pmatrix} w_l^* \\ b_l^* \end{pmatrix}, \begin{pmatrix} x_j \\ 1 \end{pmatrix} \right\rangle \right). \tag{166}$$

Next, we define the following matrices

$$M_g = \left( \beta_j^{g_l} \begin{pmatrix} x_j \\ 1 \end{pmatrix} \right)_{j \in \mathcal{J}} \in \mathbb{R}^{(n+1) \times |\mathcal{J}|}, \tag{167}$$

$$M_h = \left( \beta_j^{h_l} \begin{pmatrix} x_j \\ 1 \end{pmatrix} \right)_{j \in \mathcal{J}} \in \mathbb{R}^{(n+1) \times |\mathcal{J}|}. \tag{168}$$

Now, to find $y^0 \in h_l(w_l^*, b_l^*)$ such that $y^0 \notin g_l(w_l^*, b_l^*)$, we need to find $\epsilon^{h_l} \in [0,1]^{|\mathcal{J}|}$ such that, for all $\epsilon^{g_l} \in [0,1]^{|\mathcal{J}|}$, we have

$$M_h \epsilon^{h_l} + q_h \neq M_g \epsilon^{g_l} + q_g + q. \tag{169}$$

To check whether or not such an $\epsilon^{h_l}$ exists, we consider the following max-min problem

$$\max_{\epsilon^{h_l} \in [0,1]^{|\mathcal{J}|}} \min_{\epsilon^{g_l} \in [0,1]^{|\mathcal{J}|}} \|M_h \epsilon^{h_l} + q_h - M_g \epsilon^{g_l} - q_g - q\|_1 \tag{170}$$

If (170) admits a solution $(\epsilon^{h_l}, \epsilon^{h_l})$ with objective function value strictly larger than zero, then $y^0 = M_h \epsilon^{h_l} + q_h \in h_l(w_l^*, b_l^*)$ and $y^0 \notin g_l(w_l^*, b_l^*)$. If the optimal objective function value of (170) is zero, we conclude that $\partial h_l(w_l^*, b_l^*) \subseteq \partial g_l(w_l^*, b_l^*)$ holds true. What remains to be shown is how to solve the max-min problem in (170).

Note that (170) is equivalent to

$$
\begin{aligned}
\max_{\epsilon^{h_l}} \min_{\epsilon^{g_l}, w_1, w_2} \quad & \langle w_1, \mathbb{1} \rangle + \langle w_2, \mathbb{1} \rangle \\
\text{s.t.} \quad & M_h \epsilon^{h_l} + q_h - M_g \epsilon^{g_l} - q_g - q \leq w_1 - w_2, \\
& -M_h \epsilon^{h_l} - q_h + M_g \epsilon^{g_l} + q_g + q \leq w_1 - w_2, \\
& \epsilon^{g_l} \leq \mathbb{1}, \\
& \epsilon^{h_l} \leq \mathbb{1}, \\
& \epsilon^{g_l}, \epsilon^{h_l}, w_1, w_2 \geq 0,
\end{aligned}
\tag{171}
$$

and, thus, can be solved with the branch and bound algorithm developed in Falk (1973). We further note that $|\mathcal{J}|$ is usually small and setting $\epsilon^{h_l} = \mathbb{1}$ yields an initial solution with objective function value zero. That is, if $\partial h_l(w_l^*, b_l^*) \subseteq \partial g_l(w_l^*, b_l^*)$ holds true, we already start with an optimal solution, and if $\partial h_l(w_l^*, b_l^*) \nsubseteq \partial g_l(w_l^*, b_l^*)$, we can terminate the algorithm whenever it yields a feasible solution with objective function value larger than zero. Thus, our restarting procedure can be implemented very efficiently.

## C   Efficient Implementation of QP Solver

In this section, we show how the solution of the quadratic programs in the DCON algorithm can be solved efficiently. It turns out that a single singular value decomposition (SVD) of the matrix $M^T$ can be used in order to solve all quadratic programs by a sequence of basic linear algebra operations. To do so, we note that a quadratic program like (QP) can be solved via the alternating direction method of multipliers (ADMM); see, for instance, Boley (2013). We summarize ADMM for (QP) in Algorithm 2.

---

**Algorithm 2:** ADMM for (QP)

---
**Input:** ADMM parameter $\rho$, maximum number of iterations $\mathcal{L}$
**Output:** Solution $v^*$ to (QP)
1 Set $k \leftarrow 0$
2 Set $z^k, u^k \leftarrow 0$
3 **while** not converged and $k < \mathcal{L}$ **do**
4  $\quad$ Solve $\begin{pmatrix} Q_l + \rho I & A^T \\ A & 0 \end{pmatrix} \begin{pmatrix} v^{k+1} \\ \Phi \end{pmatrix} = \begin{pmatrix} \rho\left(z^k - u^k\right) - q_{y^*} \\ 0 \end{pmatrix}$ for $v^{k+1}$
5  $\quad$ Set $z^{k+1} \leftarrow \max\{0, v^{k+1} + u^k\}$ elementwise
6  $\quad$ Set $u^{k+1} \leftarrow u^k + v^{k+1} - z^{k+1}$
7  $\quad$ Set $k \leftarrow k + 1$
8 **end**
9 **return** $v^{k+1}$

---

In the following, we show how the linear system of equations in line 4 of Algorithm 2 can be solved efficiently. By drawing upon the so-called Schur-complement Nocedal & Wright (see, e. g., 2006), we know that the above KKT matrix can be inverted via the formula

$$\begin{pmatrix} G & A^T \\ A & 0 \end{pmatrix}^{-1} = \begin{pmatrix} C & E \\ E^T & F \end{pmatrix} \text{ with} \tag{172}$$

$$C = G^{-1} - G^{-1}A^T\left(AG^{-1}A^T\right)^{-1}AG^{-1}, \tag{173}$$

$$E = G^{-1}A^T\left(AG^{-1}A^T\right)^{-1}, \tag{174}$$

$$F = -\left(AG^{-1}A^T\right)^{-1}, \tag{175}$$

Now, let $w = \rho\left(z^k - u^k\right) - q_{y^*}$. As the right-hand side of our linear equality constraints in (QP) is zero, the above can be used to simplify line 4 to

$$v^{k+1} = Cw = G^{-1}w - G^{-1}A^T\left(AG^{-1}A^T\right)^{-1}AG^{-1}w. \tag{176}$$

Let $w$ be split into four components according to the dimensions of the submatrices in the definition of $Q_l$ in the proof of Proposition 5, i.e., $w = (w^1, w^2, w^3, w^4)$. In the following, we show how Equation (176) can be computed efficiently.

First, note that $G = Q_l + \rho I$. The simple structure of $Q_l$ now allows to directly compute the inverse of $G$, i.e.,

$$G^{-1} = \begin{pmatrix} g_1 I & g_2 I & 0 & 0 \\ g_2 I & g_3 I & 0 & 0 \\ 0 & 0 & \frac{1}{\rho}I & 0 \\ 0 & 0 & 0 & \frac{1}{\rho}I \end{pmatrix}, \tag{177}$$

where

$$\begin{array}{ccccc} g_1 & = \frac{q_3}{q_1 q_3 - q_2^2} & & q_1 & = \frac{2\left(\alpha_l^2 + \gamma\right)}{m} + \rho, \\ g_2 & = \frac{-q_2}{q_1 q_3 - q_2^2} & \text{and} & q_2 & = -\frac{2\gamma}{m}, \\ g_3 & = \frac{q_1}{q_1 q_3 - q_2^2} & & q_3 & = \frac{2\gamma}{m} + \rho. \end{array}$$

Furthermore, a Cholesky decomposition of the inverse, i. e., $G^{-1} = L^T L$, is given by defining $L$ as

$$L = \begin{pmatrix} \xi_1 I & \xi_2 I & 0 & 0 \\ 0 & \xi_3 I & 0 & 0 \\ 0 & 0 & \xi_4 I & 0 \\ 0 & 0 & 0 & \xi_4 I \end{pmatrix}, \tag{178}$$

where

$$\begin{array}{llll} \xi_1 & = \sqrt{g_1}, & \xi_3 & = \sqrt{\frac{g_1 g_3 - g_2^2}{g_1}}, \\ \xi_2 & = \frac{g_2}{\sqrt{g_1}}, & \xi_4 & = \sqrt{\frac{1}{\rho}}. \end{array}$$

Now, $AG^{-1}A^T = AL^T LA^T = (LA^T)^T LA^T = B^T B$ with

$$B = LA^T = \begin{pmatrix} (\xi_2 - \xi_1) I \\ \xi_3 I \\ \xi_4 M^T \\ -\xi_4 M^T \end{pmatrix}. \tag{179}$$

Suppose we have a singular value decomposition of $M^T$, i. e., $M^T = U\Sigma V^T$. Then,

$$B = \begin{pmatrix} (\xi_2 - \xi_1) I \\ \xi_3 I \\ \xi_4 M^T \\ -\xi_4 M^T \end{pmatrix} = \underbrace{\begin{pmatrix} V & 0 & 0 & 0 \\ 0 & V & 0 & 0 \\ 0 & 0 & U & 0 \\ 0 & 0 & 0 & U \end{pmatrix}}_{\tilde{U}} \begin{pmatrix} (\xi_2 - \xi_1) I \\ \xi_3 I \\ \xi_4 \Sigma \\ -\xi_4 \Sigma \end{pmatrix} V^T. \tag{180}$$

That is,

$$AG^{-1}A^T = B^T B = V \begin{pmatrix} (\xi_2 - \xi_1)I & \xi_3 I & \xi_4 \Sigma^T & -\xi_4 \Sigma^T \end{pmatrix} \tilde{U}^T \tilde{U} \begin{pmatrix} (\xi_2 - \xi_1)I \\ \xi_3 I \\ \xi_4 \Sigma \\ -\xi_4 \Sigma \end{pmatrix} V^T \tag{181}$$

$$= V \left( (\xi_2 - \xi_1)^2 I + \xi_3^2 I + 2\xi_4^2 \Sigma^T \Sigma \right) V^T \tag{182}$$

$$= VDV^T, \tag{183}$$

with $D = (\xi_2 - \xi_1)^2 I + \xi_3^2 I + 2\xi_4^2 \Sigma^T \Sigma$, and, hence,

$$\left( AG^{-1}A^T \right)^{-1} = VD^{-1}V^T. \tag{184}$$

By defining $h = (h^1, h^2, h^3, h^4) = G^{-1}w$, we thus have that $G^{-1}A^T \left( AG^{-1}A^T \right)^{-1} Ah$ equals

$$\begin{pmatrix} (g_2 - g_1)VD^{-1}V^T(h^2 - h^1) + (g_2 - g_1)VD^{-1}V^T M(h^3 - h^4) \\ (g_3 - g_2)VD^{-1}V^T(h^2 - h^1) + (g_3 - g_2)VD^{-1}V^T M(h^3 - h^4) \\ \frac{1}{\rho}M^T VD^{-1}V^T(h^2 - h^1) + \frac{1}{\rho}M^T VD^{-1}V^T M(h^3 - h^4) \\ -\frac{1}{\rho}M^T VD^{-1}V^T(h^2 - h^1) - \frac{1}{\rho}M^T VD^{-1}V^T M(h^3 - h^4) \end{pmatrix} \tag{185}$$

The evaluation of Equation (176) is now stated in Algorithm 3.

Note that the computations in Algorithm 3 are most of the time only scalar-vector products or vector-vector additions, and the matrix-vector products in lines 5–8 can be implemented efficiently. Furthermore, the only terms that change between quadratic programs are the scalars $g_1$, $g_2$, $g_3$, and $\xi_1$, $\xi_2$, $\xi_3$. That is, in the implementation of DCON, we only need one singular value decomposition of $M^T$ in the beginning. All subsequent steps involve only basic linear algebra subroutines that can be implemented efficiently. Note, however, that the memory requirements still may limit the above algorithm, as a singular value decomposition of $M^T$ involves a – in general – dense $m \times m$ matrix $V$.

---

**Algorithm 3:** Evaluation of Equation (176)

---

**Input:** Right hand side vector $w$
**Output:** Next iterate $v = (v^1, v^2, v^3, v^4)$

1  $h^1 \leftarrow g_1 w^1 + g_2 w^2$
2  $h^2 \leftarrow g_2 w^1 + g_3 w^2$
3  $h^3 \leftarrow \frac{1}{\rho} w^3$
4  $h^4 \leftarrow \frac{1}{\rho} w^4$
5  $e^1 \leftarrow V D^{-1} V^T (h^2 - h^1)$
6  $f^1 \leftarrow M^T e^1$
7  $e^2 \leftarrow V D^{-1} V^T M (h^3 - h^4)$
8  $f^2 \leftarrow M^T e^2$
9  $v^1 \leftarrow h^1 - (g_2 - g_1)(e^1 + e^2)$
10 $v^2 \leftarrow h^2 - (g_3 - g_2)(e^1 + e^2)$
11 $v^3 \leftarrow h^3 - \frac{1}{\rho}(f^1 + f^2)$
12 $v^4 \leftarrow h^4 + \frac{1}{\rho}(f^1 + f^2)$
13 **return** $v$

---

## D    Convergence Analysis

### D.1    Preliminaries

For some of the proofs, we need additional concepts and results summarized in the following.

**Definition 3** (Semialgebraic set (Bochnak et al., 1998))**.** *A set $\mathcal{D} \subseteq \mathbb{R}^n$ is called semialgebraic if it can be represented as the finite union of sets of the form*

$$\{x \in \mathbb{R}^n : P_1(x) = \cdots = P_{m_P}(x) = 0, Q_1(x) > 0, \ldots, Q_{m_Q}(x) > 0\}, \tag{186}$$

*where $m_P, m_Q \in \mathbb{N}_0$ and $P_1, \ldots, P_{m_P}, Q_1, \ldots, Q_{m_Q}$ are real polynomial functions.*

**Definition 4** (Semialgebraic function (Bochnak et al., 1998))**.** *A function $f : \mathbb{R}^n \to \mathbb{R}$ is called semialgebraic if its graph*

$$G(f) = \{(x, y) \in \mathbb{R}^{n+1} : y = f(x)\} \tag{187}$$

*is semialgebraic.*

Lemma 8 summarizes some results used in Zeng et al. (2019) that we also need in our convergence analysis. Thereby, we indirectly use the concept of so-called subanalytic functions. Since we are not directly working with subanalytic functions, we refrain from a rigorous definition of subanalyticity and refer to Bolte et al. (2007) for further details.

**Lemma 8.** *The following holds true:*

1. *The composition of semialgebraic functions is semialgebraic (see Proposition 2.2.6 in Bochnak et al. (1998)).*

2. *The sum of semialgebraic functions is semialgebraic (see proof of Proposition 2.2.6 in Bochnak et al. (1998)).*

3. *Semialgebraic functions are subanalytic (see Shiota (1997)).*

4. *If $f : \mathbb{R}^n \to \mathbb{R} \cup \{\infty\}$ is a subanalytic function with closed domain, which is continuous on its domain, then $f$ is a KŁ function (see Theorem 3.1 in Bolte et al. (2007)).*

### D.2    Proof of Proposition 8

*Proof.* The main idea of the proof is to show that $\mathrm{RegLoss}_\gamma$ is semialgebraic. Then, by point 3 in Lemma 8, it follows that $\mathrm{RegLoss}_\gamma$ is subanalytic and, hence, by continuity and point 4 that it is a KŁ function.

To do so, we proceed as follows. First, we rewrite $\text{RegLoss}_\gamma$ as

$$\text{RegLoss}_\gamma(\theta) = \sum_{j=1}^m \Psi_j \left( \sum_{i=1}^N f_{i,j}(\theta) \right) + \sum_{j=1}^m \sum_{i=1}^N \tilde{f}_{i,j}(\theta) + \tilde{\Psi}(\theta), \tag{188}$$

where we use the following definitions

$$\Psi_j(w) = \frac{1}{m}(y_j - w)^2, \tag{189}$$

$$f_{i,j}(\theta) = \alpha_i \sigma(\langle w_i, x_j \rangle + b_i), \tag{190}$$

$$\tilde{f}_{i,j}(\theta) = \frac{\gamma}{m} \left( \langle w_i, x_j \rangle + b_i \right)^2, \text{ and} \tag{191}$$

$$\tilde{\Psi}(\theta) = \frac{\gamma}{m} \|\alpha\|^2. \tag{192}$$

If all the above functions are semialgebraic, point 1 and 2 of Lemma 8 yield that Equation (188) is semialgebraic. That is, we only have to check each of the above functions individually.

The functions $\Psi_j$, for $j \in \{1, \ldots, m\}$, are a one-dimensional polynomial functions and thus trivially semialgebraic. The graph of $f_{i,j}$ can be written as

$$G(f_{i,j}) = \{(\theta, z): \ z = f_{i,j}(\theta)\} \tag{193}$$

$$= \{(\theta, z): \ z = 0 \ , \ -(\langle w_i, x_j \rangle + b_i) > 0\} \tag{194}$$

$$\cup \{(\theta, z): \ \alpha_i (\langle w_i, x_j \rangle + b_i) - z = 0, \ (\langle w_i, x_j \rangle + b_i) > 0\} \tag{195}$$

$$\cup \{(\theta, z): \ z = 0, \ (\langle w_i, x_j \rangle + b_i) = 0\}. \tag{196}$$

The involved functions are all multi-dimensional polynomial functions (at most quadratic) in $\theta$, where coefficients of unused variables are set to zero. Hence, $f_{i,j}$ is semialgebraic. The function $\tilde{f}_{i,j}$ is a multi-dimensional polynomial function and therefore semialgebraic. By writing $\tilde{\Psi}$ as

$$\tilde{\Psi}(\theta) = \sum_{i=1}^N \frac{1}{m} \alpha_i^2, \tag{197}$$

the structure of $\tilde{\Psi}$ is again polynomial, and, hence, it is a semialgebraic function.

Applying point 1 and 2 of Lemma 8 yields that $\text{RegLoss}_\gamma$ is semialgebraic. By point 3 of Lemma 8, it is also subanalytic. Furthermore, the loss function is continuous and, hence, by point 4 of Lemma 8, a KŁ function. The form of $\varphi$ follows directly from Theorem 3.1 in Bolte et al. (2007). □

## D.3 Proof of Lemma 2

*Proof.* First, observe that the first two summands of $g_l$ are not strictly convex. Hence, we only consider the last summand, i.e., we have to show that

$$\sum_{j=1}^m \frac{\gamma}{m}(\langle w_l, x_j \rangle + b_l)^2 \tag{198}$$

is strongly convex. We set $v = (w_l, b_l)$ and proceed as follows:

$$\sum_{j=1}^m \frac{\gamma}{m}(\langle w_l, x_j \rangle + b_l)^2 - \frac{\rho}{2}\|v\|^2 \tag{199}$$

$$= \frac{\gamma}{m}\|Mv\|^2 - \frac{\rho}{2}\|v\|^2 \tag{200}$$

$$= \left\langle v, \left( \frac{\gamma}{m}M^T M - \frac{\rho}{2}I \right)v \right\rangle. \tag{201}$$

The above function is convex if the eigenvalues of the matrix in Equation (201) are positive. The spectrum of this matrix is given by

$$\left\{ \frac{\gamma\sigma}{m} - \frac{\rho}{2} \ : \ \sigma \text{ is a singular value of } M \right\}. \tag{202}$$

By setting the above terms to zero, the claim follows. □

### D.4 Proof of Lemma 3

*Proof.* Rewriting the left-hand side of Equation (15) as a telescope sum yields

$$g_l(z^0) - h_l(z^0) - \left( g_l(z^K) - h_l(z^K) \right) \tag{203}$$

$$= \sum_{j=1}^{K} g_l(z^{j-1}) - h_l(z^{j-1}) - \left( g_l(z^j) - h_l(z^j) \right) \tag{204}$$

$$\geq \sum_{j=1}^{K} (\rho(g_l) + \rho(h_l)) \, \|z^j - z^{j-1}\|^2, \tag{205}$$

where Equation (205) follows by Lemma 1. Furthermore, we bound the sum of squared norms from below by $\sum_{j=1}^{K} \|z^j - z^{j-1}\|^2 \geq \frac{1}{K} \left\| \sum_{j=1}^{K} z^j - z^{j-1} \right\|^2 = \frac{1}{K} \|z^K - z^0\|^2$, which yields

$$g_l(z^0) - h_l(z^0) - \left( g_l(z^K) - h_l(z^K) \right) \geq \frac{\rho(g_l) + \rho(h_l)}{K} \, \|z^K - z^0\|^2 \tag{206}$$

$$\geq \frac{\rho(g_l)}{K} \, \|z^K - z^0\|^2 \tag{207}$$

$$\geq \frac{2\gamma\sigma_{\min}(M)}{Km} \, \|z^K - z^0\|^2 \tag{208}$$

$$\geq \frac{2\gamma\sigma_{\min}(M)}{\mathcal{K}m} \, \|z^K - z^0\|^2, \tag{209}$$

where the last inequality follows from Assumption 3. □

### D.5 Proof of Lemma 4

*Proof.* First, we observe that $\mathrm{O}^\alpha(\alpha)$ is strongly convex with modulus $\gamma$. Hence, the inequality $\mathrm{O}^\alpha(\alpha^k) - \mathrm{O}^\alpha(\alpha^{k+1}) \geq \langle \nabla \mathrm{O}^\alpha(\alpha^{k+1}), \alpha^k - \alpha^{k+1} \rangle + \frac{\gamma}{2} \|\alpha^k - \alpha^{k+1}\|^2$ holds true. Second, the solution of (A) fulfills $\nabla \mathrm{O}^\alpha(\alpha^{k+1}) = 0$, which proves the claim. □

### D.6 Proof of Proposition 9

*Proof.* We proceed as follows. When starting the inner iterations of Algorithm 1 with $\theta^k$, we denote the parameter vector after the $l$-th DC subproblem with $\theta_l^{\mathrm{DC},k}$ and get

$$\mathrm{RegLoss}_\gamma(\theta^k) - \mathrm{RegLoss}_\gamma(\theta^{k+1}) \tag{210}$$

$$= \mathrm{RegLoss}_\gamma(\theta^k) - \sum_{l=1}^{N} \mathrm{RegLoss}_\gamma(\theta_l^{\mathrm{DC},k}) \tag{211}$$

$$+ \sum_{l=1}^{N} \mathrm{RegLoss}_\gamma(\theta_l^{\mathrm{DC},k}) - \mathrm{RegLoss}_\gamma(\theta^{k+1}). \tag{212}$$

After reordering the summands, this yields

$$\text{RegLoss}_\gamma(\theta^k) - \text{RegLoss}_\gamma(\theta^{k+1}) \tag{213}$$

$$= \sum_{l=1}^{N} O_l^{\text{DC}}(w_l^k, b_l^k) - O_l^{\text{DC}}(w_l^{k+1}, b_l^{k+1}) + O^\alpha(\alpha^k) - O^\alpha(\alpha^{k+1}) \tag{214}$$

$$\geq \sum_{l=1}^{N} \frac{2\gamma \sigma_{\min}(M)}{\mathcal{K}m} \|(w_l^{k+1}, b_l^{k+1}) - (w_l^k, b_l^k)\|^2 + \frac{\gamma}{2}\|\alpha^k - \alpha^{k+1}\|^2 \tag{215}$$

$$\geq a \left( \sum_{l=1}^{N} \|(w_l^{k+1}, b_l^{k+1}) - (w_l^k, b_l^k)\|^2 + \|\alpha^k - \alpha^{k+1}\|^2 \right) \tag{216}$$

$$= a \, \|\theta^{k+1} - \theta^k\|^2, \tag{217}$$

where $a = \min\{\frac{2\gamma \sigma_{\min}(M)}{\mathcal{K}m}, \frac{\gamma}{2}\}$. $\qquad\square$

### D.7  Proof of Lemma 5

*Proof.* Note that, due to (H1), Algorithm 1 yields a monotonically decreasing sequence of loss function values, i.e., $\text{RegLoss}_\gamma(\theta^{k+1}) \leq \text{RegLoss}_\gamma(\theta^k)$ for all $k \in \mathbb{N}$. This ensures the boundedness of the sequence $(\theta^k)_{k \in \mathbb{N}}$, since

$$\text{RegLoss}_\gamma(\theta^0) \geq \text{RegLoss}_\gamma(\theta^k) \geq \frac{1}{m} \sum_{j=1}^{m} \left( \langle w_l^k, x_j \rangle + b_l^k \right)^2 \tag{218}$$

$$= \frac{1}{m} \left\| M \begin{pmatrix} w_l^k \\ b_l^k \end{pmatrix} \right\|^2 \geq \frac{\sigma_{\min}(M)}{m} \left\| \begin{pmatrix} w_l^k \\ b_l^k \end{pmatrix} \right\|^2, \tag{219}$$

$$\text{RegLoss}_\gamma(\theta^0) \geq \text{RegLoss}_\gamma(\theta^k) \geq \frac{1}{m}\|\alpha^k\|^2. \tag{220}$$

Hence, all trainable parameters are uniformly bounded. $\qquad\square$

### D.8  Proof of Lemma 6

*Proof.* First, Lemma 5 ensures that $\theta^k \in B_\Gamma(0)$ for all $k \in \mathbb{N}$. Second, we note that the functions

$$\beta_j^{g_l}(\theta) = \frac{1}{m} \left( \xi(2y_j\alpha_l) + \sum_{i=l+1}^{N} 2\sigma(\alpha_i\alpha_l)\sigma(\langle w_i, x_j \rangle + b_i) + \sum_{k=1}^{l-1} 2\sigma(\alpha_l\alpha_k)\sigma(\langle w_k, x_j \rangle + b_k) \right), \tag{221}$$

$$\beta_j^{h_l}(\theta) = \frac{1}{m} \left( \sigma(2y_j\alpha_l) + \sum_{i=l+1}^{N} 2\xi(\alpha_i\alpha_l)\sigma(\langle w_i, x_j \rangle + b_i) + \sum_{k=1}^{l-1} 2\xi(\alpha_l\alpha_k)\sigma(\langle w_k, x_j \rangle + b_k) \right), \tag{222}$$

are sums and products of Lipschitz functions. The claim then follows as sums of Lipschitz functions are Lipschitz and products of bounded Lipschitz functions are Lipschitz. $\qquad\square$

### D.9  Proof of Proposition 10

*Proof.* In the following, we make use of the so-called smooth variational description of Fréchet subgradients detailed in Proposition 17.

**Proposition 17** (Proposition 2.1 in Mordukhovich et al. (2006))**.** *Let* $f : \mathbb{R}^n \to \mathbb{R} \cup \{\infty\}$ *be finite at* $\bar{x}$. *Then,* $x^* \in \partial^F f(\bar{x})$ *if and only if there is a neighborhood* $U$ *of* $\bar{x}$ *and a function* $s : U \to \mathbb{R}$ *which is Fréchet differentiable at* $\bar{x}$ *with derivative* $\nabla s(\bar{x})$ *such that*

$$s(\bar{x}) = f(\bar{x}), \quad \nabla s(\bar{x}) = x^*, \quad and \quad s(x) \leq f(x) \text{ for all } x \in U. \tag{223}$$

To prove our claim, we show that $v(\theta, \epsilon_g, \epsilon_h)$ is an element of the Fréchet subdifferential of the loss function. First, we observe that, due to Proposition 2, we have that

$$\text{RegLoss}_\gamma(w_l, b_l) = c_l + g_l(w_l, b_l) - h_l(w_l, b_l). \tag{224}$$

Second, we have that for all $\epsilon^{g_l}, \epsilon^{h_l}$ fulfilling (21)

$$y_g(w_l, b_l, \epsilon^{g_l}) = \sum_{j=1}^m \beta_j^{g_l} \begin{pmatrix} x_j \\ 1 \end{pmatrix} H\left(\left\langle \begin{pmatrix} w_l \\ b_l \end{pmatrix}, \begin{pmatrix} x_j \\ 1 \end{pmatrix} \right\rangle\right) \epsilon_j^{g_l} + q \in \partial g_l(w_l, b_l), \tag{225}$$

where

$$q = \sum_{j=1}^m 2\frac{\alpha_l^2}{m} H\left(\langle w_l, x_j \rangle + b_l\right) \begin{pmatrix} x_j \\ 1 \end{pmatrix} (\langle w_l, x_j \rangle + b_l) + \sum_{j=1}^m 2\frac{\gamma}{m} \begin{pmatrix} x_j \\ 1 \end{pmatrix} (\langle w_l, x_j \rangle + b_l), \tag{226}$$

and

$$y_h(w_l, b_l, \epsilon^{h_l}) = \sum_{j=1}^m \beta_j^{h_l} \begin{pmatrix} x_j \\ 1 \end{pmatrix} H\left(\left\langle \begin{pmatrix} w_l \\ b_l \end{pmatrix}, \begin{pmatrix} x_j \\ 1 \end{pmatrix} \right\rangle\right) \epsilon_j^{h_l} \in \partial h_l(w_l, b_l). \tag{227}$$

We now show that $y_g(w_l, b_l, \epsilon^{g_l}) - y_h(w_l, b_l, \epsilon^{h_l}) \in \partial^F \text{RegLoss}_\gamma(w_l, b_l)$, where the differential operator corresponds to the partial Fréchet subdifferential with respect to $(w_l, b_l)$, whenever $\epsilon^{g_l}$ and $\epsilon^{h_l}$ fulfill (22). Note that

$$y_g(w_l, b_l, \epsilon^{g_l}) - y_h(w_l, b_l, \epsilon^{h_l}) = \left((v_{l,t}(w_l, b_l, \epsilon^{g_l}, \epsilon^{h_l}))_{t \in \{1,\dots,n\}}, v_l(w_l, b_l, \epsilon^{g_l}, \epsilon^{h_l})\right) \tag{228}$$

That is, with a slight abuse of notation exactly as given in (26) and (29). Note also that $\text{RegLoss}_\gamma$ is differentiable with respect to $\alpha$. Hence, the element $v_{\alpha,l}$ in (31) is exactly the partial derivative with respect to $\alpha$.

To prove our claim, we first note that $\partial^F \text{RegLoss}_\gamma(w_l^0, b_l^0) = \{\nabla \text{RegLoss}_\gamma(w_l^0, b_l^0)\}$ for all $(w_l^0, b_l^0)$ with $\langle w_l^0, x_j \rangle + b_l^0 \neq 0$ for all $j \in \{1, \dots, m\}$. Furthermore, this case yields $y_g(w_l^0, b_l^0, \epsilon^{g_l}) = \nabla g_l(w_l^0, b_l^0)$ and $y_h(w_l^0, b_l^0, \epsilon^{h_l}) = \nabla h_l(w_l^0, b_l^0)$. Thus, the claim directly follows in this case.

Now, assume that we have a $(w_l^0, b_l^0)$ such that there exists a non-empty subset $\mathcal{J} \subseteq \{1, \dots, m\}$ with $\langle w_l^0, x_j \rangle + b_l^0 = 0$ for all $j \in \mathcal{J}$ and $\langle w_l^0, x_j \rangle + b_l^0 \neq 0$ for all $j \notin \mathcal{J}$. By continuity, there exists a neighborhood $U$ of $(w_l^0, b_l^0)$ such that $\langle w_l, x_j \rangle + b_l \neq 0$ and $\text{sign}(\langle w_l, x_j \rangle + b_l) = \text{sign}(\langle w_l^0, x_j \rangle + b_l^0)$ for all $(w_l, b_l) \in U$ and $j \notin \mathcal{J}$. Let $\epsilon^{g_l}, \epsilon^{h_l} \in [0,1]^m$ fulfill (21) and (22), and let the function $s : U \to \mathbb{R}$ be defined as

$$s(w_l, b_l) = c_l + \sum_{j \notin \mathcal{J}} \beta_j^{g_l} \sigma(\langle w_l, x_j \rangle + b_l) + \sum_{j \in \mathcal{J}} \beta_j^{g_l} (\langle w_l, x_j \rangle + b_l) \epsilon_j^{g_l} \tag{229}$$

$$+ \sum_{j=1}^m \alpha_l^2 \sigma(\langle w_l, x_j \rangle + b_l)^2 + \gamma \sum_{j=1}^m (\langle w_l, x_j \rangle + b_l)^2 \tag{230}$$

$$- \sum_{j \notin \mathcal{J}} \beta_j^{h_l} \sigma(\langle w_l, x_j \rangle + b_l) - \sum_{j \in \mathcal{J}} \beta_j^{h_l} (\langle w_l, x_j \rangle + b_l) \epsilon_j^{h_l}. \tag{231}$$

Then, $s$ is differentiable in $U$ and $\nabla s(w_l^0, b_l^0) = y_g(w_l^0, b_l^0, \epsilon^{g_l}) - y_h(w_l^0, b_l^0, \epsilon^{h_l})$ as

$$\frac{\partial}{\partial(w_l, b_l)} \left( \sum_{j \notin \mathcal{J}} \beta_j^{g_l} \sigma(\langle w_l, x_j \rangle + b_l) + \sum_{j \in \mathcal{J}} \beta_j^{g_l} (\langle w_l, x_j \rangle + b_l) \epsilon_j^{g_l} \right) \Bigg|_{(w_l, b_l) = (w_l^0, b_l^0)} \tag{232}$$

$$= \sum_{j \notin \mathcal{J}} \beta_j^{g_l} H \left( \langle w_l^0, x_j \rangle + b_l^0 \right) \binom{x_j}{1} + \sum_{j \in \mathcal{J}} \beta_j^{g_l} \binom{x_j}{1} \epsilon_j^{g_l} \tag{233}$$

$$= \sum_{j \notin \mathcal{J}} \beta_j^{g_l} H \left( \langle w_l^0, x_j \rangle + b_l^0 \right) \binom{x_j}{1} + \sum_{j \in \mathcal{J}} \beta_j^{g_l} H \left( \langle w_l^0, x_j \rangle + b_l^0 \right) \binom{x_j}{1} \epsilon_j^{g_l} \tag{234}$$

$$= \sum_{j=1}^{m} \beta_j^{g_l} H \left( \langle w_l^0, x_j \rangle + b_l^0 \right) \binom{x_j}{1} \epsilon_j^{g_l}, \tag{235}$$

and analogously

$$\frac{\partial}{\partial(w_l, b_l)} \left( - \sum_{j \notin \mathcal{J}} \beta_j^{h_l} \sigma(\langle w_l, x_j \rangle + b_l) - \sum_{j \in \mathcal{J}} \beta_j^{h_l} (\langle w_l, x_j \rangle + b_l) \epsilon_j^{h_l} \right) \Bigg|_{(w_l, b_l) = (w_l^0, b_l^0)} \tag{236}$$

$$= - \sum_{j=1}^{m} \beta_j^{h_l} H \left( \langle w_l^0, x_j \rangle + b_l^0 \right) \epsilon_j^{h_l} \binom{x_j}{1}. \tag{237}$$

To prove that $\nabla s(w_l^0, b_l^0) \in \partial^F \mathrm{RegLoss}_\gamma(w_l^0, b_l^0)$, we make use of Proposition 17. That is, we have to show that

(a)  $s(w_l^0, b_l^0) = \mathrm{RegLoss}_\gamma(w_l^0, b_l^0)$ holds true, and

(b)  $s(w_l, b_l) \leq \mathrm{RegLoss}_\gamma(w_l, b_l)$ for all $(w_l, b_l) \in U$.

For point (a), we arrive at

$$s(w_l^0, b_l^0) = c_l + \sum_{j \notin \mathcal{J}} \beta_j^{g_l} \sigma(\langle w_l^0, x_j \rangle + b_l^0) + \sum_{j \in \mathcal{J}} \beta_j^{g_l} (\langle w_l^0, x_j \rangle + b_l^0) \epsilon_j^{g_l} \tag{238}$$

$$+ \sum_{j=1}^{m} \alpha_l^2 \sigma(\langle w_l^0, x_j \rangle + b_l^0)^2 + \gamma \sum_{j=1}^{m} \left( \langle w_l^0, x_j \rangle + b_l^0 \right)^2 \tag{239}$$

$$- \sum_{j \notin \mathcal{J}} \beta_j^{h_l} \sigma(\langle w_l^0, x_j \rangle + b_l^0) - \sum_{j \in \mathcal{J}} \beta_j^{h_l} (\langle w_l^0, x_j \rangle + b_l^0) \epsilon_j^{h_l} \tag{240}$$

$$= c_l + \sum_{j=1}^{m} \beta_j^{g_l} \sigma(\langle w_l^0, x_j \rangle + b_l^0) \tag{241}$$

$$+ \sum_{j=1}^{m} \alpha_l^2 \sigma(\langle w_l^0, x_j \rangle + b_l^0)^2 + \gamma \sum_{j=1}^{m} \left( \langle w_l^0, x_j \rangle + b_l^0 \right)^2 \tag{242}$$

$$- \sum_{j=1}^{m} \beta_j^{h_l} \sigma(\langle w_l^0, x_j \rangle + b_l^0) \tag{243}$$

$$= c_l + g_l(w_l^0, b_l^0) - h_l(w_l^0, b_l^0) = \mathrm{RegLoss}_\gamma(w_l^0, b_l^0). \tag{244}$$

For point (b), we proceed as follows. Let $(w_l, b_l) \in U$. Then,

$$s(w_l, b_l) = c_l + \sum_{j \notin \mathcal{J}} \beta_j^{g_l} \sigma(\langle w_l, x_j \rangle + b_l) + \sum_{j \in \mathcal{J}} \beta_j^{g_l} (\langle w_l, x_j \rangle + b_l) \epsilon_j^{g_l} \tag{245}$$

$$+ \sum_{j=1}^m \alpha_l^2 \sigma(\langle w_l, x_j \rangle + b_l)^2 + \gamma \sum_{j=1}^m (\langle w_l, x_j \rangle + b_l)^2 \tag{246}$$

$$- \sum_{j \notin \mathcal{J}} \beta_j^{h_l} \sigma(\langle w_l, x_j \rangle + b_l) - \sum_{j \in \mathcal{J}} \beta_j^{h_l} (\langle w_l, x_j \rangle + b_l) \epsilon_j^{h_l} \tag{247}$$

$$\leq c_l + \sum_{j \notin \mathcal{J}} \beta_j^{g_l} \sigma(\langle w_l, x_j \rangle + b_l) + \sum_{j \in \mathcal{J}} \beta_j^{g_l} \sigma(\langle w_l, x_j \rangle + b_l) \tag{248}$$

$$+ \sum_{j=1}^m \alpha_l^2 \sigma(\langle w_l, x_j \rangle + b_l)^2 + \gamma \sum_{j=1}^m (\langle w_l, x_j \rangle + b_l)^2 \tag{249}$$

$$- \sum_{j \notin \mathcal{J}} \beta_j^{h_l} \sigma(\langle w_l, x_j \rangle + b_l) - \sum_{j \in \mathcal{J}} \beta_j^{h_l} \sigma(\langle w_l, x_j \rangle + b_l) \tag{250}$$

$$= \text{RegLoss}_\gamma(w_l, b_l). \tag{251}$$

holds true if and only if

$$\sum_{j \in \mathcal{J}} (\beta_j^{g_l} \epsilon_j^{g_l} - \beta_j^{h_l} \epsilon_j^{h_l})(\langle w_l, x_j \rangle + b_l) \leq \sum_{j \in \mathcal{J}} (\beta_j^{g_l} - \beta_j^{h_l}) \sigma(\langle w_l, x_j \rangle + b_l). \tag{252}$$

For $(\langle w_l, x_j \rangle + b_l) \leq 0$ we have that

$$(\beta_j^{g_l} \epsilon_j^{g_l} - \beta_j^{h_l} \epsilon_j^{h_l})(\langle w_l, x_j \rangle + b_l) \leq 0 = (\beta_j^{g_l} - \beta_j^{h_l}) \sigma(\langle w_l, x_j \rangle + b_l), \tag{253}$$

as $(\beta_j^{g_l} \epsilon_j^{g_l} - \beta_j^{h_l} \epsilon_j^{h_l}) \geq 0$ by (22). For $(\langle w_l, x_j \rangle + b_l) > 0$ we have that

$$(\beta_j^{g_l} \epsilon_j^{g_l} - \beta_j^{h_l} \epsilon_j^{h_l})(\langle w_l, x_j \rangle + b_l) \leq (\beta_j^{g_l} - \beta_j^{h_l}) \sigma(\langle w_l, x_j \rangle + b_l) \Leftrightarrow \tag{254}$$

$$(\beta_j^{g_l} \epsilon_j^{g_l} - \beta_j^{h_l} \epsilon_j^{h_l}) \leq (\beta_j^{g_l} - \beta_j^{h_l}), \tag{255}$$

which again holds by (22). Thus, the inequality in (252) holds true and the claim follows by Proposition 17. $\square$

### D.10 Proof of Lemma 7

*Proof.* Let $k \in \mathbb{N}$ and $l \in \{1, \dots, N\}$ be arbitrary. We denote with $(w_l^{k+1}, b_l^{k+1})$ the element $(w_l, b_l)$ in $\theta^{k+1}$. By Assumption 4 and Proposition 10, we know that

$$v_l^{k+1} = \sum_{j=1}^m \beta_j^{g_l}(\theta^{k+1}) \begin{pmatrix} x_j \\ 1 \end{pmatrix} H\left(\left\langle \begin{pmatrix} w_l^{k+1} \\ b_l^{k+1} \end{pmatrix}, \begin{pmatrix} x_j \\ 1 \end{pmatrix} \right\rangle\right) \epsilon_j^{g_l, k} \tag{256}$$

$$+ \sum_{j=1}^m 2 \frac{(\alpha_l^{k+1})^2}{m} H\left(\left\langle \begin{pmatrix} w_l^{k+1} \\ b_l^{k+1} \end{pmatrix}, \begin{pmatrix} x_j \\ 1 \end{pmatrix} \right\rangle\right) \begin{pmatrix} x_j \\ 1 \end{pmatrix} \left(\left\langle \begin{pmatrix} w_l^{k+1} \\ b_l^{k+1} \end{pmatrix}, \begin{pmatrix} x_j \\ 1 \end{pmatrix} \right\rangle\right) \tag{257}$$

$$+ \sum_{j=1}^m 2 \frac{\gamma}{m} \begin{pmatrix} x_j \\ 1 \end{pmatrix} \left\langle \begin{pmatrix} w_l^{k+1} \\ b_l^{k+1} \end{pmatrix}, \begin{pmatrix} x_j \\ 1 \end{pmatrix} \right\rangle \tag{258}$$

$$- \sum_{j=1}^m \beta_j^{h_l}(\theta^{k+1}) \begin{pmatrix} x_j \\ 1 \end{pmatrix} H\left(\left\langle \begin{pmatrix} w_l^{k+1} \\ b_l^{k+1} \end{pmatrix}, \begin{pmatrix} x_j \\ 1 \end{pmatrix} \right\rangle\right) \epsilon_j^{h_l, k} \in \partial_{(w_l, b_l)}^L \text{RegLoss}_\gamma(\theta^{k+1}), \tag{259}$$

where $\epsilon^{g_l, k}$ and $\epsilon^{h_l, k}$ are given as in Assumption 4. Furthermore, Assumption 3 yields

$$\sum_{j=1}^{m} \beta_j^{g_l}(\theta_l^{\mathrm{DC},k}) \begin{pmatrix} x_j \\ 1 \end{pmatrix} H\left( \left\langle \begin{pmatrix} w_l^{k+1} \\ b_l^{k+1} \end{pmatrix}, \begin{pmatrix} x_j \\ 1 \end{pmatrix} \right\rangle \right) \epsilon_j^{g_l,k} \tag{260}$$

$$+ \sum_{j=1}^{m} 2\frac{(\alpha_l^k)^2}{m} H\left( \left\langle \begin{pmatrix} w_l^{k+1} \\ b_l^{k+1} \end{pmatrix}, \begin{pmatrix} x_j \\ 1 \end{pmatrix} \right\rangle \right) \begin{pmatrix} x_j \\ 1 \end{pmatrix} \left( \left\langle \begin{pmatrix} w_l^{k+1} \\ b_l^{k+1} \end{pmatrix}, \begin{pmatrix} x_j \\ 1 \end{pmatrix} \right\rangle \right) \tag{261}$$

$$+ \sum_{j=1}^{m} 2\frac{\gamma}{m} \begin{pmatrix} x_j \\ 1 \end{pmatrix} \left\langle \begin{pmatrix} w_l^{k+1} \\ b_l^{k+1} \end{pmatrix}, \begin{pmatrix} x_j \\ 1 \end{pmatrix} \right\rangle \tag{262}$$

$$- \sum_{j=1}^{m} \beta_j^{h_l}(\theta_l^{\mathrm{DC},k}) \begin{pmatrix} x_j \\ 1 \end{pmatrix} H\left( \left\langle \begin{pmatrix} w_l^{k+1} \\ b_l^{k+1} \end{pmatrix}, \begin{pmatrix} x_j \\ 1 \end{pmatrix} \right\rangle \right) \epsilon_j^{h_l,k} = 0. \tag{263}$$

Then,

$$v_l^{k+1} = v_l^{k+1} - 0 \tag{264}$$

$$= \sum_{j=1}^{m} \left( \beta_j^{g_l}(\theta^{k+1}) - \beta_j^{g_l}(\theta_l^{\mathrm{DC},k}) \right) \begin{pmatrix} x_j \\ 1 \end{pmatrix} H\left( \left\langle \begin{pmatrix} w_l^{k+1} \\ b_l^{k+1} \end{pmatrix}, \begin{pmatrix} x_j \\ 1 \end{pmatrix} \right\rangle \right) \epsilon_j^{g_l,k} \tag{265}$$

$$+ \sum_{j=1}^{m} 2\frac{(\alpha_l^{k+1})^2 - (\alpha_l^k)^2}{m} H\left( \left\langle \begin{pmatrix} w_l^{k+1} \\ b_l^{k+1} \end{pmatrix}, \begin{pmatrix} x_j \\ 1 \end{pmatrix} \right\rangle \right) \begin{pmatrix} x_j \\ 1 \end{pmatrix} \left( \left\langle \begin{pmatrix} w_l^{k+1} \\ b_l^{k+1} \end{pmatrix}, \begin{pmatrix} x_j \\ 1 \end{pmatrix} \right\rangle \right) \tag{266}$$

$$- \sum_{j=1}^{m} \left( \beta_j^{h_l}(\theta^{k+1}) - \beta_j^{h_l}(\theta_l^{\mathrm{DC},k}) \right) \begin{pmatrix} x_j \\ 1 \end{pmatrix} H\left( \left\langle \begin{pmatrix} w_l^{k+1} \\ b_l^{k+1} \end{pmatrix}, \begin{pmatrix} x_j \\ 1 \end{pmatrix} \right\rangle \right) \epsilon_j^{h_l,k}, \tag{267}$$

and, by writing $(\alpha_l^{k+1})^2 - (\alpha_l^k)^2$ as $(\alpha_l^{k+1} - \alpha_l^k)(\alpha_l^{k+1} + \alpha_l^k)$ and using Lemma 5 and Lemma 6, we yield $C > 0$ independent of $l$ such that

$$\|v_l^{k+1}\|_1 \leq C\|\theta^{k+1} - \theta_l^{\mathrm{DC},k}\|_1 + C|\alpha_l^{k+1} - \alpha_l^k|. \tag{268}$$

$\square$

## D.11   Proof of Proposition 11

*Proof.* First, we observe that $v_\alpha(\theta^{k+1}) = (v_{\alpha,l}(\theta^{k+1}))_{l \in \{1,\ldots,N\}} = \nabla \mathrm{O}^\alpha(\alpha^{k+1}) = 0$. Then, the proof is a direct application of Lemma 7. Let

$$v^{k+1} = (v_1^{k+1}, v_2^{k+1}, \ldots, v_N^{k+1}, 0) \in \partial^L \mathrm{RegLoss}_\gamma(\theta^{k+1}), \tag{269}$$

where $v_l^{k+1}$ for $l \in \{1,\ldots,N\}$ is given as in Lemma 7. Then,

$$\|v^{k+1}\|_1 = \sum_{l=1}^{N} \|v_l^{k+1}\|_1 \leq C \sum_{l=1}^{N} \|\theta^{k+1} - \theta_l^{\mathrm{DC},k}\|_1 + C \sum_{l=1}^{N} |\alpha_l^{k+1} - \alpha_l^k| \tag{270}$$

$$\leq C \sum_{l=1}^{N} \|\theta^{k+1} - \theta^k\|_1 + C\|\alpha^{k+1} - \alpha^k\|_1 \tag{271}$$

$$\leq (N+1)C\|\theta^{k+1} - \theta^k\|_1. \tag{272}$$

Since all norms are equivalent in finite dimensions, this proves the claim. $\square$

### D.12 Proof of Proposition 12

*Proof.* The following is based on the theory in Attouch et al. (2013). First, we note that Assumption 1 implies $\sigma_{\min}(M) > 0$, and, hence, there exists at least one solution to (PNN) by Proposition 1.

Now, let $\theta^* \in \arg\min_\theta \text{RegLoss}_\gamma(\theta)$. The claim follows by Theorem 2.10 and condition (H4) in Attouch et al. (2013). For the sake of clarity, we shortly restate the two statements in our setting.

(H4): For any $\delta > 0$, there exist a $0 < \rho < \delta$ and $\nu > 0$ such that

$$\left.\begin{array}{l} x \in B(x^*, \rho), \ f(x) < f(x^*) + \nu \\ y \notin B(x^*, \rho) \end{array}\right\} \Rightarrow f(x) < f(y) + a\|y - x\|^2,$$

where $a$ is the parameter of the sufficient decrease condition. With the above condition, it is possible to prove the convergence to local minima.

**Theorem 3** (Theorem 2.10 in Attouch et al. (2013)). *Let $f : \mathbb{R}^n \to \mathbb{R} \cup \{\infty\}$ be a proper lower semicontinuous function which satisfies the KŁ property at some local minimizer $x^*$. Assume that (H4) holds at $x^*$. Then, for any $r > 0$, there exist $u \in (0, r)$ and $\mu > 0$ such that the inequalities*

$$\|x^0 - x^*\| < u \qquad and \qquad f(x^*) < f(x^0) < f(x^*) + \mu \tag{273}$$

*imply the following: any sequence $(x^k)_{k \in \mathbb{N}}$ that starts from $x^0$ and that satisfies (H1) and (H2) has (i) the finite length property, (ii) remains in $B(x^*, r)$, and (iii) converges to some $\bar{x} \in B(x^*, r)$ critical point of $f$ with $f(\bar{x}) = f(x^*)$.*

From Remark 2.11 in Attouch et al. (2013), we know that (H4) is satisfied for a local minimum $x^*$ if the function $f$ satisfies

$$f(y) \geq f(x^*) - \frac{a}{4}\|y - x^*\|^2 \text{ for all } y \in \mathbb{R}^n. \tag{274}$$

Now, it is easy to see that, for all $\theta \in \mathbb{R}^\mathcal{N}$, we have

$$\text{RegLoss}_\gamma(\theta) \geq \text{RegLoss}_\gamma(\theta^*) \geq \text{RegLoss}_\gamma(\theta^*) - \frac{a}{4}\|\theta - \theta^*\|^2, \tag{275}$$

and, hence, the claim follows by Theorem 3. $\qquad\qquad\square$

### D.13 Proof of Proposition 13

*Proof.* The proof essentially follows the one in Attouch & Bolte (2009) where it is tailored to the proximal algorithm for nonsmooth functions. Hence, we restate the proof and adapt it to our setting and notation.

As in Attouch & Bolte (2009), we assume w.l.o.g that $\text{RegLoss}_\gamma(\theta^*) = 0$. Now, let $S_k$ denote the tail of the series of distances between iterates, i. e.,

$$S_k = \sum_{i=k}^\infty \|\theta^{i+1} - \theta^i\|. \tag{276}$$

Then, we first observe that

$$\|\theta^k - \theta^*\| \leq \|\theta^k - \theta^{k+1}\| + \|\theta^{k+1} - \theta^*\| \leq \ \ldots \tag{277}$$

$$\leq \sum_{i=k}^K \|\theta^{i+1} - \theta^i\| + \|\theta^{K+1} - \theta^*\| \xrightarrow[K \to \infty]{} S_k. \tag{278}$$

Hence, it is sufficient to bound $S_k$. To do so, we prove the intermediate result

$$S_k \leq \frac{r}{1-r}\|\theta^k - \theta^{k-1}\| + \frac{C_1}{r(1-r)}\text{RegLoss}_\gamma(\theta^k)^{1-\xi}, \tag{279}$$

for a constant $C_1 > 0$, $r \in (0,1)$, $\xi$ as specified in the KŁ property, and for $k$ sufficiently large.

Let $\varphi(s) = C_{\mathrm{KL}} s^{1-\xi}$ be the function specified in the KŁ property. Then, we have

$$\varphi(\mathrm{RegLoss}_\gamma(\theta^i)) - \varphi(\mathrm{RegLoss}_\gamma(\theta^{i+1})) \tag{280}$$

$$\geq \varphi'(\mathrm{RegLoss}_\gamma(\theta^i)) \left(\mathrm{RegLoss}_\gamma(\theta^i) - \mathrm{RegLoss}_\gamma(\theta^{i+1})\right) \tag{281}$$

$$\geq \varphi'(\mathrm{RegLoss}_\gamma(\theta^i)) a \|\theta^{i+1} - \theta^i\|^2, \tag{282}$$

where Equation (281) follows by the concavity of $\varphi$ and Equation (282) follows by (H1). Now, for $i \geq K_0$, we can use the KŁ inequality and yield

$$a\|\theta^{i+1} - \theta^i\|^2 \leq \frac{\varphi(\mathrm{RegLoss}_\gamma(\theta^i)) - \varphi(\mathrm{RegLoss}_\gamma(\theta^{i+1}))}{\varphi'(\mathrm{RegLoss}_\gamma(\theta^i))} \tag{283}$$

$$\leq \left(\varphi(\mathrm{RegLoss}_\gamma(\theta^i)) - \varphi(\mathrm{RegLoss}_\gamma(\theta^{i+1}))\right) \|v^i\| \tag{284}$$

$$\leq \left(\varphi(\mathrm{RegLoss}_\gamma(\theta^i)) - \varphi(\mathrm{RegLoss}_\gamma(\theta^{i+1}))\right) b\|\theta^i - \theta^{i-1}\|, \tag{285}$$

where the last inequality follows by (H2). Hence, we have

$$\frac{\|\theta^{i+1} - \theta^i\|^2}{\|\theta^i - \theta^{i-1}\|} \leq \frac{b}{a} \left(\varphi(\mathrm{RegLoss}_\gamma(\theta^i)) - \varphi(\mathrm{RegLoss}_\gamma(\theta^{i+1}))\right) \tag{286}$$

$$= C_1 \left(\left(\mathrm{RegLoss}_\gamma(\theta^i)\right)^{1-\xi} - \left(\mathrm{RegLoss}_\gamma(\theta^{i+1})\right)^{1-\xi}\right). \tag{287}$$

Now, let $r \in (0,1)$ be arbitrary. If $\|\theta^{i+1} - \theta^i\| \geq r\|\theta^i - \theta^{i-1}\|$, the above inequality yields

$$\|\theta^{i+1} - \theta^i\| \leq \frac{C_1}{r} \left(\left(\mathrm{RegLoss}_\gamma(\theta^i)\right)^{1-\xi} - \left(\mathrm{RegLoss}_\gamma(\theta^{i+1})\right)^{1-\xi}\right). \tag{288}$$

Hence, we have that

$$\|\theta^{i+1} - \theta^i\| \leq r\|\theta^i - \theta^{i-1}\| + \frac{C_1}{r} \left(\left(\mathrm{RegLoss}_\gamma(\theta^i)\right)^{1-\xi} - \left(\mathrm{RegLoss}_\gamma(\theta^{i+1})\right)^{1-\xi}\right). \tag{289}$$

By summing up the above inequality from $K_0$ to some $K \geq K_0$, we yield

$$\sum_{i=K_0}^{K} \|\theta^{i+1} - \theta^i\| \tag{290}$$

$$\leq r \sum_{i=K_0}^{K} \|\theta^i - \theta^{i-1}\| + \frac{C_1}{r} \sum_{i=K_0}^{K} \left(\left(\mathrm{RegLoss}_\gamma(\theta^i)\right)^{1-\xi} - \left(\mathrm{RegLoss}_\gamma(\theta^{i+1})\right)^{1-\xi}\right) \tag{291}$$

$$= r \sum_{i=K_0-1}^{K-1} \|\theta^{i+1} - \theta^i\| + \frac{C_1}{r} \left(\left(\mathrm{RegLoss}_\gamma(\theta^{K_0})\right)^{1-\xi} - \left(\mathrm{RegLoss}_\gamma(\theta^{K+1})\right)^{1-\xi}\right) \tag{292}$$

$$= r \sum_{i=K_0}^{K} \|\theta^{i+1} - \theta^i\| + \frac{C_1}{r} \left(\left(\mathrm{RegLoss}_\gamma(\theta^{K_0})\right)^{1-\xi} - \left(\mathrm{RegLoss}_\gamma(\theta^{K+1})\right)^{1-\xi}\right) \tag{293}$$

$$+ r\|\theta^{K_0} - \theta^{K_0-1}\| - r\|\theta^{K+1} - \theta^K\| \tag{294}$$

$$\leq r \sum_{i=K_0}^{K} \|\theta^{i+1} - \theta^i\| + \frac{C_1}{r} \left(\left(\mathrm{RegLoss}_\gamma(\theta^{K_0})\right)^{1-\xi} - \left(\mathrm{RegLoss}_\gamma(\theta^{K+1})\right)^{1-\xi}\right) \tag{295}$$

$$+ r\|\theta^{K_0} - \theta^{K_0-1}\|. \tag{296}$$

Hence, by rearranging terms, we arrive at

$$\sum_{i=K_0}^{K} \|\theta^{i+1} - \theta^i\| \leq \frac{r}{1-r}\|\theta^{K_0} - \theta^{K_0-1}\| \tag{297}$$

$$+ \frac{C_1}{r(1-r)} \left(\left(\mathrm{RegLoss}_\gamma(\theta^{K_0})\right)^{1-\xi} - \left(\mathrm{RegLoss}_\gamma(\theta^{K+1})\right)^{1-\xi}\right). \tag{298}$$

For $K \to \infty$, this yields Equation (279).

To prove our claim, we again make use of the KŁ inequality. From Corollary 16 in Bolte et al. (2007b) and the resulting definition of $\varphi$, we know that

$$\frac{\text{RegLoss}_\gamma(\theta^k)^\xi}{\|v^k\|} = \frac{\left(\text{RegLoss}_\gamma(\theta^k) - \text{RegLoss}_\gamma(\theta^*)\right)^\xi}{\|v^k\|} \leq (1 - \xi)C_{\text{KL}} \tag{299}$$

for $k$ large enough and $\|v^k\| \neq 0$.

Now, we are ready to prove each of the three cases.

Case 1: $\xi = 0$

Let $I = \{k \in \mathbb{N} : \theta^k \neq \theta^{k-1}\}$ and let $k \in I$ be large enough. We assume w.l.o.g. that $\|v^k\| \neq 0$. Otherwise, we would have that $v^k = 0$ for all $k \in I$ and, by (H2), for all $k \in \mathbb{N} \setminus I$, and, thus, the algorithm would have been already initialized with a critical point. Now, from Equation (299), we have $\|v^k\| \geq C_{\text{KL}}^{-1} > 0$. From (H2), we get $b^2\|\theta^k - \theta^{k-1}\|^2 \geq \|v^k\|^2 \geq C_{\text{KL}}^{-2}$, and, hence,

$$\|\theta^k - \theta^{k-1}\|^2 \geq \frac{C_{\text{KL}}^{-2}}{b^2} > 0. \tag{300}$$

By (H1), this yields

$$\text{RegLoss}_\gamma(\theta^k) \leq \text{RegLoss}_\gamma(\theta^{k-1}) - a\|\theta^k - \theta^{k-1}\|^2 \leq \text{RegLoss}_\gamma(\theta^{k-1}) - \frac{aC_{\text{KL}}^{-2}}{b^2}, \tag{301}$$

and, hence,

$$\text{RegLoss}_\gamma(\theta^k) - \text{RegLoss}_\gamma(\theta^{k-1}) \leq -\frac{aC_{\text{KL}}^{-2}}{b^2} < 0. \tag{302}$$

Since the left-hand-side of the above inequality converges to zero, this implies that $I$ is finite and, hence, proves the first claim.

Case 2: $\xi \in (0, \frac{1}{2}]$

Let $k \geq K_0$. We assume w.l.o.g. that $S_k > 0$ for all $k \geq K_0$ and $\|v^k\| \neq 0$. Now, Equation (279) yields

$$S_k \leq \frac{r}{1-r}(S_{k-1} - S_k) + \frac{C_1}{r(1-r)}\text{RegLoss}_\gamma(\theta^k)^{1-\xi}. \tag{303}$$

From Equation (299), we have

$$\left(\frac{\text{RegLoss}_\gamma(\theta^k)^\xi}{\|v^k\|}\right)^{\frac{1-\xi}{\xi}} \leq ((1-\xi)C_{\text{KL}})^{\frac{1-\xi}{\xi}} \tag{304}$$

$$\Leftrightarrow \quad \text{RegLoss}_\gamma(\theta^k)^{1-\xi} \leq ((1-\xi)C_{\text{KL}})^{\frac{1-\xi}{\xi}}\|v^k\|^{\frac{1-\xi}{\xi}} \tag{305}$$

$$\leq ((1-\xi)C_{\text{KL}})^{\frac{1-\xi}{\xi}} b^{\frac{1-\xi}{\xi}}\|\theta^k - \theta^{k-1}\|^{\frac{1-\xi}{\xi}} \tag{306}$$

$$= C_2\|\theta^k - \theta^{k-1}\|^{\frac{1-\xi}{\xi}} \tag{307}$$

$$\Leftrightarrow \quad \text{RegLoss}_\gamma(\theta^k)^{1-\xi} \leq C_2(S_{k-1} - S_k)^{\frac{1-\xi}{\xi}}, \tag{308}$$

and, hence,

$$S_k \leq \frac{r}{1-r}(S_{k-1} - S_k) + \frac{C_1 C_2}{r(1-r)}(S_{k-1} - S_k)^{\frac{1-\xi}{\xi}}. \tag{309}$$

Let $K_1 \geq K_0$ be such that $S_{k-1} - S_k < 1$ for $k \geq K_1$. Since $\xi \in (0, \frac{1}{2}]$, we have that $\frac{1-\xi}{\xi} \geq 1$, and, thus, Equation (309) yields $S_k \leq C_3 (S_{k-1} - S_k)$ for a constant $C_3 > 0$. With $q = \frac{C_3}{1+C_3}$, this yields

$$\|\theta^k - \theta^*\| \leq S_k \leq q S_{k-1} \leq q^{k-K_1} S_{K_1} =: a_k, \tag{310}$$

for $k \geq K_1$. The sequence $a_k$ converges Q-linearly to zero with rate $0 < q < 1$, i.e.,

$$\lim_{k \to \infty} \frac{a_{k+1}}{a_k} = q. \tag{311}$$

Thus, by definition, the sequence $(\theta^k)_{k \in \mathbb{N}}$ converges R-linearly (Nocedal & Wright, 2006).

Case 3: $\xi \in (\frac{1}{2}, 1)$

Analogously to case 2, we yield Equation (309). Since $\xi \in (\frac{1}{2}, 1)$ and $S_k \to 0$, we have that $\frac{1-\xi}{\xi} < 1$ and that there exists $K_2 \geq K_0$ and a constant $C_4 > 0$ such that

$$S_k^{\frac{\xi}{1-\xi}} \leq C_4 (S_{k-1} - S_k) \tag{312}$$

for all $k \geq K_2$. By proceeding analogously to Attouch & Bolte (2009) (see derivations after Equation (13) therein), there exists a constant $C_5 > 0$ such that

$$S_k \leq C_5 k^{-\frac{1-\xi}{2\xi-1}} \tag{313}$$

for all $k \geq K_2$. This yields

$$\|\theta^k - \theta^*\| \leq S_k \leq C_5 k^{-\frac{1-\xi}{2\xi-1}} =: a_k. \tag{314}$$

The sequence $a_k$ converges Q-sublinearly, i.e.,

$$\lim_{k \to \infty} \frac{a_{k+1}}{a_k} = \lim_{k \to \infty} \frac{k^{\frac{1-\xi}{2\xi-1}}}{(k+1)^{\frac{1-\xi}{2\xi-1}}} = 1, \tag{315}$$

and, hence, the claim follows. $\qquad \square$

### D.14 Proof of Proposition 14

*Proof.* Let $a^k = \text{RegLoss}_\gamma(\theta^k)$ and $a^* = \text{RegLoss}_\gamma(\theta^*)$. By Equation (299) from Proposition 13, we know that

$$\frac{(a^k - a^*)^\xi}{\|v^k\|} \leq (1-\xi) C_{\text{KL}} \tag{316}$$

for $k$ large enough and $\|v^k\| \neq 0$. Hence, we yield

$$a\|\theta^{k+1} - \theta^k\|^2 \geq ab^{-2}\|v^{k+1}\|^2 \geq ab^{-2}\frac{(a^{k+1} - a^*)^{2\xi}}{((1-\xi)C_{\text{KL}})^2} = C(a^{k+1} - a^*)^{2\xi}, \tag{317}$$

for large $k$, where the first inequality is due to (H2). By (H1), we then have

$$a^k \geq a^{k+1} + a\|\theta^{k+1} - \theta^k\|^2 \geq a^{k+1} + C(a^{k+1} - a^*)^{2\xi}, \tag{318}$$

and, thus,

$$a^k - a^* \geq a^{k+1} - a^* + C(a^{k+1} - a^*)^{2\xi} \tag{319}$$

for large enough $k$. Note that we assume $\|v^k\| \neq 0$ since, otherwise, the algorithm would have converged in a finite number of iterations.

We start with assuming that $\xi > \frac{1}{2}$ and yield

$$\lim_{k\to\infty} \frac{|a^{k+1} - a^*|}{|a^k - a^*|} = \lim_{k\to\infty} \frac{a^{k+1} - a^*}{a^k - a^*} \leq \lim_{k\to\infty} \frac{a^{k+1} - a^*}{a^{k+1} - a^* + C\left(a^{k+1} - a^*\right)^{2\xi}} \tag{320}$$

$$= \lim_{k\to\infty} \frac{1}{1 + C\left(a^{k+1} - a^*\right)^{2\xi-1}} = 1, \tag{321}$$

and, hence, sub-linear convergence. Furthermore, for $\xi = \frac{1}{2}$, the above limit is bounded by $1/(1+C)$, which results in linear convergence.

Now, we assume that $\xi \in (\frac{1}{2(q+1)}, \frac{1}{2q}]$ for $q \in \mathbb{N}$ and proceed as follows. First,

$$\left(a^k - a^*\right)^q \geq \left(a^{k+1} - a^* + C\left(a^{k+1} - a^*\right)^{2\xi}\right)^q \tag{322}$$

holds for $k$ large enough. Second, we yield

$$\lim_{k\to\infty} \frac{|a^{k+1} - a^*|}{|a^k - a^*|^q} = \lim_{k\to\infty} \frac{a^{k+1} - a^*}{\left(a^k - a^*\right)^q} \leq \lim_{k\to\infty} \frac{a^{k+1} - a^*}{\left(a^{k+1} - a^* + C\left(a^{k+1} - a^*\right)^{2\xi}\right)^q} \tag{323}$$

$$= \lim_{k\to\infty} \frac{a^{k+1} - a^*}{\sum_{l=0}^{q} \binom{q}{l}\left(a^{k+1} - a^*\right)^{q-l} C^l \left(a^{k+1} - a^*\right)^{2\xi l}} \tag{324}$$

$$= \lim_{k\to\infty} \frac{1}{\sum_{l=0}^{q} \binom{q}{l} C^l \left(a^{k+1} - a^*\right)^{q+(2\xi-1)l-1}} \tag{325}$$

$$\leq \lim_{k\to\infty} \frac{1}{C^q \left(a^{k+1} - a^*\right)^{q+(2\xi-1)q-1}}. \tag{326}$$

Now, for $\xi < \frac{1}{2q}$, it follows that $q + (2\xi - 1)q - 1 < 0$ and, hence, that the above limit is zero. For $\xi = \frac{1}{2q}$, we have that the above limit is bounded by $1/C^q$, which proves the claim. Note that the special case $\xi = 1/2$ has been analyzed separately above. $\qquad\square$

### D.15  Proof of Proposition 15

*Proof.* Let $\xi \leq \frac{1}{2q}$ for $q \in \mathbb{N}_{\geq 2}$. Furthermore, let $U^*$ be a neighborhood of $\theta^*$ in which $\mathrm{RegLoss}_\gamma$ is strictly convex. Now, let $r \in (0, 1)$ be such that $\overline{B}_r(\theta^*) \subseteq U^*$, i.e., the closed ball with radius $r$ around $\theta^*$ lies inside $U^*$. We define the function $\mathrm{RegLoss}_\gamma^{\mathrm{ext}}$ via

$$\mathrm{RegLoss}_\gamma^{\mathrm{ext}}(\theta) = \begin{cases} \mathrm{RegLoss}_\gamma(\theta) - \mathrm{RegLoss}_\gamma(\theta^*), & \text{if } \theta \in \overline{B}_r(\theta^*), \\ \infty, & \text{else.} \end{cases} \tag{327}$$

Note that $\mathrm{RegLoss}_\gamma^{\mathrm{ext}}$ is a proper, convex, lower-semicontinuous function with $\min \mathrm{RegLoss}_\gamma^{\mathrm{ext}} = 0$ and $\{\theta^*\} = \arg\min \mathrm{RegLoss}_\gamma^{\mathrm{ext}}$. From Theorem 5 in Bolte et al. (2017), we thus get

$$\|\theta - \theta^*\| \leq C_{\mathrm{KL}} |\mathrm{RegLoss}_\gamma^{\mathrm{ext}}(\theta)|^{1-\xi} \tag{328}$$

for $\theta \in B_\rho(\theta^*)$ with $r \geq \rho > 0$ sufficiently small. Furthermore, from Theorem 3.1.8 in Nesterov (2003), we know that there exists a constant $L$ such that $\mathrm{RegLoss}_\gamma$ is $L$-Lipschitz in $\overline{B}_{\frac{r}{2}}(\theta^*)$, and from Proposition 14, we know that there exists a constant $C_q > 0$ such that $|\mathrm{RegLoss}_\gamma(\theta^{k+1}) - \mathrm{RegLoss}_\gamma(\theta^*)| \leq C_q |\mathrm{RegLoss}_\gamma(\theta^k) - \mathrm{RegLoss}_\gamma(\theta^*)|^q$, for large enough $k$.

Now, for $k$ large enough, we yield

$$\|\theta^{k+1} - \theta^*\| \leq C_{\mathrm{KL}}|\mathrm{RegLoss}_\gamma(\theta^{k+1}) - \mathrm{RegLoss}_\gamma(\theta^*)|^{1-\xi} \tag{329}$$

$$\leq C_{\mathrm{KL}}C_q^{1-\xi}|\mathrm{RegLoss}_\gamma(\theta^k) - \mathrm{RegLoss}_\gamma(\theta^*)|^{q(1-\xi)} \tag{330}$$

$$\leq C_{\mathrm{KL}}C_q^{1-\xi}L^{q(1-\xi)}\|\theta^k - \theta^*\|^{q(1-\xi)} \tag{331}$$

$$\leq C_{\mathrm{KL}}C_q^{1-\xi}L^{q(1-\xi)}\|\theta^k - \theta^*\|^{q-\frac{1}{2}}, \tag{332}$$

and, thus, $\|\theta^{k+1} - \theta^*\| \leq C\|\theta^k - \theta^*\|^{q-\frac{1}{2}}$ for $C = C_{\mathrm{KL}}C_q^{1-\xi}L^{q(1-\xi)}$. $\qquad\square$

### D.16  Proof of Proposition 16

Assume that $\theta^* = (\alpha^*, W^*, b^*)$ fulfills

$$\langle w_i^*, x_j \rangle + b_i^* \neq 0 \quad \forall i \in \{1, \ldots, N\} \text{ and } \forall j \in \{1, \ldots, m\}. \tag{333}$$

If (333) holds true, there exists a neighborhood $U^*$ of $\theta^*$ such that $\langle w_i, x_j \rangle + b_i \neq 0$ for all $i \in \{1, \ldots, N\}$, $j \in \{1, \ldots, m\}$, and $\theta \in U^*$. That is, $\mathrm{RegLoss}_\gamma$ is a twice continuously differentiable function in $U^*$. Furthermore, by assumption, $\nabla^2\mathrm{RegLoss}_\gamma(\theta^*)$ is invertible. From Proposition 1 in Huang et al. (2019), it then follows that $\mathrm{RegLoss}_\gamma$ fulfills the KŁ property at $\theta^*$ with $\xi = 1/2$, i.e., there exists a $C > 0$ and $r > 0$ such that

$$|\mathrm{RegLoss}_\gamma(\theta) - \mathrm{RegLoss}_\gamma(\theta^*)|^{\frac{1}{2}} \leq C\|\nabla\mathrm{RegLoss}_\gamma(\theta)\|, \tag{334}$$

for all $\theta \in B_r(\theta^*) \subseteq U^*$.

# E   Discussion of our Findings in Context of the KŁ Literature

Proposition 13 can be seen as a standard result in the KŁ literature (see, e.g., Attouch & Bolte, 2009). However, Proposition 14 and Proposition 15 follow from stronger assumptions on the underlying objective function. The main difference is that the standard assumptions on the objective function $f$ in the KŁ literature (see, e.g., Attouch et al., 2013) are usually the following:

- The function $f$ is assumed to be proper but is allowed to take infinite values, i.e., $f : \mathbb{R}^n \to \mathbb{R} \cup \{\infty\}$. This allows to incorporate convex constraints in the objective function via a characteristic function. In our case, $\mathrm{RegLoss}_\gamma$ is always finite.

- The function $f$ is assumed to be lower-semicontinuous. This allows for a much larger class of optimization problems, but renders the analysis of the convergence in value obsolete, as $x^k \to x^*$ does not imply $f(x^k) \to f(x^*)$. In our case, $\mathrm{RegLoss}_\gamma$ is continuous.

As such, our results in Proposition 14 and Proposition 15 come from the fact that $f$ is finite and continuous in our case. In fact, we can extend the results in Attouch et al. (2013) under these additional assumptions as follows.

**Proposition 18** (Extension 1 of Theorem 2.9 in Attouch et al. (2013))**.** *Let all assumptions of Theorem 2.9 in Attouch et al. (2013) hold. Furthermore, let $f : \mathbb{R}^n \to \mathbb{R}$ be (finite), continuous, and a KŁ function with $\varphi$ as given in Proposition 8. Let $\xi$ be the KŁ exponent associated with $\bar{x}$. Then, the following holds true:*

- *If $\xi \in (\frac{1}{2(q+1)}, \frac{1}{2q}]$, $f(x^k)$ converges to $f(\bar{x})$ with order $q \in \mathbb{N}$.*
- *If $\xi > \frac{1}{2}$, $f(x^k)$ converges Q-sublinearly to $f(\bar{x})$.*

*Furthermore, if $\xi \in (\frac{1}{2(q+1)}, \frac{1}{2q})$, we even observe super-Q-convergence.*

Note that the proof of Proposition 14 merely uses the continuity of $f$, the KŁ property, (H2), and (H1). Hence, the proof of the above proposition follows the exact same structure. Furthermore, the following holds true:

**Proposition 19** (Extension 2 of Theorem 2.9 in Attouch et al. (2013))**.** *Under the assumptions of Proposition 18, let $f$ admit a neighborhood $\bar{U}$ of $\bar{x}$ in which $f$ is strictly convex. Then, the following holds true: If $\xi \leq \frac{1}{2q}$ for $q \in \mathbb{N}_{\geq 2}$, the sequence $(x^k)_{k \in \mathbb{N}}$ converges with order at least $q - \frac{1}{2}$.*

Again, Proposition 15 merely uses the continuity of $f$ and the finiteness of $f$ inside the ball $\overline{B}_r(\bar{x})$, thus the proof of the above proposition follows the one of Proposition 15.

We illustrate these results based on an example. We consider the proximal algorithm given by

$$x^{k+1} \in \arg\min \left\{ f(y) + \frac{1}{2\lambda}\|y - x\|^2 : y \in \mathbb{R}^n \right\}, \tag{335}$$

where $\lambda$ is a positive parameter that can vary for each $k$ but remains bounded, i.e., $\lambda \in [\underline{\lambda}, \overline{\lambda}] \subseteq (0, \infty)$. We demonstrate that, under the above assumptions on the objective function $f$, the proximal algorithm achieves a much faster convergence than the one derived in Attouch & Bolte (2009). However, at this point, we want to emphasize that their analysis holds under much weaker assumptions, which allows to consider more general optimization problems.

Attouch et al. (2013) derived $(H1)$ and $(H2)$ for the proximal algorithm (see Equations (33)–(35) therein) given that $f$ is a proper, lower-semicontinuous function that is bounded from below. Furthermore, if the function $f$ is continuous, $(H3)$ directly follows. Now, let $q \in \mathbb{N}_{\geq 2}$ and $f$ be defined as follows $f(x) = \|x\|^{\frac{2q}{2q-1}}$. Then, $f$ fulfills the KŁ property at the global minimizer $\bar{x} = 0$ with $\xi = 1/2q$. To show this, we define

$\varphi(s) = s^{1-\frac{1}{2q}} = s^{\frac{2q-1}{2q}}$ and yield $\varphi'(s) = \frac{2q-1}{2q}s^{-\frac{1}{2q}}$. Now,

$$\varphi'(f(x)) = \frac{2q-1}{2q}\|x\|^{-\frac{1}{2q-1}}, \tag{336}$$

$$\|\nabla f(x)\| = \frac{2q}{2q-1}\|x\|^{\frac{1}{2q-1}} \tag{337}$$

for all $x \neq 0$. That is, $\varphi'(f(x))\|\nabla f(x)\| = 1$ for all $x \neq 0$. Furthermore, $f$ fulfills all assumptions of Proposition 18 and Proposition 19. That is, we expect the iterates of the proximal algorithm in this setting to converge with order of at least $q-\frac{1}{2}$. In the following, we will prove analytically that the iterates converge even faster with order $2(q-\frac{1}{2})$. In addition, we demonstrate our results numerically. Note that the analysis in Attouch & Bolte (2009) guarantees only R-linear convergence in this setting.

To compute the next iterate $x^{k+1}$ given $x^k$, we consider the function $g(x) = f(x) + \frac{1}{2\lambda}\|x - x^k\|^2$. As $g$ is convex and differentiable for $x \neq 0$ we have that $\nabla g(x^{k+1}) = 0$ or $x^{k+1} = 0$. The latter results in finite convergence. Hence, we assume $x^{k+1} \neq 0$ in our analysis. The gradient of $g$ is given by

$$\nabla g(x) = \frac{2q}{2q-1}\|x\|^{\frac{2(1-q)}{2q-1}}x + \frac{1}{\lambda}(x - x^k), \tag{338}$$

and, hence, the following equality holds

$$x^{k+1} = x^k - \frac{2q\lambda}{2q-1}\|x^{k+1}\|^{\frac{2(1-q)}{2q-1}}x^{k+1}, \tag{339}$$

which gives $\left(1 + \frac{2q\lambda}{2q-1}\|x^{k+1}\|^{\frac{2(1-q)}{2q-1}}\right)\|x^{k+1}\| = \|x^k\|$. Thus,

$$\frac{\|x^{k+1}\|}{\|x^k\|^{2q-1}} = \frac{\|x^{k+1}\|}{\left(1 + \frac{2q\lambda}{2q-1}\|x^{k+1}\|^{\frac{2(1-q)}{2q-1}}\right)^{2q-1}\|x^{k+1}\|^{2q-1}} \tag{340}$$

$$= \frac{\|x^{k+1}\|}{\left(\sum_{l=0}^{2q-1}\binom{2q-1}{l}\left(\frac{2q\lambda}{2q-1}\right)^l\|x^{k+1}\|^{\frac{2(1-q)l}{2q-1}}\right)\|x^{k+1}\|^{2q-1}} \tag{341}$$

$$= \frac{1}{\left(\sum_{l=0}^{2q-1}\binom{2q-1}{l}\left(\frac{2q\lambda}{2q-1}\right)^l\|x^{k+1}\|^{\frac{4q^2-6q+2l(1-q)+2}{2q-1}}\right)} \tag{342}$$

$$\leq \frac{1}{\left(\frac{2q\lambda}{2q-1}\right)^{2q-1}\|x^{k+1}\|^0} \tag{343}$$

$$= \frac{1}{\left(\frac{2q\lambda}{2q-1}\right)^{2q-1}}. \tag{344}$$

To confirm our results numerically, we set $\lambda = 0.1$ and vary $q \in \{2, 3, 4, 5, 6\}$. We use the above proximal algorithm to solve $\min_{x\in\mathbb{R}^{10}} f(x)$ with 100 random starting points for each $q$. Furthermore, we estimate the convergence order via

$$q \approx \frac{\log\left(\frac{\|x^{k+1}-x^k\|}{\|x^k-x^{k-1}\|}\right)}{\log\left(\frac{\|x^k-x^{k-1}\|}{\|x^{k-1}-x^{k-2}\|}\right)}, \tag{345}$$

for large $k$. The results are reported in Table 3.

Evidently, the theoretical convergence orders are also observed in the numerical experiments. Our example shows that faster, i.e., super-linear, convergence orders can be achieved under additional assumptions on the objective function and, thereby, links our analysis to the general KŁ literature.

Table 3: Estimated convergence order (mean and std.) across different values for $q \in \{2, 3, 4, 5, 6\}$.

| $q = 2$ | | $q = 3$ | | $q = 4$ | | $q = 5$ | | $q = 6$ | |
|---|---|---|---|---|---|---|---|---|---|
| Mean | (Std.) | Mean | (Std.) | Mean | (Std.) | Mean | (Std.) | Mean | (Std.) |
| 2.9931 | (0.0325) | 4.9834 | (0.0702) | 6.9582 | (0.1486) | 8.9711 | (0.1479) | 10.9398 | (0.2334) |

## F  Numerical Analysis

### F.1  Implementation of DCON

**General implementation details.** For our experiments, DCON is implemented as a Python package using C++ code to accelerate computations. It is built using cmake. For building the Python interface, we use pybind11.[6] For solving the quadratic programs, DCON requires Gurobi.[7] To accelerate linear algebra operations, our package uses the Intel Math Kernel Library.[8] We use a Python class called `DCON` through which we can easily access the DCON algorithm via a `.fit` routine. This class also implements a function `get_keras` to return the trained model as a keras model. An implementation of DCON using only Python code can be downloaded from GitHub[9].

**DC subproblem.** In our implementation, the DC subproblem is approached as follows. DCA is stopped if either the norm of the difference of two successive iterates is smaller than $10^{-12}$ or a maximum number of DCA iterations is reached. The latter can be passed as a parameter.

**Alpha subproblem.** The solution of the alpha subproblem given in Section 4.2 is computed via a singular value decomposition (SVD). For this, we make use of the Eigen library[10], particularly the divide-and-conquer SVD algorithm `bdcsvd`.

**Convergence criterion.** A parameter `n_epochs` is used to set the number of outer iterations $\mathcal{M}$. If an integer is passed to `n_epochs` in the `.fit` routine, DCON stops after the specified number of iterations. If `n_epochs` is set to "`auto`", DCON stops when the distance between two successive iterations is smaller than $10^{-6}$.

### F.2  Datasets

We searched the UCI machine learning repository using a systematic procedure. For this, we set the filter options as follows:

- Default Task: Regression

- Attribute Type: Numerical

- Data Type: Multivariate

- Instances: 100 to 1000

Afterward, we filtered for datasets where *Regression* is the unique task in the column *Default Task*. Altogether, this led to ten datasets. In addition, we filtered datasets that have at least 100 training instances after the train-validation-test split, yielding nine benchmark datasets listed in Table 4. We note that the range of instances was chosen to strike a balance between computational feasibility and rigorous evaluation. Datasets with fewer than 100 training instances might not provide enough data to meaningfully train and test a neural network, while those with more than 1000 instances could introduce prohibitive computational demands for 30 train-test splits as performed in this work. In summary, these criteria were chosen to provide a fair and rigorous evaluation of the model performance.

---

[6]`https://github.com/pybind/pybind11`, last accessed 02/12/21.
[7]`https://www.gurobi.com/products/gurobi-optimizer/`, last accessed 02/12/21.
[8]`https://software.intel.com/content/www/us/en/develop/tools/math-kernel-library.html`, last accessed 02/12/21.
[9]`https://github.com/DanielTschernutter/DCON`
[10]`http://eigen.tuxfamily.org/`, last accessed 02/12/21.

Table 4: Datasets

| Dataset | Description | Num. of covariates n |
|---|---|---|
| DS1 | Computer Hardware Data Set[1] | 9 |
| DS2 | Forest Fires Data Set[2] | 13 |
| DS3 | Stock Portfolio Performance Data Set[3] | 12 |
| DS4 | Yacht Hydrodynamics Data Set[4] | 7 |
| DS5 | Facebook Metrics Data Set[5] | 19 |
| DS6 | Residential Building Data Set[6] | 105 |
| DS7 | Real Estate Valuation Data Set[7] | 7 |
| DS8 | QSAR Fish Toxicity Data Set[8] | 7 |
| DS9 | QSAR Aquatic Toxicity Data Set[9] | 9 |

[1] https://archive.ics.uci.edu/ml/datasets/Computer+Hardware, last accessed 03/20/20.

[2] https://archive.ics.uci.edu/ml/datasets/Forest+Fires, last accessed 03/20/20.

[3] https://archive.ics.uci.edu/ml/datasets/Stock+portfolio+performance, last accessed 03/20/20.

[4] https://archive.ics.uci.edu/ml/datasets/Yacht+Hydrodynamics, last accessed 03/20/20.

[5] https://archive.ics.uci.edu/ml/datasets/Facebook+metrics, last accessed 03/20/20.

[6] https://archive.ics.uci.edu/ml/datasets/Residential+Building+Data+Set, last accessed 03/20/20.

[7] https://archive.ics.uci.edu/ml/datasets/Real+estate+valuation+data+set, last accessed 03/20/20.

[8] https://archive.ics.uci.edu/ml/datasets/QSAR+fish+toxicity, last accessed 03/20/20.

[9] https://archive.ics.uci.edu/ml/datasets/QSAR+aquatic+toxicity, last accessed 03/20/20.

### F.3 Preprocessing

Each of the datasets are preprocessed using standard approaches, while taking into account dataset-dependent restrictions and recommendations. The following describes the steps taken to preprocess the datasets.

**Dataset DS1.** We drop the columns `VENDOR`, `MODEL`, and `ERP`. Furthermore, we use RobustScaler and MinMaxScaler for features and target variable.[11] We split the data into 80 % for training, 10 % for validation, and 10 % for testing. This split is repeated 30 times to obtain 30 different splits of the data.

**Dataset DS2.** We encode `month` and `day` into numbers 1–12 and 1–7, respectively. As recommended, we log-transform `area`. Finally, we scale the features and the target variable with the RobustScaler and the MinMaxScaler, respectively. We split the data into 80 % for training, 10 % for validation, and 10 % for testing. This split is repeated 30 times to obtain 30 different splits of the data.

**Dataset DS3.** We concatenate the sheets `1st period` to `4th period` and keep the columns `Large B/P`, `Large ROE`, `Large S/P`, `Large Return Rate in the last quarter`, `Large Market Value`, and `Small systematic Risk` as training features. The column `Annual Return.1` represents our target variable. Finally, we scale the features and the target variable with the RobustScaler and the MinMaxScaler, respectively. We split the data into 80 % for training, 10 % for validation, and 10 % for testing. This split is repeated 30 times to obtain 30 different splits of the data.

**Dataset DS4.** We drop the column `prismatic_coefficient` due to missing values. Finally, we scale the features and the target variable with the RobustScaler and the MinMaxScaler, respectively. We split the data into 80 % for training, 10 % for validation, and 10 % for testing. This split is repeated 30 times to obtain 30 different splits of the data.

**Dataset DS5.** We drop the columns `comment`, `like`, and `share`. We use one-hot encoding for the column `Type` and drop all samples with missing values. Finally, we scale the features and the target variable with the RobustScaler and the MinMaxScaler, respectively. We split the data into 80 % for training, 10 % for validation, and 10 % for testing. This split is repeated 30 times to obtain 30 different splits of the data.

---

[11] We use the scalers of sklearn.preprocessing

**Dataset DS6.** We drop the columns START YEAR, START QUARTER, COMPLETION YEAR, and COMPLETION QUARTER. We use a time lag of 4 as this was found to be effective in earlier research (Rafiei & Adeli, 2018). We use CONSTRUCTION COSTS as the target variable. Finally, we scale the features and the target variable with the RobustScaler and the MinMaxScaler, respectively. We split the data into 80 % for training, 10 % for validation, and 10 % for testing. This split is repeated 30 times to obtain 30 different splits of the data.

**Dataset DS7.** We drop the column No. We scale the features and the target variable with the RobustScaler and the MinMaxScaler, respectively. We split the data into 80 % for training, 10 % for validation, and 10 % for testing. This split is repeated 30 times to obtain 30 different splits of the data.

**Dataset DS8.** We scale the features and the target variable with the RobustScaler and the MinMaxScaler, respectively. We split the data into 80 % for training, 10 % for validation, and 10 % for testing. This split is repeated 30 times to obtain 30 different splits of the data.

**Dataset DS9.** We scale the features and the target variable with the RobustScaler and the MinMaxScaler, respectively. We split the data into 80 % for training, 10 % for validation, and 10 % for testing. This split is repeated 30 times to obtain 30 different splits of the data.

### F.4 Hyperparameter

All hyperparameters and their tuning ranges are reported in Table 5. Of note, DCON has no hyperparameter related to training, only one related to the neural network architecture. In contrast to that, Adam has hyperparameters related to both the neural network architecture *and* the training process.

Table 5: Hyperparameter tuning ranges.

| Hyperparameters | Tuning range |
|---|---|
| **Adam (Hyperparameters related to training)** | |
| Learning rate | $\{10^{-3}, 5 \cdot 10^{-3}, 5 \cdot 10^{-4}\}$ |
| First moment exponential decay rate $\beta_1$ | $\{0.9, 0.99\}$ |
| Batch size | $\{64, 128, m\}$ |
| **Adam (Hyperparameters related to neural network structure)** | |
| Regularization parameter | $\{10^{-2}, 10^{-3}\}$ |
| **DCON (Hyperparameters related to neural network structure)** | |
| Regularization parameter $\gamma$ | $\{10^{-2}, 10^{-3}, 10^{-4}, 10^{-5}\}$ |

Note: The patience for early stopping was set to 10 epochs for Adam.

### F.5 Discussion of Generalization to Unseen Data

For the following analysis, we follow Mohri et al. (2018) and denote with $\mathcal{X} \subseteq \mathbb{R}^n$ the input space and with the measurable set $\mathcal{Y} \subseteq \mathbb{R}$ the target space. Furthermore, let $\mathcal{D}$ be a distribution over $\mathcal{X} \times \mathcal{Y}$ and the training set $\mathcal{S} = ((x_1, y_1), (x_2, y_2), \dots, (x_m, y_m))$ be i.i.d. samples drawn from $\mathcal{D}$. The class of single hidden layer neural networks is denoted by $\mathcal{H}$ and the elements depending on the actual parameters $\theta$ by $h_\theta$. Then, from Theorem 11.3 in Mohri et al. (2018), it follows that, for $\delta > 0$ and $h_\theta \in \mathcal{H}$,

$$\mathbb{E}_{(x,y) \sim \mathcal{D}} \left( (h_\theta(x) - y)^2 \right) \leq \frac{1}{m} \sum_{j=1}^{m} (h_\theta(x_i) - y_i)^2 + 4M\hat{\mathcal{R}}_{\mathcal{S}}(\mathcal{H}) + 3M^2 \sqrt{\frac{\log(\frac{2}{\delta})}{2m}} \qquad (346)$$

holds true with probability $1 - \delta$, where $M$ is such that $|h_\theta(x) - y| \leq M$ for all $(x, y) \in \mathcal{X} \times \mathcal{Y}$ and $h_\theta \in \mathcal{H}$, and $\hat{\mathcal{R}}_{\mathcal{S}}(\mathcal{H})$ denotes the empirical Rademacher complexity of the class $\mathcal{H}$. A broad stream of literature provides bounds for the empirical Rademacher complexity of neural networks. For instance, Theorem 2 in Golowich et al. (2018) gives a bound in $\mathcal{O}(\frac{1}{\sqrt{m}})$ under suitable norm constraints on the neural network parameters and $\mathcal{X} = \{x \in \mathbb{R}^n : \|x\| \leq B\}$. That is, given $\delta > 0$, a bounded input and target space, and defining

$\mathcal{H}_r = \{h_\theta \in \mathcal{H} : \|\theta\| \leq r\}$ for $r > 0$, one yields a generalization bound of the form

$$\mathbb{E}_{(x,y)\sim\mathcal{D}}\left((h_\theta(x) - y)^2\right) \leq \frac{1}{m}\sum_{j=1}^{m}(h_\theta(x_i) - y_i)^2 + \mathcal{O}(\frac{1}{\sqrt{m}}), \tag{347}$$

which holds true with probability $1 - \delta$ for all $h_\theta \in \mathcal{H}_r$.

Thus, one reason that DCON achieves a better generalization to unseen data compared to Adam in our numerical experiments (Section 6) may be attributed to the superior training performance of DCON.

### F.6 Convergence Plots

In the following, we show the convergence plots for each combination of dataset and layer size for the experiments in Section 6.3. The plots are in Appendices F.6 to F.6. At this point, we note again that $\epsilon_g$ is determined by solving $\min_{\epsilon^{g_l}\in\mathcal{C}}\sum_{t=1}^{n}(v_{l,t}(\theta_l^{\mathrm{DC}}, \epsilon^{g_l}, \mathbb{1}))^2 + (v_l(\theta_l^{\mathrm{DC}}, \epsilon^{g_l}, \mathbb{1}))^2$ after each DC subproblem for each inner iteration. Here, the set $\mathcal{C}$ decodes the constraints in (21). As mentioned in the main paper, the rationale is to find the values for $\epsilon^{g_l}$ that set the corresponding entries of $v$ to zero for $\epsilon^{h_l} = \mathbb{1}$, which exist due to Proposition 7. Sometimes, numerical issues lead to poor estimates of the correct values of $\epsilon_g$. The main problem arises in identifying the correct $j \in \{1, \ldots, m\}$ for which $\epsilon_j^{g_l} \in [0, 1]$ and building the corresponding objective function. In our implementation, we decided to vary the corresponding $\epsilon_j^{g_l} \in [0, 1]$ if $|\langle w_l, x_j \rangle + b_l| < \delta$ with $\delta = 10^{-6}$. Afterward, the objective function is built by setting

$$v_{l,t}(\theta, \epsilon^{g_l}) = \sum_{j=1}^{m}\beta_j^{g_l}H^\delta\left(\langle w_l, x_j \rangle + b_l\right)x_{j,t}\epsilon_j^{g_l} + \sum_{j=1}^{m}2\frac{\alpha_l^2}{m}H\left(\langle w_l, x_j \rangle + b_l\right)x_{j,t}\left(\langle w_l, x_j \rangle + b_l\right) \tag{348}$$

$$+ \sum_{j=1}^{m}2\frac{\gamma}{m}x_{j,t}\left(\langle w_l, x_j \rangle + b_l\right) - \sum_{j=1}^{m}\beta_j^{h_l}H\left(\langle w_l, x_j \rangle + b_l\right)x_{j,t}, \tag{349}$$

$$v_l(\theta, \epsilon^{g_l}) = \sum_{j=1}^{m}\beta_j^{g_l}H^\delta\left(\langle w_l, x_j \rangle + b_l\right)\epsilon_j^{g_l} + \sum_{j=1}^{m}2\frac{\alpha_l^2}{m}H\left(\langle w_l, x_j \rangle + b_l\right)\left(\langle w_l, x_j \rangle + b_l\right) \tag{350}$$

$$+ \sum_{j=1}^{m}2\frac{\gamma}{m}\left(\langle w_l, x_j \rangle + b_l\right) - \sum_{j=1}^{m}\beta_j^{h_l}H\left(\langle w_l, x_j \rangle + b_l\right), \tag{351}$$

where $H^\delta(x) = 1$ if $x > -\delta$ and zero else.

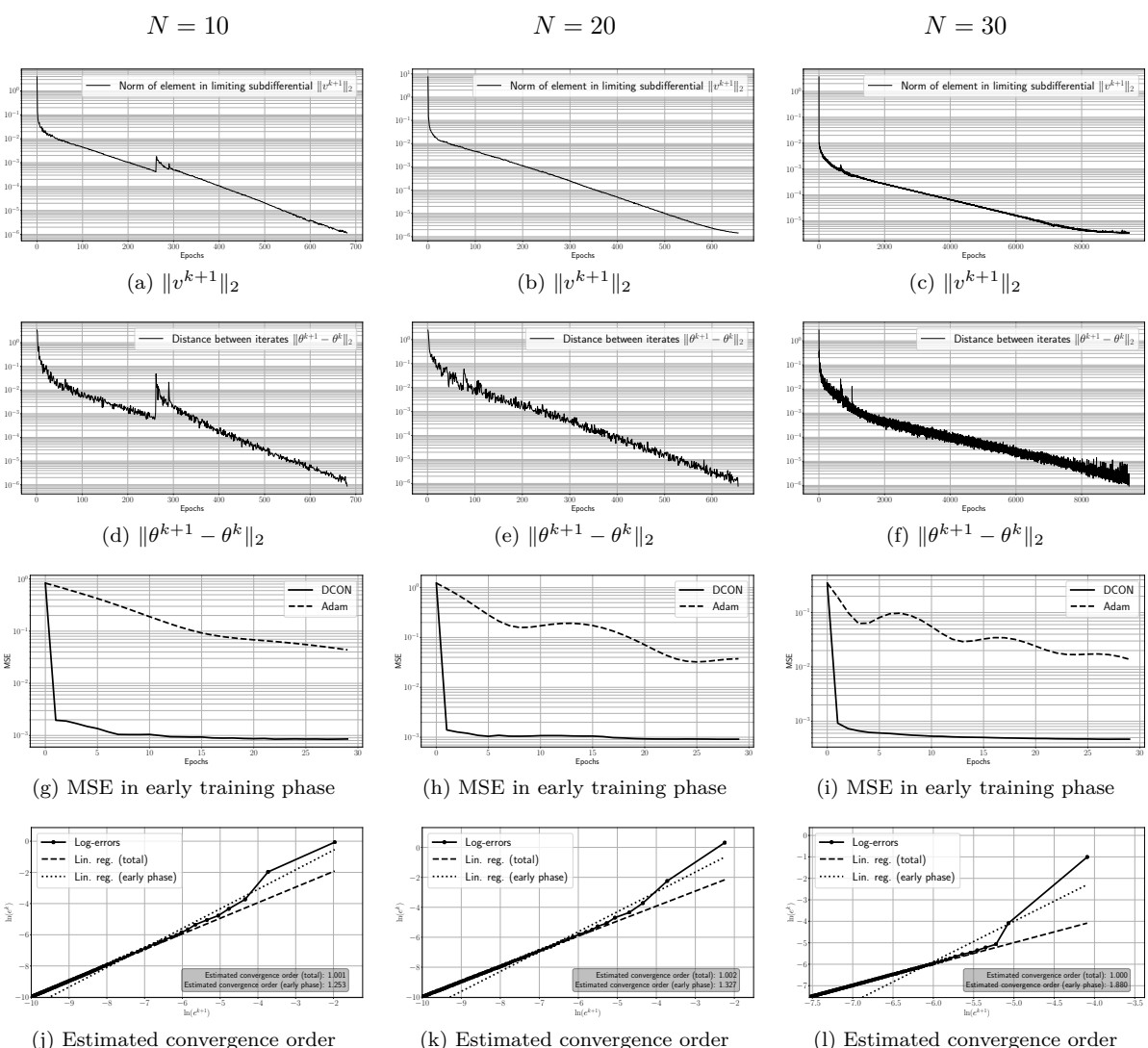

Figure 4: Convergence plots for dataset DS1.

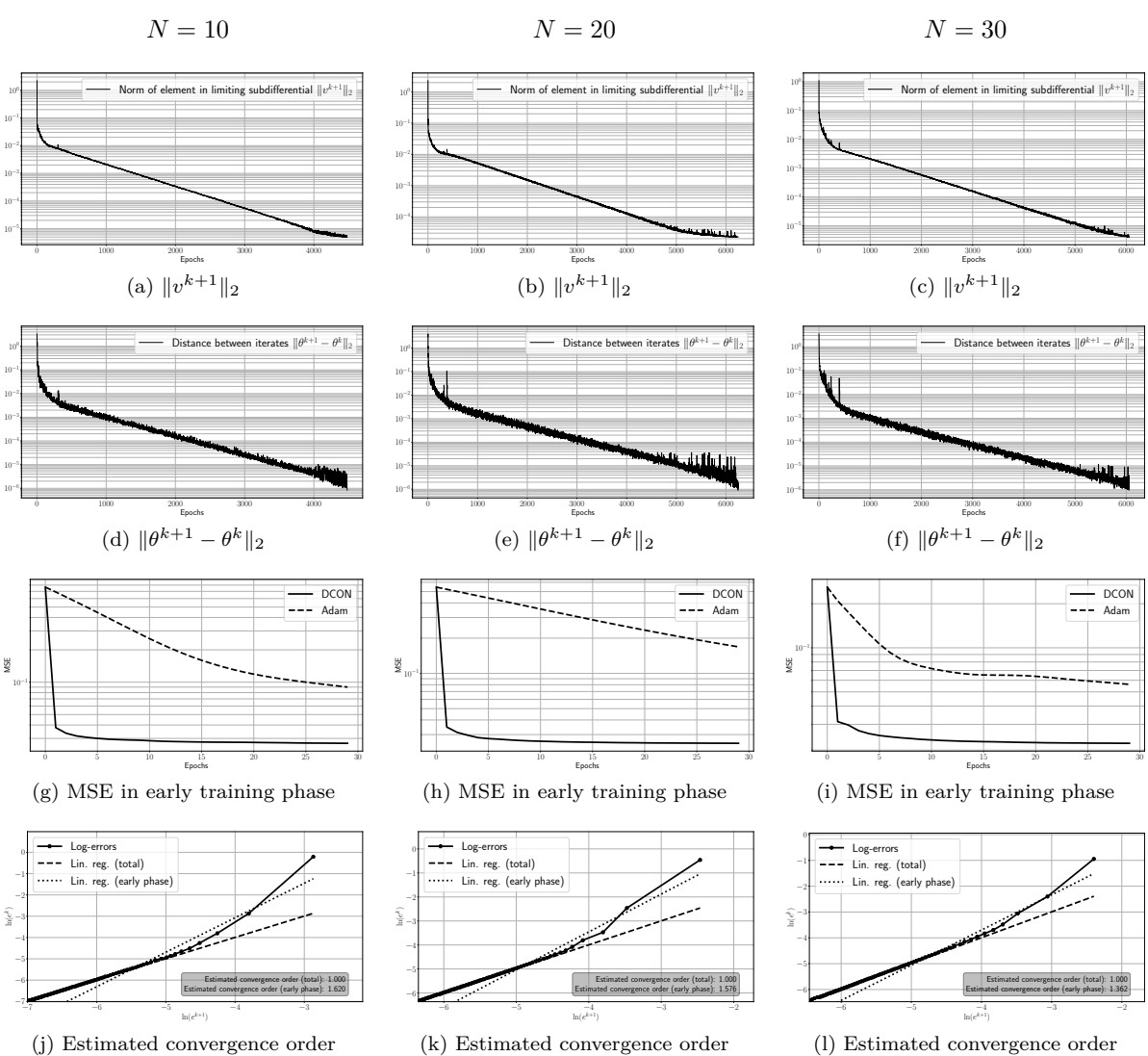

Figure 5: Convergence plots for dataset DS2.

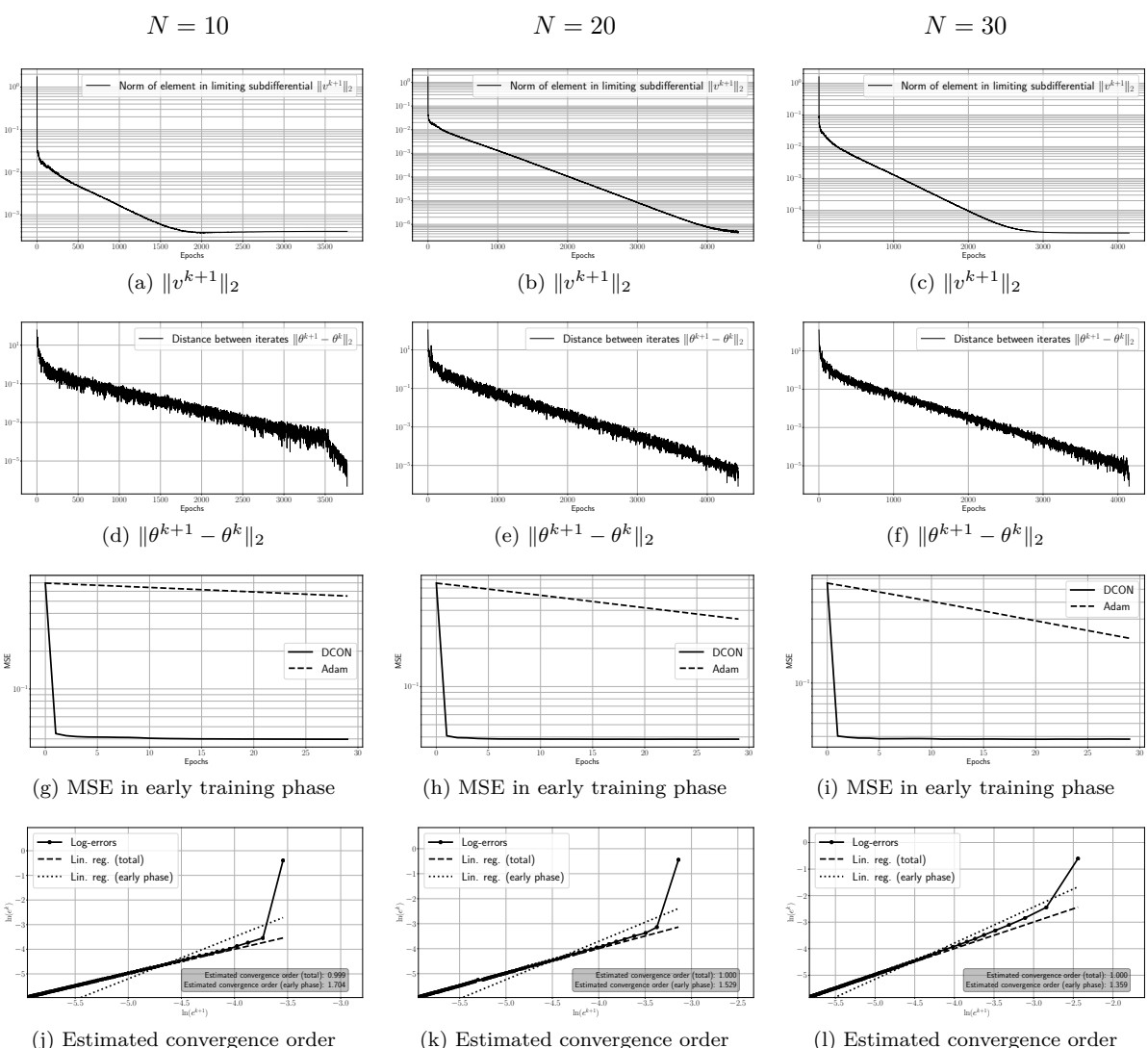

Figure 6: Convergence plots for dataset DS3.

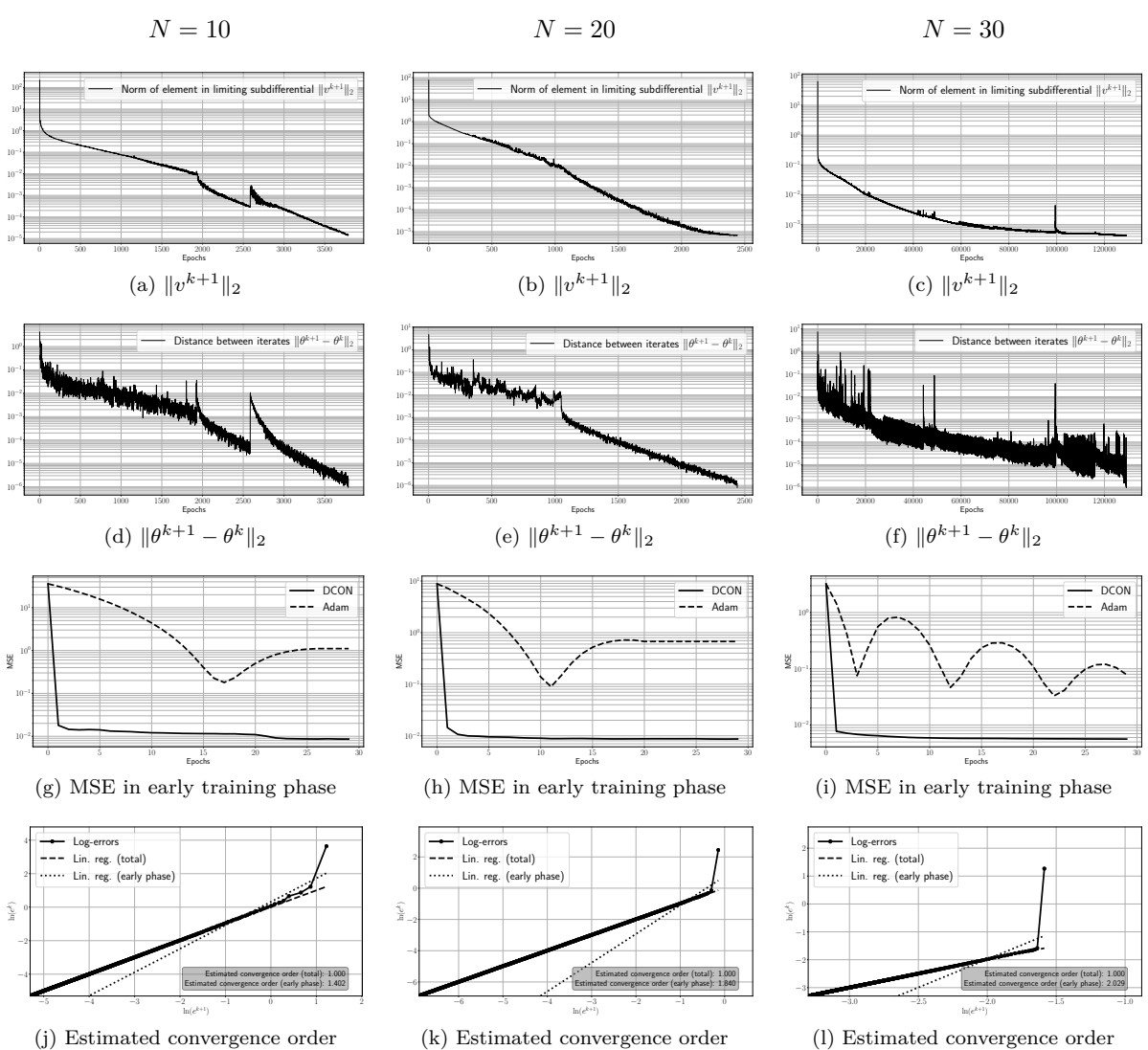

Figure 7: Convergence plots for dataset DS4.

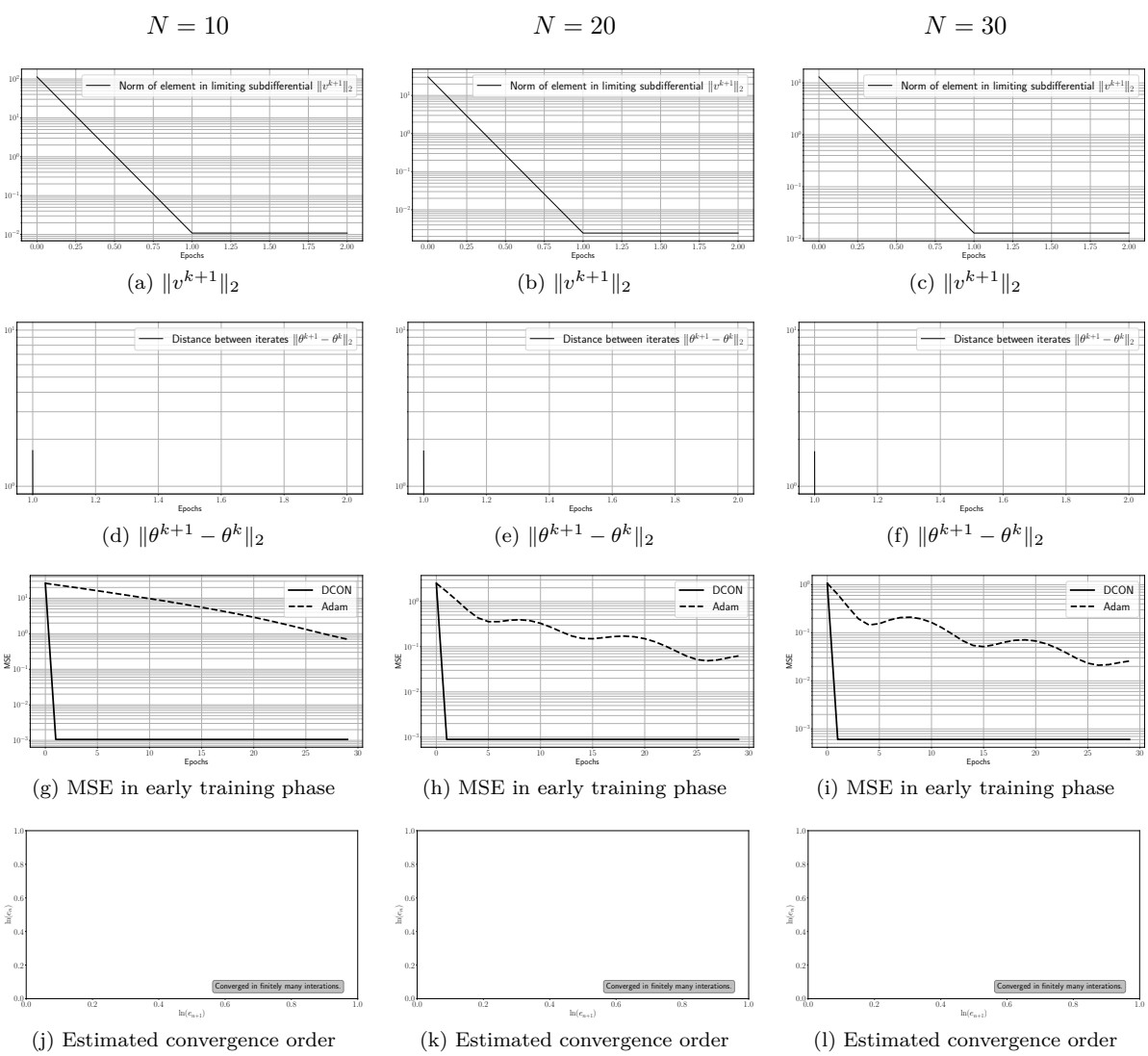

Figure 8: Convergence plots for dataset DS5.

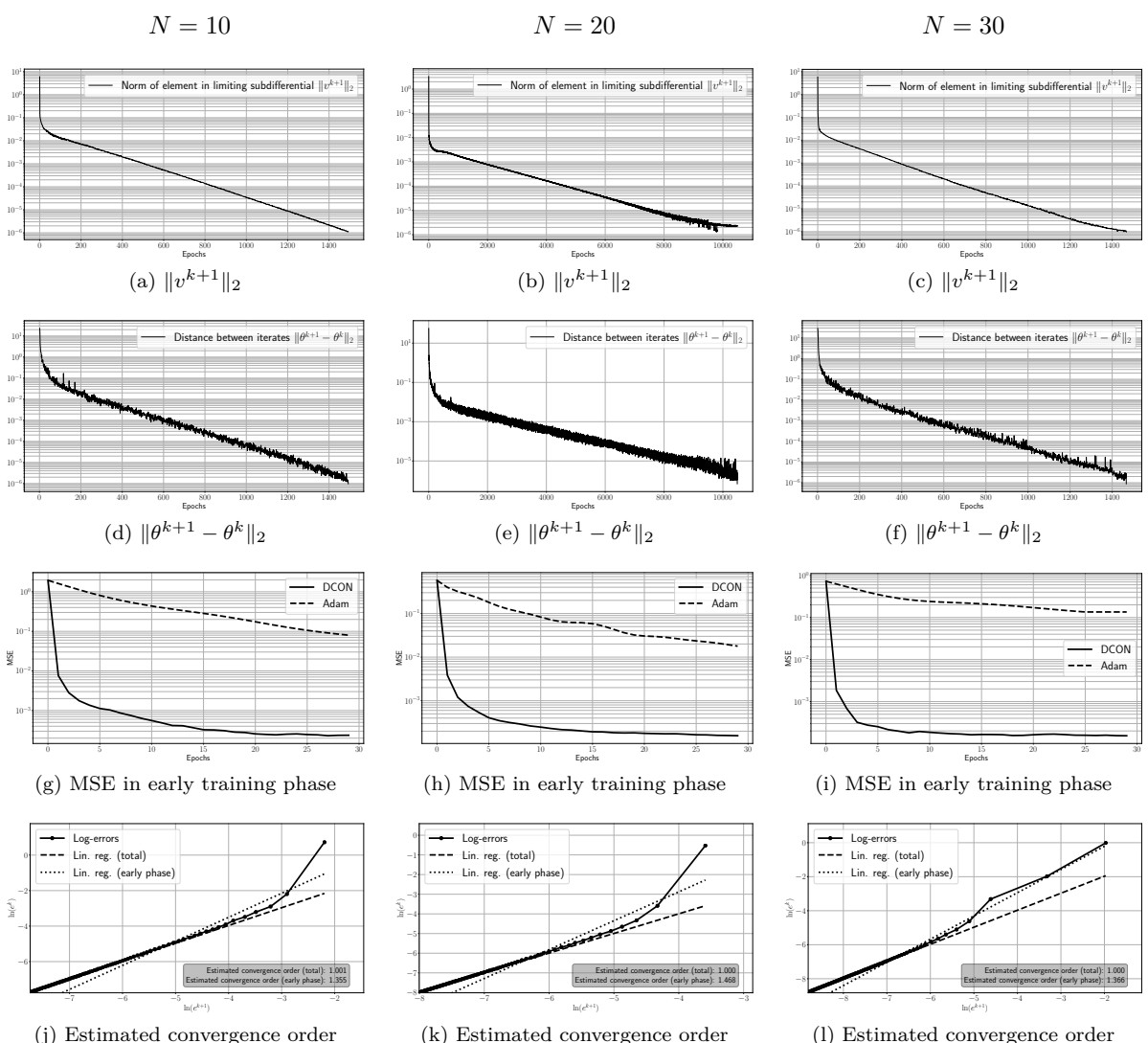

Figure 9: Convergence plots for dataset DS6.

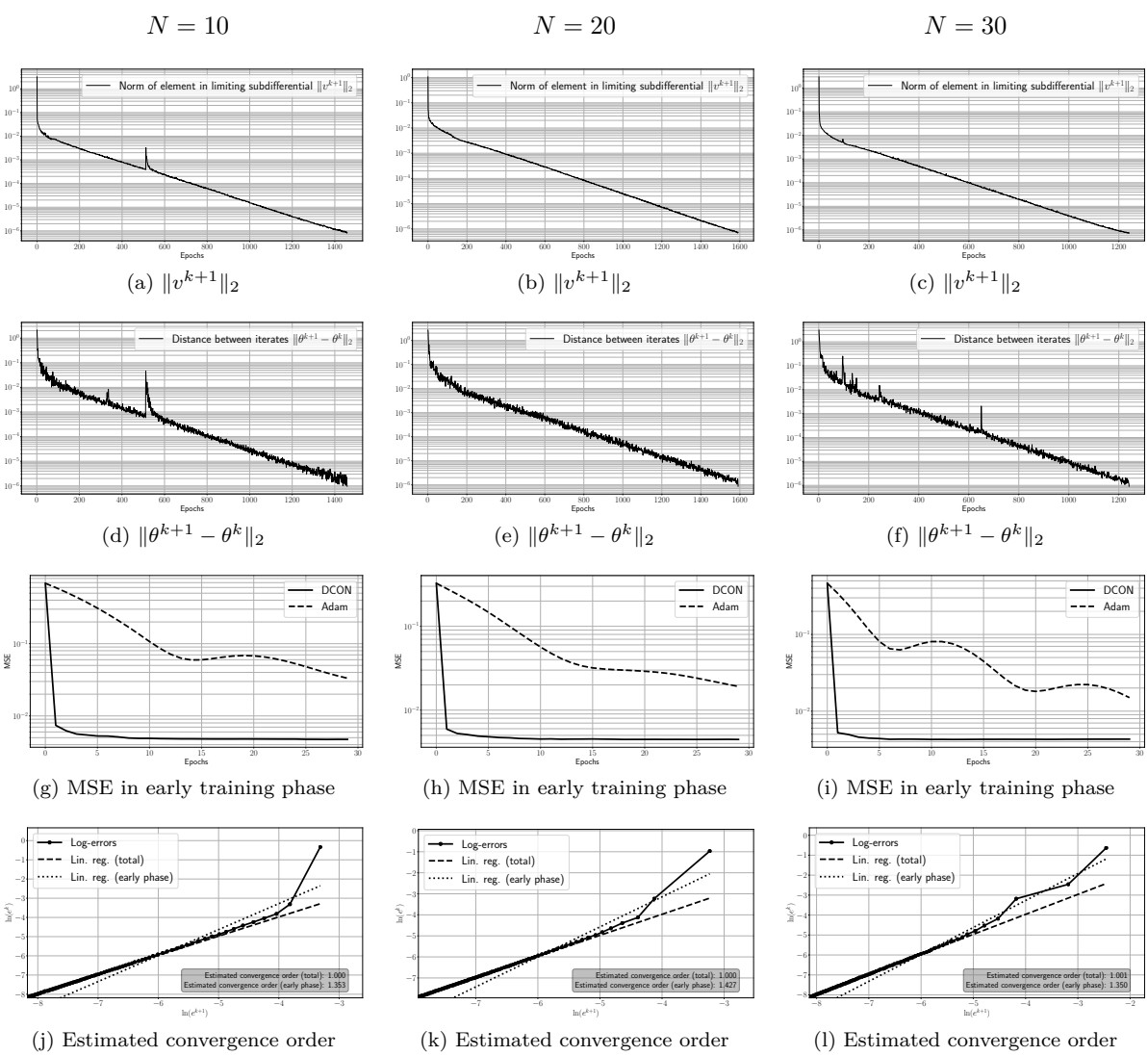

Figure 10: Convergence plots for dataset DS7.

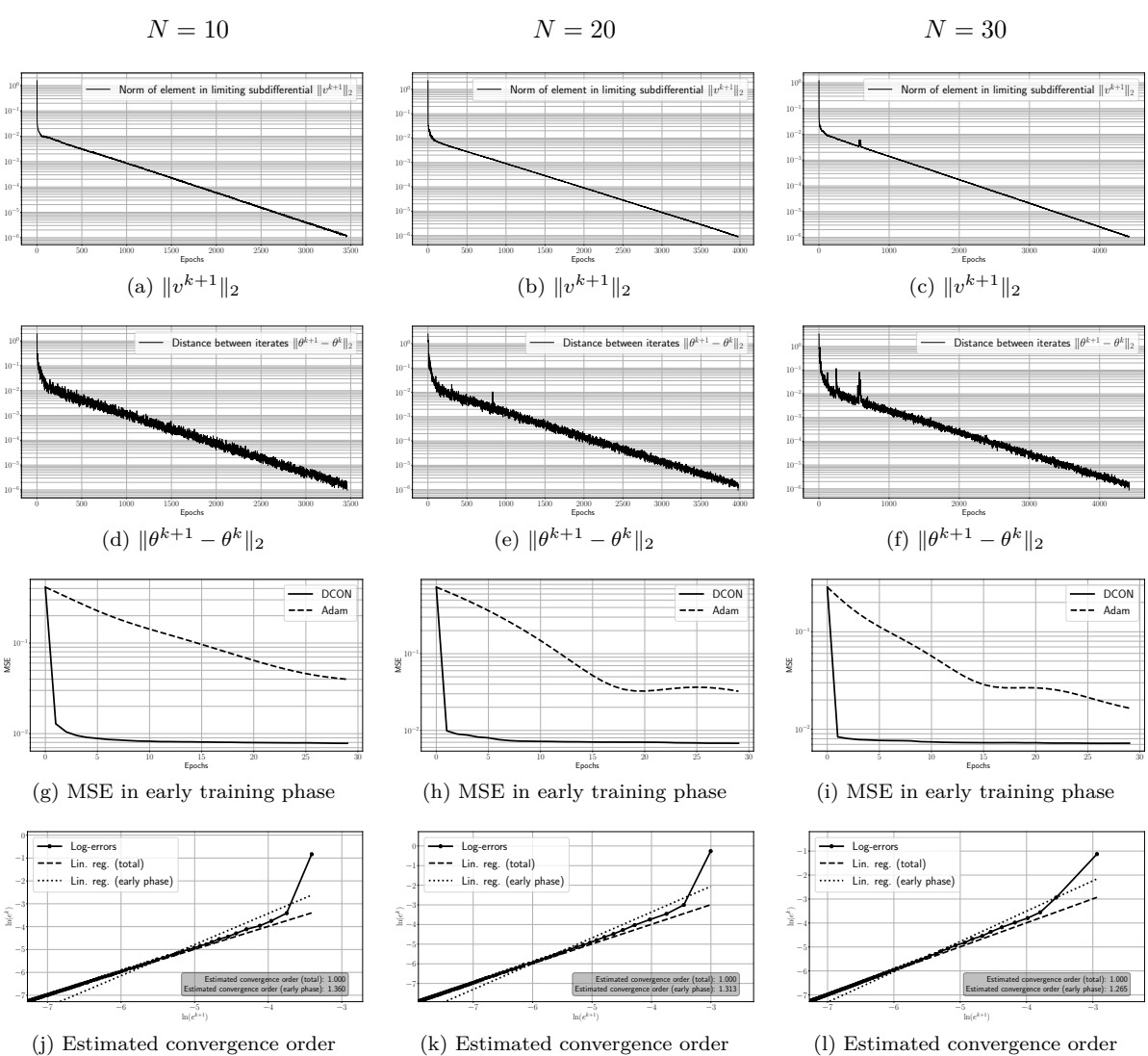

Figure 11: Convergence plots for dataset DS8.

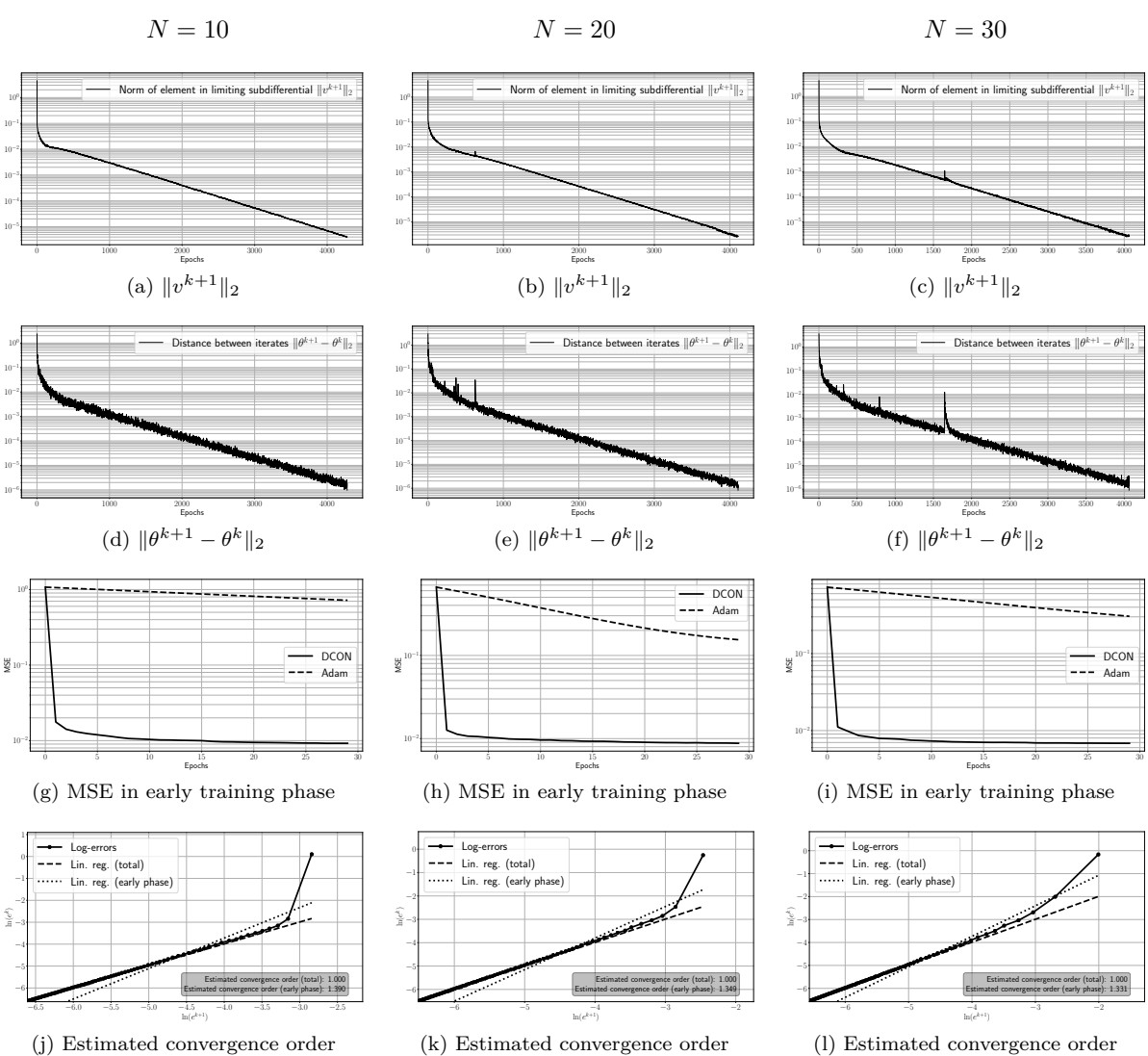

Figure 12: Convergence plots for dataset DS9.

Figures (a) to (c) show how the norm of the element in the limiting subdifferential approaches zero as the number of iterations increase for different hidden layer sizes. Figures (d) to (f) report the distance between the parameter vectors of two successive iterations, i. e., $\|\theta^{k+1} - \theta^k\|_2$, and should empirically analyze convergence. We find that the difference between two successive iterates decreases gradually in all experiments. Figures (g) to (i) report the MSE for training with Adam and DCON, while Figures (j) to (l) estimate the convergence rate of DCON. We observe linear convergence in all experiments except for dataset 5, where DCON terminates after finitely many iterations for all three hidden layer sizes.

## G  Experiments using the MNIST benchmark dataset

**Preprocessing.** First, the image data (color codes of each pixel) are scaled to lie within zero and one. Second, the images, originally of size $28 \times 28$, are down-sized using interpolation to images of size $10 \times 10$ using sklearn. The reason is to fulfill Assumption 1 for $M$. Otherwise, due to the large number of pixels that show white background, the matrix $M$ is singular. Afterward, we scale the data again with a MinMaxScaler. The target variables are encoded as explained in the main paper. We split the data into $70\,\%$ for training and $30\,\%$ for validation. For the experiment on the MNIST subset, we only use the first 10,000 samples of the training set. The test data are already provided in the MNIST benchmark dataset.

**Implementation of scalable version.** For the scalable version of DCON, we implemented the ADMM approach described in Appendix C. That is, we refrain from using Gurobi to solve (QP), and, instead, we use cuBLAS[12] and Thrust[13] to compute the basic linear algebra subroutines in Algorithms 2 and 3. In addition, we use an acceleration approach via over-relaxation; see Boley (2013) for details. We set the ADMM parameter $\rho = 1$, $\alpha_{\mathrm{relax}} = 1.4$ for over-relaxation, and the maximum number of ADMM iterations $\mathcal{L} = 500$. The choice was determined via trial and error for the MNIST benchmark dataset and then hard-coded. We stop Algorithm 2 when both the primal and dual residuals are smaller than $10^{-3}$; see Boley (2013) for details. The scalable version of DCON runs on a GPU using CUDA.

**Hardware.** We performed the MNIST experiment on a server with an Nvidia Tesla V100 with $32\,\mathrm{GB}$ of RAM. After the train-validation split, we yield $m = 42000$, which results in a memory requirement of $\sim 14$ GB for the matrix $V$ (in double-precision floating-point arithmetic). For comparison, medium-sized datasets as defined in the main paper require merely 800 MB. Furthermore, state-of-the-art training algorithms usually work in single-precision floating-point arithmetic, while, recently, even half precision floating-point arithmetic is used to speedup computations.

---

[12]https://docs.nvidia.com/cuda/cublas/index.html, last accessed 02/12/21.
[13]https://docs.nvidia.com/cuda/thrust/index.html, last accessed 02/12/21.

# H    Runtime Experiments

In the following, we compare DCON and Adam in terms of runtime. The experimental setup is as follows. We consider between 50 and 1000 training examples of the MNIST dataset and construct 5 to 50 random Fourier features following Rahimi & Recht (2007). Then, we use the same parameter settings as in the main paper for DCON and fix the number of hidden neurons to $N = 10$ and the regularization parameter to $\gamma = 0.01$. First, we let DCON train for 30 iterations and measure the runtime in seconds. Second, we train again with Adam for each hyperparameter combination in Table 5 and stop the training when it reaches the same mean squared error as DCON.

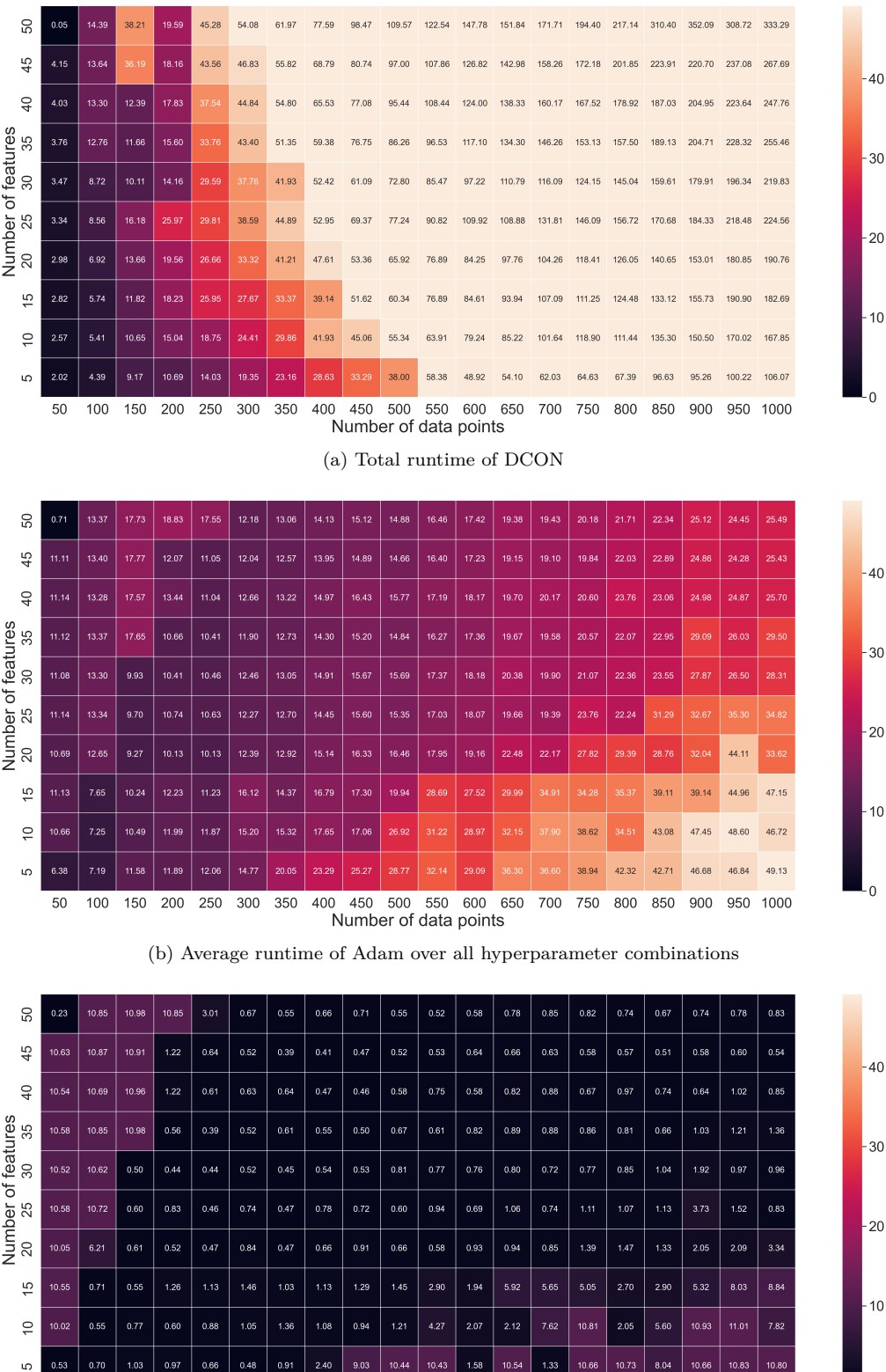

(a) Total runtime of DCON

(b) Average runtime of Adam over all hyperparameter combinations

(c) Best runtime of Adam over all hyperparameter combinations

Figure 13: Runtime for DCON and Adam.

Figure 13 shows the total runtime of DCON, the average runtime per hyperparameter combination of Adam, and the best runtime over all hyperparameter combinations of Adam. Evidently, Adam has difficulties reaching the same mean squared error as DCON in the "small number of features" and "large number of training examples" (lower right) region. Furthermore, DCON is consistently faster than Adam in the "small number of training examples" (left) region. At this point, we also want to emphasize that the numbers reported in Figure 13c strongly favor Adam, as they report the runtime of the fastest run among all hyperparameter combinations to reach the same mean squared error as DCON. These combinations are – of course – a priori unknown, and we thus merely report the runtime for transparency reasons.

DCON requires no hyperparameter optimization and the above comparisons are made with respect to the average or best runtime of Adam per hyperparameter combination. Thus, for a more realistic comparison, we would need to compare the runtime of DCON to the total runtime of Adam, i.e., the total runtime needed for all hyperparameter combinations in Table 5. Figure 14 shows the percentage improvements of DCON over Adam with respect to total runtime. We can see large improvements by a factor of up to 5.2 in the "small number of training examples" (left) region but still consistent improvements of around 2% in the "large number of training examples" and "large number of features" (upper right) region.

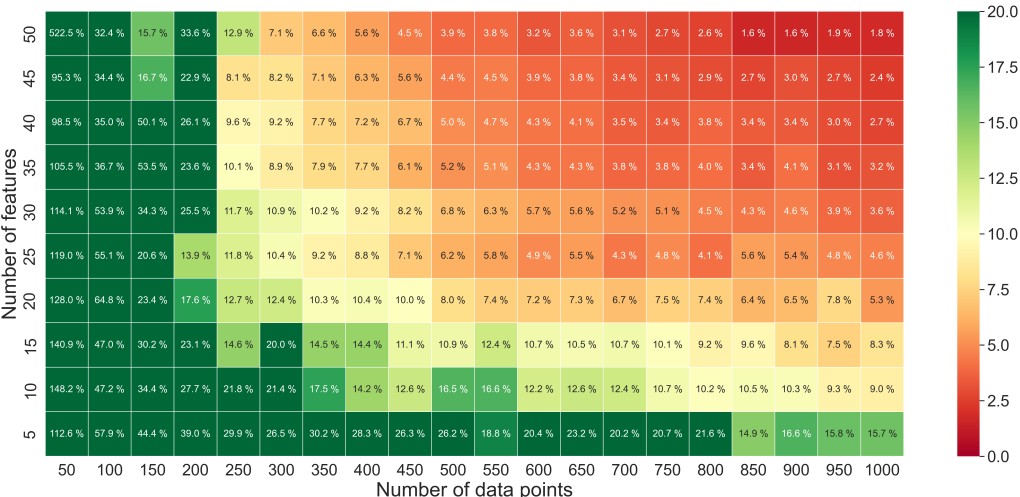

Figure 14: Percentage runtime improvements of DCON over Adam.

# I  Experiments with Additional Baselines

Here, we also compared DCON against other, non-neural baselines. In particular, we consider (i) linear regression, (ii) lasso, (iii) ridge regression, and (iv) kernel ridge regression. The hyperparameter grids can be found in Table 6.

Table 6: Hyperparameter tuning ranges for additional baselines.

| Hyperparameters | Tuning range |
|---|---|
| **Lasso** | |
| Regularization parameter | $\{10^{-5}, 10^{-4}, 10^{-3}, 10^{-2}, 10^{-1}, 1, 10\}$ |
| **Ridge** | |
| Regularization parameter | $\{10^{-5}, 10^{-4}, 10^{-3}, 10^{-2}, 10^{-1}, 1, 10\}$ |
| **Kernel ridge** | |
| Regularization parameter | $\{10^{-3}, 10^{-2}, 10^{-1}, 1, 10, 10^{2}, 10^{3}\}$ |
| Kernel | $\{\mathrm{linear}, \mathrm{poly}, \mathrm{rbf}\}$ |

Hyperparameters were tuned on the validation set.

Results are reported in Tables 7 to 10. Evidently, DCON achieves large performance improvements on average against linear models. Furthermore, we observe improvements in the training loss, on average, up to 31% for kernel ridge regression.

Table 7: Relative performance improvement in mean squared error of DCON over linear regression.

| | **Training** | | | | | | **Test** | | | | | |
|---|---|---|---|---|---|---|---|---|---|---|---|---|
| | $N = 10$ | | $N = 20$ | | $N = 30$ | | $N = 10$ | | $N = 20$ | | $N = 30$ | |
| | Mean | (Std.) | Mean | (Std.) | Mean | (Std.) | Mean | (Std.) | Mean | (Std.) | Mean | (Std.) |
| DS 1 | 3.11 | (1.09) | 2.71 | (0.68) | 2.95 | (0.79) | 0.47 | (0.75) | 0.51 | (0.60) | 0.45 | (0.75) |
| DS 2 | 0.34 | (0.06) | 0.54 | (0.08) | 0.67 | (0.09) | −0.13 | (0.22) | −0.24 | (0.16) | −0.20 | (0.15) |
| DS 3 | 0.18 | (0.06) | 0.23 | (0.06) | 0.26 | (0.06) | −0.07 | (0.14) | −0.10 | (0.13) | −0.13 | (0.13) |
| DS 4 | 3.88 | (2.84) | 3.73 | (1.67) | 5.09 | (1.53) | 2.72 | (2.01) | 2.87 | (1.85) | 3.27 | (1.82) |
| DS 5 | −0.94 | (0.02) | −0.89 | (0.03) | −0.84 | (0.04) | −0.85 | (0.11) | −0.84 | (0.09) | −0.81 | (0.10) |
| DS 6 | 1.73 | (0.58) | 6.53 | (2.03) | 3.40 | (0.86) | 0.02 | (0.47) | 0.03 | (0.52) | 0.14 | (0.55) |
| DS 7 | 0.51 | (0.14) | 0.67 | (0.13) | 0.72 | (0.13) | 0.27 | (0.14) | 0.35 | (0.15) | 0.39 | (0.16) |
| DS 8 | 0.26 | (0.06) | 0.36 | (0.05) | 0.42 | (0.05) | 0.07 | (0.08) | 0.09 | (0.10) | 0.09 | (0.09) |
| DS 9 | 0.48 | (0.12) | 0.73 | (0.11) | 0.86 | (0.11) | 0.06 | (0.13) | 0.04 | (0.10) | 0.06 | (0.18) |
| Average | 1.06 | (0.55) | 1.62 | (0.54) | 1.50 | (0.41) | 0.28 | (0.45) | 0.30 | (0.41) | 0.36 | (0.44) |

Results are based on 30 runs with different train-test splits. Reported is the mean performance improvement (e. g., 0.1 means 10 %) and the standard deviation (Std.) in parentheses.

Table 8: Relative performance improvement in mean squared error of DCON over Lasso.

| | Training | | | | | | Test | | | | | |
|---|---|---|---|---|---|---|---|---|---|---|---|---|
| | $N = 10$ | | $N = 20$ | | $N = 30$ | | $N = 10$ | | $N = 20$ | | $N = 30$ | |
| | Mean | (Std.) | Mean | (Std.) | Mean | (Std.) | Mean | (Std.) | Mean | (Std.) | Mean | (Std.) |
| DS 1 | 3.73 | (2.32) | 3.31 | (2.10) | 3.66 | (2.77) | 0.47 | (0.83) | 0.54 | (0.82) | 0.49 | (1.03) |
| DS 2 | 0.36 | (0.08) | 0.57 | (0.08) | 0.70 | (0.09) | −0.16 | (0.20) | −0.28 | (0.15) | −0.23 | (0.14) |
| DS 3 | 0.24 | (0.13) | 0.29 | (0.13) | 0.32 | (0.14) | −0.05 | (0.16) | −0.08 | (0.13) | −0.11 | (0.14) |
| DS 4 | 3.93 | (2.83) | 3.78 | (1.67) | 5.15 | (1.49) | 2.73 | (2.01) | 2.88 | (1.84) | 3.28 | (1.82) |
| DS 5 | −0.87 | (0.33) | −0.77 | (0.60) | −0.57 | (1.35) | −0.81 | (0.27) | −0.81 | (0.22) | −0.68 | (0.81) |
| DS 6 | 2.23 | (1.26) | 7.55 | (2.20) | 4.10 | (1.40) | 0.09 | (0.63) | 0.07 | (0.52) | 0.21 | (0.61) |
| DS 7 | 0.55 | (0.17) | 0.71 | (0.16) | 0.76 | (0.16) | 0.30 | (0.19) | 0.38 | (0.20) | 0.42 | (0.20) |
| DS 8 | 0.28 | (0.07) | 0.39 | (0.09) | 0.45 | (0.09) | 0.09 | (0.09) | 0.11 | (0.10) | 0.11 | (0.09) |
| DS 9 | 0.51 | (0.15) | 0.77 | (0.14) | 0.90 | (0.17) | 0.08 | (0.13) | 0.06 | (0.12) | 0.08 | (0.20) |
| Average | 1.22 | (0.82) | 1.84 | (0.80) | 1.72 | (0.85) | 0.30 | (0.50) | 0.32 | (0.46) | 0.40 | (0.56) |

Results are based on 30 runs with different train-test splits. Reported is the mean performance improvement (e.g., 0.1 means 10 %) and the standard deviation (Std.) in parentheses.

Table 9: Relative performance improvement in mean squared error of DCON over Ridge.

| | Training | | | | | | Test | | | | | |
|---|---|---|---|---|---|---|---|---|---|---|---|---|
| | $N = 10$ | | $N = 20$ | | $N = 30$ | | $N = 10$ | | $N = 20$ | | $N = 30$ | |
| | Mean | (Std.) | Mean | (Std.) | Mean | (Std.) | Mean | (Std.) | Mean | (Std.) | Mean | (Std.) |
| DS 1 | 3.13 | (1.10) | 2.73 | (0.69) | 2.97 | (0.80) | 0.43 | (0.76) | 0.48 | (0.62) | 0.42 | (0.77) |
| DS 2 | 0.34 | (0.06) | 0.54 | (0.08) | 0.67 | (0.09) | −0.17 | (0.17) | −0.27 | (0.16) | −0.23 | (0.12) |
| DS 3 | 0.18 | (0.05) | 0.24 | (0.06) | 0.26 | (0.06) | −0.08 | (0.14) | −0.11 | (0.12) | −0.14 | (0.13) |
| DS 4 | 3.91 | (2.83) | 3.76 | (1.67) | 5.13 | (1.51) | 2.72 | (2.01) | 2.87 | (1.85) | 3.26 | (1.82) |
| DS 5 | −0.93 | (0.03) | −0.88 | (0.04) | −0.82 | (0.09) | −0.85 | (0.11) | −0.84 | (0.09) | −0.81 | (0.11) |
| DS 6 | 1.82 | (0.57) | 6.78 | (2.00) | 3.55 | (0.83) | 0.02 | (0.47) | 0.04 | (0.54) | 0.15 | (0.58) |
| DS 7 | 0.51 | (0.14) | 0.67 | (0.13) | 0.72 | (0.13) | 0.27 | (0.15) | 0.35 | (0.15) | 0.39 | (0.16) |
| DS 8 | 0.26 | (0.06) | 0.36 | (0.05) | 0.42 | (0.05) | 0.07 | (0.08) | 0.09 | (0.10) | 0.09 | (0.09) |
| DS 9 | 0.48 | (0.12) | 0.74 | (0.11) | 0.87 | (0.11) | 0.07 | (0.13) | 0.05 | (0.10) | 0.07 | (0.18) |
| Average | 1.08 | (0.55) | 1.66 | (0.54) | 1.53 | (0.41) | 0.28 | (0.45) | 0.30 | (0.41) | 0.36 | (0.44) |

Results are based on 30 runs with different train-test splits. Reported is the mean performance improvement (e.g., 0.1 means 10 %) and the standard deviation (Std.) in parentheses.

Table 10: Relative performance improvement in mean squared error of DCON over Kernel Ridge.

| | Training | | | | | | Test | | | | | |
|---|---|---|---|---|---|---|---|---|---|---|---|---|
| | $N = 10$ | | $N = 20$ | | $N = 30$ | | $N = 10$ | | $N = 20$ | | $N = 30$ | |
| | Mean | (Std.) | Mean | (Std.) | Mean | (Std.) | Mean | (Std.) | Mean | (Std.) | Mean | (Std.) |
| DS 1 | 1.22 | (2.82) | 1.16 | (2.75) | 1.21 | (2.80) | 15.98 | (71.11) | 48.60 | (246.09) | 26.70 | (126.44) |
| DS 2 | 0.32 | (0.20) | 0.52 | (0.25) | 0.65 | (0.24) | −0.06 | (0.41) | −0.16 | (0.41) | −0.14 | (0.35) |
| DS 3 | 0.18 | (0.29) | 0.23 | (0.29) | 0.25 | (0.30) | −0.03 | (0.19) | −0.06 | (0.19) | −0.08 | (0.20) |
| DS 4 | −0.89 | (0.08) | −0.89 | (0.04) | −0.86 | (0.05) | −0.84 | (0.11) | −0.84 | (0.08) | −0.82 | (0.07) |
| DS 5 | −0.89 | (0.10) | −0.83 | (0.15) | −0.74 | (0.23) | 92.91 | (471.78) | 149.62 | (774.18) | 117.87 | (584.89) |
| DS 6 | 0.29 | (2.67) | 2.70 | (8.06) | 1.25 | (4.91) | 3.04 | (7.68) | 2.60 | (6.80) | 2.77 | (7.73) |
| DS 7 | −0.15 | (0.16) | −0.06 | (0.17) | −0.03 | (0.19) | −0.03 | (0.25) | 0.04 | (0.31) | 0.06 | (0.29) |
| DS 8 | −0.08 | (0.29) | 0.00 | (0.30) | 0.04 | (0.32) | 0.07 | (0.23) | 0.10 | (0.26) | 0.10 | (0.26) |
| DS 9 | −0.20 | (0.25) | −0.05 | (0.33) | 0.02 | (0.35) | 0.61 | (2.03) | 0.58 | (1.93) | 0.59 | (1.84) |
| Average | −0.02 | (0.76) | 0.31 | (1.37) | 0.20 | (1.04) | 12.41 | (61.53) | 22.28 | (114.47) | 16.34 | (80.23) |

Results are based on 30 runs with different train-test splits. Reported is the mean performance improvement (e.g., 0.1 means 10 %) and the standard deviation (Std.) in parentheses.

## J   Sensitivity Analysis

### J.1   Experiments for Over-Parameterized Neural Networks

In the following, we compare DCON to SGD in an over-parameterized setting as discussed in the related work section of the main paper. That is, the number of hidden neurons $N$ is set to a very large number compared to the number of training samples $m$. Previous research proved for this setting that gradient descent converges to a globally optimal solution (Du et al., 2019a;b; Zeyuan et al., 2019; Zou & Gu, 2019). To do so, we choose dataset DS7 ($m = 297$) on which DCON performed worst in our main experiments in Section 6.2 and vary the number of hidden units $N$ in $\{2^9, 2^{10}, 2^{11}, 2^{12}, 2^{13}\}$. We choose the hyperparameters according to the best-performing ones for $N = 30$. For Adam, we use early stopping with a patience of 50 monitoring the training loss to eventually observe convergence of SGD. Our experiments are shown in Figure 15.

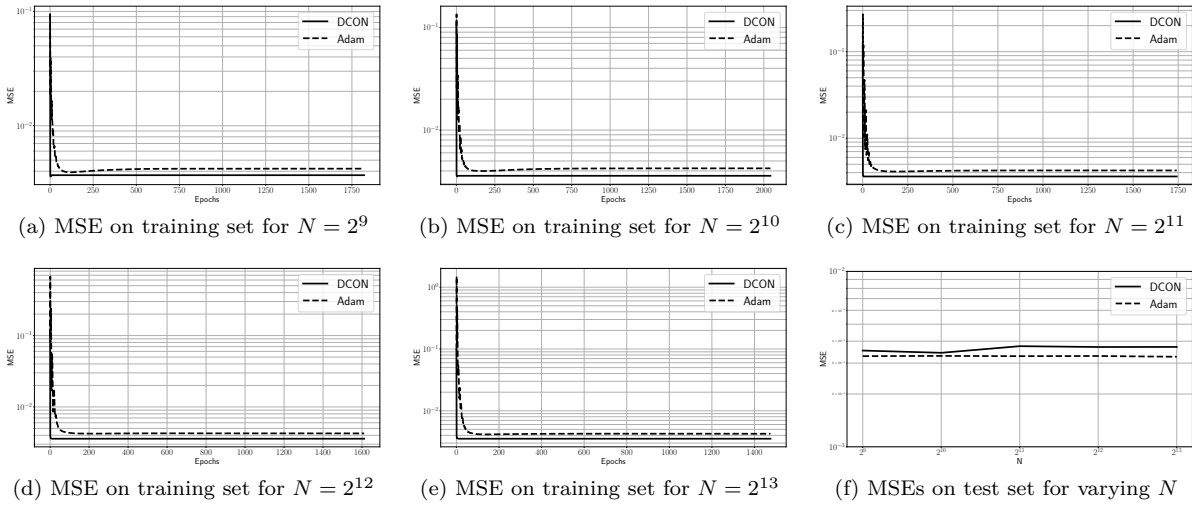

(a) MSE on training set for $N = 2^9$     (b) MSE on training set for $N = 2^{10}$     (c) MSE on training set for $N = 2^{11}$

(d) MSE on training set for $N = 2^{12}$     (e) MSE on training set for $N = 2^{13}$     (f) MSEs on test set for varying $N$

Figure 15: Performance on training and test set for DS7 with varying number of hidden neurons.

As expected, Figure 15 indicates the convergence of Adam in the over-parameterized setting, yielding a constant performance on the test set for $N \in \{2^9, 2^{10}, 2^{11}, 2^{12}, 2^{13}\}$. Furthermore, we observe that DCON also benefits from over-parameterization, yielding similar results as Adam. However, DCON converges much faster.

### J.2   Fast Convergence in Over-Regularized Settings

Our experiments also indicate a relationship between the magnitude of the regularization parameter $\gamma$ and convergence speed. We analyze this numerically in the following. To do so, we use the datasets DS1 and DS7 and vary the regularization parameter $\gamma \in \{10^r : r \in \{-5, -4, \ldots, 3, 4\}\}$. We set $N = 30$ and stop DCON when $\|\theta^{k+1} - \theta^k\| < 10^{-4}$. Afterward, we plot the resulting number of epochs and the resulting test performance in Figure 16.

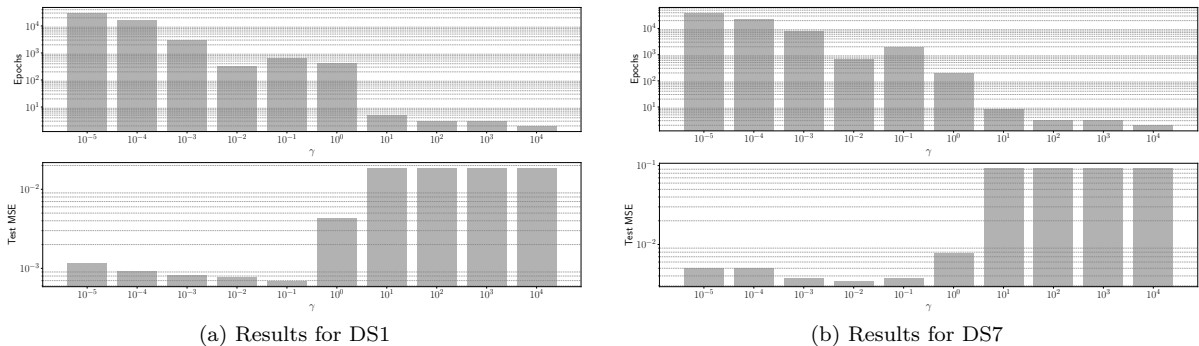

(a) Results for DS1  (b) Results for DS7

Figure 16: Number of epochs and test performance for DS1 and DS7 with varying regularization parameter: Results are based on datasets DS1 and DS7 with $N = 30$. We vary the regularization parameter $\gamma \in \{10^r : r \in \{-5, -4, \ldots, 3, 4\}\}$ and stop DCON when $\|\theta^{k+1} - \theta^k\| < 10^{-4}$. The number of epochs is visualized in the upper plot, while the lower plot shows the corresponding test performance.

As we can see, DCON converges faster for larger values of $\gamma$. At the same time, the test performance increases due to over-regularization. Further theoretical analyses of this behavior are beyond the scope of this paper and left for future research. Nevertheless, a theoretical relationship between the regularization parameter $\gamma$ and convergence speed might help to determine a value of $\gamma$ that balances both metrics.

## K  Future Work

### K.1  A Parallel Approach for Solving (QP)

As mentioned in Appendix C, our algorithm may be limited by memory requirements. The reason is that the singular value decomposition of $M^T$ involves a – in general – dense $m \times m$ matrix $V$. As a remedy, we propose a decomposition into batches in the following. This eventually allows one to solve (QP) very efficiently in the case of larger $m$. To do so, we consider again (QP), which is given by

$$
\inf \ \left\langle \begin{pmatrix} \beta - d \\ d \\ 0 \\ 0 \end{pmatrix}, \begin{pmatrix} v^1 \\ v^2 \\ v^3 \\ v^4 \end{pmatrix} \right\rangle + \frac{1}{m} \begin{pmatrix} v^1 \\ v^2 \\ v^3 \\ v^4 \end{pmatrix}^T \begin{pmatrix} (\alpha^2 + \gamma)I & -\gamma I & 0 & 0 \\ -\gamma I & \gamma I & 0 & 0 \\ 0 & 0 & 0 & 0 \\ 0 & 0 & 0 & 0 \end{pmatrix} \begin{pmatrix} v^1 \\ v^2 \\ v^3 \\ v^4 \end{pmatrix}
$$
$$
\text{s.t.} \quad \begin{pmatrix} -I & I & M & -M \end{pmatrix} \begin{pmatrix} v^1 \\ v^2 \\ v^3 \\ v^4 \end{pmatrix} = 0,
$$
$$
\begin{pmatrix} v^1 & v^2 & v^3 & v^4 \end{pmatrix} \geq 0.
$$
(352)

For a simpler notation, we dropped the superscript $g_l$ from $\beta$ and the subscripts $y^*$ from $d$, and $l$ from $\alpha$. Now, assume we split the dataset of size $m$ into $B$ batches of sizes $m_b > 0$ for $b \in \{1, \dots, B\}$ with $m = m_1 + m_2 + \cdots + m_B$. For the rest of this section, we use the following notation:

$$v^1 = (v_1^1, v_2^1, \dots, v_B^1) \text{ with } v_b^1 \in \mathbb{R}^{m_b} \text{ for all } b \in \{1, \dots, B\}, \tag{353}$$
$$v^2 = (v_1^2, v_2^2, \dots, v_B^2) \text{ with } v_b^2 \in \mathbb{R}^{m_b} \text{ for all } b \in \{1, \dots, B\}, \tag{354}$$
$$\beta = (\beta_1, \beta_2, \dots, \beta_B) \text{ with } \beta_b \in \mathbb{R}^{m_b} \text{ for all } b \in \{1, \dots, B\}, \tag{355}$$
$$d = (d_1, d_2, \dots, d_B) \text{ with } d_b \in \mathbb{R}^{m_b} \text{ for all } b \in \{1, \dots, B\}, \tag{356}$$
$$I_b \in \mathbb{R}^{m_b \times m_b} \text{ the identity matrix of size } m_b \times m_b, \tag{357}$$
$$I_{n+1} \in \mathbb{R}^{(n+1) \times (n+1)} \text{ the identity matrix of size } (n+1) \times (n+1), \tag{358}$$
$$M_b \in \mathbb{R}^{m_b \times (n+1)} \text{ the matrix formed by the rows of } M \text{ corresponding to batch } b, \tag{359}$$
$$A_b = \begin{pmatrix} -I_b & I_b & M_b & -M_b \end{pmatrix}. \tag{360}$$

At this point, we emphasize that we use the term "batch" in this setting to denote the partition of the training set into smaller subsets. This follows the terminology from batch processing in parallel computing where data is processed in chunks to reduce peak memory consumption. It should not be mistaken with batches from traditional gradient-based neural learning, which are used to compute only inexact approximations of the gradient to speed up computations. Conversely, our approach still solves the underlying problem exactly as demonstrated in the following.

For the above splitting, we yield

$$
\left\langle \begin{pmatrix} \beta - d \\ d \\ 0 \\ 0 \end{pmatrix}, \begin{pmatrix} v^1 \\ v^2 \\ v^3 \\ v^4 \end{pmatrix} \right\rangle = \left\langle \begin{pmatrix} \beta_1 - d_1 \\ \beta_2 - d_2 \\ \vdots \\ \beta_B - d_B \\ d_1 \\ d_2 \\ \vdots \\ d_B \\ 0 \\ 0 \end{pmatrix}, \begin{pmatrix} v_1^1 \\ v_2^1 \\ \vdots \\ v_B^1 \\ v_1^2 \\ v_2^2 \\ \vdots \\ v_B^2 \\ v^3 \\ v^4 \end{pmatrix} \right\rangle = \sum_{b=1}^{B} \left\langle \begin{pmatrix} \beta_b - d_b \\ d_b \\ 0 \\ 0 \end{pmatrix}, \begin{pmatrix} v_b^1 \\ v_b^2 \\ v^3 \\ v^4 \end{pmatrix} \right\rangle
$$
(361)

for the linear term and

$$
\frac{1}{m}\begin{pmatrix} v^1 \\ v^2 \\ v^3 \\ v^4 \end{pmatrix}^T \begin{pmatrix} (\alpha^2+\gamma)I & -\gamma I & 0 & 0 \\ -\gamma I & \gamma I & 0 & 0 \\ 0 & 0 & 0 & 0 \\ 0 & 0 & 0 & 0 \end{pmatrix} \begin{pmatrix} v^1 \\ v^2 \\ v^3 \\ v^4 \end{pmatrix} \tag{362}
$$

$$
= \frac{1}{m}\begin{pmatrix} v_1^1 \\ v_2^1 \\ \vdots \\ v_B^1 \\ v_1^2 \\ v_2^2 \\ \vdots \\ v_B^2 \\ v^3 \\ v^4 \end{pmatrix}^T \begin{pmatrix} (\alpha^2+\gamma)I_1 & 0 & \cdots & 0 & -\gamma I_1 & 0 & \cdots & 0 & 0 & 0 \\ 0 & (\alpha^2+\gamma)I_2 & \cdots & \vdots & 0 & -\gamma I_2 & \cdots & \vdots & \vdots & \vdots \\ \vdots & \vdots & \ddots & \vdots & \vdots & \vdots & \ddots & \vdots & \vdots & \vdots \\ 0 & \cdots & \cdots & (\alpha^2+\gamma)I_B & 0 & \cdots & \cdots & -\gamma I_B & \vdots & \vdots \\ -\gamma I_1 & 0 & \cdots & 0 & \gamma I_1 & 0 & \cdots & 0 & 0 & 0 \\ 0 & -\gamma I_2 & \cdots & 0 & 0 & \gamma I_2 & \cdots & 0 & 0 & 0 \\ \vdots & \vdots & \ddots & \vdots & \vdots & \vdots & \ddots & \vdots & \vdots & \vdots \\ 0 & \cdots & \cdots & -\gamma I_B & 0 & \cdots & \cdots & \gamma I_B & 0 & 0 \\ 0 & \cdots & \cdots & 0 & 0 & \cdots & \cdots & 0 & 0 & 0 \\ 0 & \cdots & \cdots & 0 & 0 & \cdots & \cdots & 0 & 0 & 0 \end{pmatrix} \begin{pmatrix} v_1^1 \\ v_2^1 \\ \vdots \\ v_B^1 \\ v_1^2 \\ v_2^2 \\ \vdots \\ v_B^2 \\ v^3 \\ v^4 \end{pmatrix} \tag{363}
$$

$$
= \sum_{b=1}^{B} \frac{1}{m}\begin{pmatrix} v_b^1 \\ v_b^2 \\ v^3 \\ v^4 \end{pmatrix}^T \begin{pmatrix} (\alpha^2+\gamma)I_b & -\gamma I_b & 0 & 0 \\ -\gamma I_b & \gamma I_b & 0 & 0 \\ 0 & 0 & 0 & 0 \\ 0 & 0 & 0 & 0 \end{pmatrix} \begin{pmatrix} v_b^1 \\ v_b^2 \\ v^3 \\ v^4 \end{pmatrix} \tag{364}
$$

for the quadratic term. By defining

$$
q_{y^*,b} = \begin{pmatrix} \beta_b - d_b \\ d_b \\ 0 \\ 0 \end{pmatrix}, \tag{365}
$$

$$
Q_{l,b} = \frac{2}{m}\begin{pmatrix} (\alpha^2+\gamma)I_b & -\gamma I_b & 0 & 0 \\ -\gamma I_b & \gamma I_b & 0 & 0 \\ 0 & 0 & 0 & 0 \\ 0 & 0 & 0 & 0 \end{pmatrix}, \tag{366}
$$

$$
v_b = (v_b^1, v_b^2, v_b^3, v_b^4), \tag{367}
$$

we yield the equivalence of $(\mathsf{QP})$ with

$$
\begin{aligned}
\inf \quad & \sum_{b=1}^{B} \langle q_{y^*,b}, v_b \rangle + \frac{1}{2} v_b^T Q_{l,b} v_b \\
\text{s.t.} \quad & A_b v_b = 0 \text{ for all } b \in \{1, \ldots, B\}, \\
& v_b \geq 0 \text{ for all } b \in \{1, \ldots, B\}, \\
& v^3 \geq 0, \\
& v^4 \geq 0, \\
& v_b^3 = v^3 \text{ for all } b \in \{1, \ldots, B\}, \\
& v_b^4 = v^4 \text{ for all } b \in \{1, \ldots, B\}.
\end{aligned} \tag{368}
$$

Note that the $B$ quadratic programs from above are merely coupled by the common variables $v^3$ and $v^4$. As a last step, we introduce a quadratic penalty term for the last two equality constraints. We then yield

$$
\begin{aligned}
\inf \quad & \sum_{b=1}^{B} \langle q_{y^*,b}, v_b \rangle + \frac{1}{2} v_b^T Q_{l,b} v_b + \frac{\mu_1}{2} \sum_{b=1}^{B} \|v_b^3 - v^3\|^2 + \frac{\mu_2}{2} \sum_{b=1}^{B} \|v_b^4 - v^4\|^2 \\
\text{s.t.} \quad & A_b v_b = 0 \text{ for all } b \in \{1, \ldots, B\}, \\
& v_b \geq 0 \text{ for all } b \in \{1, \ldots, B\}, \\
& v^3 \geq 0, \\
& v^4 \geq 0,
\end{aligned} \tag{QPP}
$$

where $\mu_1 > 0$ and $\mu_2 > 0$ are penalty parameters. To solve (QPP), we make again use of a block coordinate descent approach outlined in the following.

Updating $v^3$: To update $v^3$, we solve the following quadratic program

$$
\begin{aligned}
\inf \quad & \frac{B\mu_1}{2} (v^3)^T I_{n+1} v^3 - \left\langle \mu_1 \sum_{b=1}^{B} v_b^3, v^3 \right\rangle \\
\text{s.t.} \quad & v^3 \geq 0.
\end{aligned} \tag{369}
$$

Updating $v^4$: To update $v^4$, we solve the quadratic program

$$
\begin{aligned}
\inf \quad & \frac{B\mu_4}{2} (v^4)^T I_{n+1} v^4 - \left\langle \mu_2 \sum_{b=1}^{B} v_b^4, v^4 \right\rangle \\
\text{s.t.} \quad & v^4 \geq 0.
\end{aligned} \tag{370}
$$

Updating $v_b$: Finally, to update $v_b$, we solve the quadratic program

$$
\begin{aligned}
\inf \quad & \langle \tilde{q}_{y^*,b}, v_b \rangle + \frac{1}{2} v_b^T \tilde{Q}_{l,b} v_b \\
\text{s.t.} \quad & A_b v_b = 0, \\
& v_b \geq 0,
\end{aligned} \tag{371}
$$

where

$$
\tilde{q}_{y^*,b} = \begin{pmatrix} \beta_b - d_b \\ d_b \\ -\mu_1 v^3 \\ -\mu_2 v^4 \end{pmatrix}, \tag{372}
$$

$$
\tilde{Q}_{l,b} = \frac{2}{m} \begin{pmatrix} (\alpha^2 + \gamma) I_b & -\gamma I_b & 0 & 0 \\ -\gamma I_b & \gamma I_b & 0 & 0 \\ 0 & 0 & \frac{m\mu_1}{2} I_{n+1} & 0 \\ 0 & 0 & 0 & \frac{m\mu_2}{2} I_{n+1} \end{pmatrix}. \tag{373}
$$

Note that the updates of $v^3$ and $v^4$ can be computed by solving quadratic programs with system matrices of size $(n+1) \times (n+1)$. Afterward, all quadratic programs given in (371) are decoupled from one another and, therefore, can be solved in parallel. Furthermore, all of the above subproblems are strictly convex. The ADMM approach derived in the last section can still be used to solve these quadratic programs by adjusting the coefficients in Equation (177) accordingly.

In summary, our parallel algorithm for solving (QP) is outlined in Algorithm 4.

---

**Algorithm 4:** Parallel (QP)-Solver

---

**Input:** Batch sizes $m_1, \ldots, m_B$, Parameter $w > 1$ for increasing the penalty parameter
**Output:** Solution $(v^1, v^2, v^3, v^4)$ of (QP)

1  Initialize $v^3$, $v^4$, and $v_b$ for all $b \in \{1, \ldots, m\}$
2  **while** convergence criterion not met **do**
3      **while** convergence criterion not met **do**
4          **for** $b \in \{1, \ldots, B\}$ **do**
5              Compute $\tilde{q}_{y^*, b}$ and $\tilde{Q}_{l, b}$
6              Update $v_b$ by solving

$$\inf \quad \left\langle \tilde{q}_{y^*, b}, v_b \right\rangle + \frac{1}{2} v_b^T \tilde{Q}_{l, b} v_b$$
$$\text{s.t.} \quad A_b v_b = 0,$$
$$v_b \geq 0.$$

7          **end**
8          Update $v^3$ by solving

$$\inf \quad \frac{B\mu_1}{2}(v^3)^T I_{n+1} v^3 - \left\langle \mu_1 \sum_{b=1}^{B} v_b^3, v^3 \right\rangle$$
$$\text{s.t.} \quad v^3 \geq 0.$$

9          Update $v^4$ by solving

$$\inf \quad \frac{B\mu_4}{2}(v^4)^T I_{n+1} v^4 - \left\langle \mu_2 \sum_{b=1}^{B} v_b^4, v^4 \right\rangle$$
$$\text{s.t.} \quad v^4 \geq 0.$$

10      **end**
11      Update $\mu_1 \leftarrow w\mu_1$
12      Update $\mu_2 \leftarrow w\mu_2$
13  **end**
14  **return** $(v_1^1, \ldots, v_B^1, v_1^2, \ldots, v_B^2, v^3, v^4)$

---

We refrain from an in-depth convergence analysis at this point. Nevertheless, one can establish the convergence up to a subsequence of the inner `while`-loop to a solution of (QPP) for fixed penalty parameters via Proposition 2.7.1 in Bertsekas (2016). Moreover, the outer `while`-loop converges up to a subsequence to a solution of (QP) due to Theorem 17.1 in Nocedal & Wright (2006).

Algorithm 4 allows one to solve (QP) very efficiently. The inner `for`-loop consists of $B$ decoupled quadratic programs, which can all be solved in parallel. With the above, algorithm DCON can thus be scaled to much larger problem instances. For future research, more sophisticated approaches for handing the equality constraints $v_b^3 = v^3$ for all $b \in \{1, \ldots, B\}$ and $v_b^4 = v^4$ for all $b \in \{1, \ldots, B\}$ in (368) could be of interest. One example is, for instance, the augmented Lagrangian method.

### K.2 An inexact DCA Approach for the DC Subproblem

To further counteract the computational complexity of DCON, it might be worth considering inexact versions of DCA, see Zhang & Yamada (2023) and the references therein. In this way, it might be possible to use an iterative solver (as the one presented in the last section) to solve (QP) and stop the computations prematurely if a certain threshold of optimality is reached. Note that a more involved convergence analysis taking into account these inexact DC steps might be necessary in this case.

### K.3 Extension for Deep Neural Networks

To extent DCON to deeper neural networks a greedy layer-wise approach might be considered (compare to Bengio et al. (2006)). That is, one first trains a shallow neural network and then uses the features decoded in the hidden layer of that network as an input to train another shallow neural network. In that way, new layers are stacked on top of the previous ones, while each of them is trained individually using DCON. As the

performance should increase with each layer, one can also consider stopping the training process prematurely after a fixed number of epochs for each layer. Even a single epoch so that every neuron is considered only once for each layer and a "full" training for the last layer is thinkable.

## References in Appendix

Hedy Attouch and Jérôme Bolte. On the convergence of the proximal algorithm for nonsmooth functions involving analytic features. *Mathematical Programming*, 116(1):5–16, 2009. ISSN 1436-4646.

Hedy Attouch, Jérôme Bolte, and Benar Fux Svaiter. Convergence of descent methods for semi-algebraic and tame problems: Proximal algorithms, forward-backward splitting, and regularized Gauss-Seidel methods. *Mathematical Programming*, 137(1):91–129, 2013. ISSN 1436-4646.

Alberto Bemporad, Komei Fukuda, and Fabio D Torrisi. Convexity recognition of the union of polyhedra. *Computational Geometry*, 18(3):141–154, 2001.

Yoshua Bengio, Pascal Lamblin, Dan Popovici, and Hugo Larochelle. Greedy layer-wise training of deep networks. *Advances in Neural Information Processing Systems (NeurIPS)*, 19, 2006.

Dimitri P. Bertsekas. *Nonlinear Programming*. Athena Scientific, Belmont, MA, 3rd edition, 2016.

Jacek Bochnak, Michel Coste, and Marie-Françoise Roy. *Real Algebraic Geometry*. Springer, Berlin, Heidelberg, 1998. ISBN 978-3-642-08429-4.

Daniel Boley. Local linear convergence of the alternating direction method of multipliers on quadratic or linear programs. *SIAM Journal on Optimization*, 23(4):2183–2207, 2013.

Jérôme Bolte, Aris Daniilidis, and Adrian Lewis. The łojasiewicz inequality for nonsmooth subanalytic functions with applications to subgradient dynamical systems. *SIAM Journal on Optimization*, 17(4): 1205–1223, 2007a.

Jérôme Bolte, Aris Daniilidis, Adrian Lewis, and Masahiro Shiota. Clarke subgradients of stratifiable functions. *SIAM Journal on Optimization*, 18(2):556–572, 2007b.

Jérôme Bolte, Trong Phong Nguyen, Juan Peypouquet, and Bruce W Suter. From error bounds to the complexity of first-order descent methods for convex functions. *Mathematical Programming*, 165(2):471–507, 2017.

Simon Du, Jason D. Lee, Haochuan Li, Liwei Wang, and Xiyu Zhai. Gradient descent finds global minima of deep neural networks. *International Conference on Machine Learning (ICML)*, pp. 1675–1685, 2019a.

Simon Du, Xiyu Zhai, Barnabas Poczos, and Aarti Singh. Gradient descent provably optimizes overparameterized neural networks. *International Conference on Learning Representations (ICLR)*, 2019b.

James E Falk. A linear max—min problem. *Mathematical Programming*, 5(1):169–188, 1973.

Noah Golowich, Alexander Rakhlin, and Ohad Shamir. Size-independent sample complexity of neural networks. In *Conference On Learning Theory (COLT)*, pp. 297–299, 2018.

Jean-Baptiste Hiriart-Urruty and Claude Lemaréchal. *Fundamentals of Convex Analysis*. Springer, Berlin, Heidelberg, 2004.

Meng Huang, Ming-Jun Lai, Abraham Varghese, and Zhiqiang Xu. On dc based methods for phase retrieval. In *International Conference Approximation Theory*, pp. 87–121, 2019.

Thi Hoai Le An and Pham Dinh Tao. Solving a class of linearly constrained indefinite quadratic problems by d.c. algorithms. *Journal of Global Optimization*, 11(3):253–285, 1997. ISSN 1573-2916.

Thi Hoai Le An and Pham Dinh Tao. The DC (difference of convex functions) programming and DCA revisited with DC models of real world nonconvex optimization problems. *Annals of Operations Research*, 133(1):23–46, 2005. ISSN 1572-9338.

Mehryar Mohri, Afshin Rostamizadeh, and Ameet Talwalkar. *Foundations of Machine Learning*. Adaptive computation and machine learning. MIT Press, Cambridge, MA, 2nd edition, 2018.

Boris S Mordukhovich, Nguyen Mau Nam, and ND Yen. Fréchet subdifferential calculus and optimality conditions in nondifferentiable programming. *Optimization*, 55(5-6):685–708, 2006.

Yurii Nesterov. *Introductory Lectures on Convex Optimization: A Basic Course*, volume 87. Springer, New York, NY, 2003.

Jorge Nocedal and Stephen Wright. *Numerical Optimization*. Springer, New York, NY, 2006.

Mohammad Hossein Rafiei and Hojjat Adeli. Novel machine-learning model for estimating construction costs considering economic variables and indexes. *Journal of Construction Engineering and Management*, 144 (12), 2018.

Ali Rahimi and Benjamin Recht. Random features for large-scale kernel machines. *Advances in Neural Information Processing Systems (NeurIPS)*, 2007.

Ralph Tyrrell Rockafellar. *Convex Analysis*, volume 36. Princeton University Press, Princeton, NJ, 1997.

Masahiro Shiota. *Geometry of Subanalytic and Semialgebraic Sets*, volume 150 of *Progress in Mathematics*. Birkhäuser, Boston, MA, 1997. ISBN 9781461273783.

Pham Dinh Tao and Le Thi Hoai An. A DC optimization algorithm for solving the trust-region subproblem. *SIAM Journal on Optimization*, 8(2):476–505, 1998.

Pham Dinh Tao and LT Hoai An. Convex analysis approach to DC programming: theory, algorithms and applications. *Acta mathematica vietnamica*, 22(1):289–355, 1997.

Jinshan Zeng, Tim Tsz-Kit Lau, Shaobo Lin, and Yuan Yao. Global convergence of block coordinate descent in deep learning. *International Conference on Machine Learning (ICML)*, pp. 7313–7323, 2019.

Allen-Zhu Zeyuan, Li Yuanzhi, and Song Zhao. A convergence theory for deep learning via over-parameterization. *International Conference on Machine Learning (ICML)*, pp. 242–252, 2019.

Yi Zhang and Isao Yamada. An inexact proximal linearized dc algorithm with provably terminating inner loop. *arXiv preprint arXiv:2305.06520*, 2023.

Difan Zou and Quanquan Gu. An improved analysis of training over-parameterized deep neural networks. In *Advances in Neural Information Processing Systems (NeurIPS)*, pp. 2055–2064, 2019.

