# OpenReview forum: "A Globally Convergent Algorithm for Neural Network Parameter Optimization Based on Difference-of-Convex Functions"
_TMLR — Accepted by TMLR_

### Review · Reviewer_eoLq · 2023-10-22

**Summary Of Contributions:**

This paper proposes an algorithm named DCON for optimizing the parameters of single hidden layer neural networks. DCON directly solve PNN, and partly decompose the loss as a DC function. They use a BCD approach to loop over different blocks and derive efficient procedures to (approximately) solve the defined subproblems. They prove DCON converges in value and give the conditions under which the training loss converges.

**Audience:**

Yes

**Claims And Evidence:**

Yes

**Requested Changes:**

1. I suggest that the author add a detailed explanation of the alpha subproblem in Proposition 3 and introduce how to generate the alpha subproblem.
2. It is recommended that the author supplement the basis for the filter options introduced in appendix F.2 and give a reasonable explanation.
3. I suggest that the author add experimental analysis of Figures 3 to 11 in appendix F.6. For example, (g) to (i) in Figure 10 respectively represent the changes in MSE values of Adam and DCON with the increase of epoch under the conditions of N=10, N=20, and N=30. It can be seen that DCON increases with the training epoch. The model stabilizes faster as the increase in
4. I suggest that the author adjust the order of writing and write all the preparation work in the third section, so that the article will be more organized.

**Strengths And Weaknesses:**

Strengths:
1. This paper proposes a novel method for optimizing the parameters of single hidden layer neural networks.

2.The writing of the whole article is relatively smooth, first deducing the subproblems in BCD, then providing methods for solving different subproblems, and finally presenting the proof of convergence and convergence conditions.

3.This paper provides sufficient theoretical proof and proves both the proposition and assumption given.

4.This paper gives a relatively detailed introduction to the experimental settings and details.
Weaknesses：
1.This paper mentions alpha subproblem in Proposition 3, but does not give a good explanation of the occurrence of alpha subproblem, making it difficult to understand during the reading process.

2.Also in section 4.2.2 Solution of Alpha Subproblem, the solution to the alpha subproblem is relatively brief.

3.The author introduces the filter options for the data set in the appendix F.2, but does not give a basis for filtering the data set in this way.

4.Appendix F.6 does not provide a text analysis of Figures 3 to 11. It only provides an experimental effect diagram, which cannot explain the experimental effect well.

5.The location of Preliminaries in subsection 5.1.1 is strange. I think it can be presented in section 3.

---

> ### Author Response · Authors · 2023-11-22
> **Response to Reviewer eoLq**
>
> We would like to thank the reviewer for the positive and helpful feedback. Below we answer all of your questions point by point.
>
> _1. I suggest that the author add a detailed explanation of the alpha subproblem in Proposition 3 and introduce how to generate the alpha subproblem._
>
> We thank the reviewer for this suggestion. For better understanding, we added an explanation and visualization in Appendix B.1.
>
> **Action:** We added new explanations (see our **Appendix B.1**).
>
> _2. It is recommended that the author supplement the basis for the filter options introduced in appendix F.2 and give a reasonable explanation_
>
> Thank you for this comment. The filter options were carefully chosen to align with the focus on optimizing parameters in single hidden layer neural networks for regression tasks. In particular, the range of instances was chosen to strike a balance between computational feasibility and rigorous evaluation. Datasets with fewer than 100 instances might not provide enough data to meaningfully train and test a neural network, while those with more than 1000 instances could introduce prohibitive computational demands for 30 train-test splits as performed in this work. In summary, these criteria were chosen to provide a fair and rigorous evaluation of the model performance.
>
> **Action:** We added an explanation to justify our choice of filter options (see our new materials in **Appendix F.2**).
>
> _3. I suggest that the author add experimental analysis of Figures 3 to 11 in appendix F.6. For example, (g) to (i) in Figure 10 respectively represent the changes in MSE values of Adam and DCON with the increase of epoch under the conditions of N=10, N=20, and N=30. It can be seen that DCON increases with the training epoch. The model stabilizes faster as the increase in_
>
> Thank you for this suggestion. We **added further explanations for the figures** in Appendix F.6.
>
> _4. I suggest that the author adjust the order of writing and write all the preparation work in the third section, so that the article will be more organized._
>
> Thank you for your suggestion regarding the organization of our manuscript. We have carefully checked our manuscript to follow the following logic: (i) preparation work (=Section 3), (ii) our algorithm (=Section 4), and (iii) our theoretical analysis (=Section 5). Thereby, we differentiate the preparation work (=Section 3) from our contributions, namely, the derivation of the subproblems (Section 4) and the convergence conditions (Section 5). We hope this explanation clarifies our structural choices. We hope that the organization helps the logic flow but we are also open to further suggestions if the reviewer team has a strong preference.

---

### Review · Reviewer_QEZY · 2023-11-12

**Summary Of Contributions:**

This paper proposes a new approach for optimizing neural networks that have one hidden layer and ReLU activations.   The main insight is that if one considers this objective with respect to the incoming weights to a single arbitrary hidden neuron, this objective is a difference of two convex functions -- which is a type of objective for which specialized solvers exist.  So the idea is to repeatedly sweep through all the neurons and optimize the objective w.r.t the incoming weights to each one.  After each outer iteration, the objective is optimized w.r.t the weights of the second-layer weights, which is just a least squares problem.

**Audience:**

Yes

**Claims And Evidence:**

Yes

**Requested Changes:**

1.  Since one of the main concerns here is scalability, could the authors include an experiment where they show how wall clock runtime (say, to time to reach a fixed objective value) varies as a function of (1) number of training data points, and (2) data dimension?  (To construct problems of varying data dimension, you could take the dataset to be random feature encodings of MNIST or CIFAR of varying data dimension).  https://gregorygundersen.com/blog/2019/12/23/random-fourier-features/.

2. In the introduction, the authors write,
> Foremost, stochastic gradient descent (SGD) and variants thereof (e. g., using momentum) are applied to optimize neural networks. However, gradient-based methods can get stuck in local minima, do not have general convergence guarantees, and require expert knowledge during training due to many hyperparameters (e. g., initial learning rate, momentum)

The context makes it seem like the authors are implying that their proposed method does not suffer from these three limitations.  I do agree that the proposed method involves less hyperparameter tuning, but I do not necessarily agree that the proposed method is more prone to avoiding local minima, or that it enjoys stronger guarantees than gradient descent does under analogous assumptions.  Could the authors clarify?

**Strengths And Weaknesses:**

Strengths:
 - the overall idea is novel and interesting
 - experiments show that on certain small scale problems (the largest problem considered is a 10k subset of MNIST), the proposed approach empirically finds a global optimum
- experiments show that the proposed method can sometimes outperform Adam on a per-iteration basis (where iteration means _outer iteration_ for the proposed approach, which involves much more computation than an iteration of Adam).

Weaknesses:
- as with standard approaches to training, the approach proposed here is ultimately heuristic, in the sense that there are no guarantees of global convergence to a global optimum that are provable under realistic assumptions
- the approach is much less scalable than standard approaches to training

---

> ### Author Response · Authors · 2023-11-22
> **Response to Reviewer QEZY**
>
> We want to thank the reviewer for the positive and constructive feedback. We answer all questions in the following.
>
> _1. Since one of the main concerns here is scalability, could the authors include an experiment where they show how wall clock runtime (say, to time to reach a fixed objective value) varies as a function of  (1) number of training data points, and (2) data dimension? (To construct problems of varying data dimension, you could take the dataset to be random feature encodings of MNIST or CIFAR of varying data dimension)._
>
> Thank you for this comment. We followed your suggestions and added **new runtime experiments** (see our **new Appendix H**). We find that DCON is indeed slower than Adam if the number of training examples gets large. However, we can still yield runtime improvements given that DCON does not rely on any hyperparameters for training. Furthermore, our main contribution at this point is of theoretical nature and, while potential approaches to scale DCON to larger datasets are discussed in the Appendix, we leave them for future work.
>
> **Action:** We add new runtime experiments (see our **new Appendix H**).
>
> _2. In the introduction, the authors write, “Foremost, stochastic gradient descent (SGD) and variants thereof (e. g., using momentum) are applied to optimize neural networks. However, gradient-based methods can get stuck in local minima, do not have general convergence guarantees, and require expert knowledge during training due to many hyperparameters (e. g., initial learning rate, momentum)” The context makes it seem like the authors are implying that their proposed method does not suffer from these three limitations. I do agree that the proposed method involves less hyperparameter tuning, but I do not necessarily agree that the proposed method is more prone to avoiding local minima, or that it enjoys stronger guarantees than gradient descent does under analogous assumptions. Could the authors clarify?_
>
> Thank you for your feedback. We have revised the manuscript to remove the statement that gradient-based methods get stuck in local minima. Instead, we now focus on our primary contribution, namely, global convergence. Here, it is important to note that our DCON algorithm offers stronger convergence guarantees compared to SGD, especially as general convergence results for SGD often rely on assumptions like over-parameterization. In contrast, DCON's convergence is independent of assumptions regarding the number of hidden neurons, which we believe is a significant advantage of our approach.
>
> **Action:** We have carefully revised our introduction to clarify our contribution (see our revised **Section 1**).

---

> > ### Comment · Reviewer_QEZY · 2023-11-28
> > **thanks**
> >
> > Thanks for adding this clarification, and for adding the runtime experiments.  For the runtime experiments, I think it would be proper to also include a heatmap of runtime with "best hyperparameter" for Adam, rather than only "average over hyperparameters."  I do understand that this is a little unfair in Adam's favor, since any algorithm that has an additional hyperparameter is going to be better if you are allowed to optimizer over that hyperparameter, but you could note this explicitly.

---

> > > ### Author Response · Authors · 2023-12-03
> > > **Response to Reviewer QEZY**
> > >
> > > Thank you very much for your comment. For reasons of transparency, we followed your suggestion and also **added a heatmap** showing the “best” runtime of Adam. The **new plot** is in **Appendix H**. As you already pointed out, these numbers are favoring Adam as the best performing hyperparameters are a-priori unknown. We also added this information to our text.

---

### Review · Reviewer_JRQp · 2023-11-13

**Summary Of Contributions:**

The paper presents a new optimization algorithm for single layer deep network for regression tasks in a deterministic regime for small to medium datasets with an appropriate regularization term and a ReLU activation function. The algorithm is based on using the specific structure of the problem use the design difference of convex functions framework. Each iteration solves several subproblems, one per coordinate of the intermediate layer and one for the outer layer, not counting additional decompositions to solve the subproblem associated to each coordinate.

The authors present several proofs of convergence: convergence towards limiting critical point, local convergence and convergence in values. The proofs follow a template of previous work (Attouch et al 2013) but each step required a careful analysis of the problem. Rates of convergence are provided.

Several experiments illustrate the performance of the algorithm on small (up to 1 000 samples) to medium datasets (10 000 samples). The algorithm outperforms a standard stochastic baseline (Adam) in terms of training and testing loss on UCI datasets (up to 1 000 samples) and is on par with Adam on the MNIST dataset. Some experiments confirm the theoretical findings. No comparisons on time are provided, while it appears that the proposed method can be computationally very expensive.

**Audience:**

Yes

**Broader Impact Concerns:**

This is a technical paper presenting a new optimization algorithm. No concern of ethical implications are foreseen.

**Claims And Evidence:**

Yes

**Requested Changes:**

**Required changes**:

- The paper could really benefit from being able to justify the relevance of the proposed objective. One possibility would be to compare the given model to some much simpler models like classical linear regression for example with least squares or non-parametric regression with kernels. This would strengthen the case for needing a new optimizer for the current objective.
- It is unclear whether the baseline and the proposed algorithm were run on the same objective given the specific choice of regularization chosen by the authors for their algorithms. The authors need to clarify that. Moreover the authors need to justify clearly their choice of regularization. If the authors could also explain how the algorithm could tackle standard $\ell_2$ regularization it would be great.
- The authors need to detail fully the computational complexity of the given algorithm and compare to the computational complexity of SGD. Experiments in time in the appendix would also help understand the relevance of the proposed algorithm in practice. Poor results in time would not make the paper rejected but they cannot be missing for the paper to be accepted.

**Suggestions/Questions**:
- Given that the algorithm is not yet practical enough to tackle usual tasks in deep learning, the authors may consider changing the tone of a paper from "we have an algorithm outperforming baselines" to "we explore alternatives to baselines and their advantages". For example, on page 2, the authors say "DCON offers a key benefit for machine learning practice". It is unclear how "practical" the DCON algorithm truly is especially in terms of overall computational complexity. If DCON takes 10 times more time to run, a simple Adam could have been launched with just 10 hyper-parameters for example. In the numerical examples, the authors often affirm that DCON is superior. Adding a pinch of salt and explaining something like "in epochs, DCON appear superior" would help emphasize the exploratory perspective taken by the authors. Otherwise if the authors consider that DCON should be used as is, and that the work is mature enough for real applications, more experiments are needed. Right now, having to subsample MNIST images and consider only a fifth of the whole training data suggests that the algorithm is not mature enough for applications. To be clear, the paper proposes a thorough and original proof of convergence that would appeal to the audience. It may just be worth balancing better the cost of such theoretical contributions in terms of practical considerations.
- The authors may consider changing the title to "... algorithm for single layer neural network parameter optimization".
- For this paper to have larger audience and potential follow-ups, it would be great if the authors could suggest ways to extend the algorithm beyond single layer neural networks even if a proof of convergence would be lost.
- How close or far is the current objective from some simple hierarchical Bayes model? Rather than focusing on neural networks which are generally deeper in practice, the authors may consider applications in Bayesian statistics that may often deal with composition similar to the ones used here.
- Could the proofs be extended in a stochastic setting?
- Can the proofs take into account approximate solutions of the subproblems?
- Why did the authors consider a one vs all approach for the MNIST dataset while they could have considered a simple squared loss on one-hot labels?

*Other details*:

- Eq. 10, 11: unfortunate notation $y_k$ given that $y$ was already used as the targeted values for regression.
- Page 17, test loss paragraph: a factor of 0.64 reads as if the algorithm was performing worse, maybe write
- Page 19, paragraph before conclusion: "many datasets in practice and theory are of similar size": first what are datasets in theory? second, I would disagree that many datasets are of such size (MNIST, CIFAR10, or even numerous UCI datasets have much larger sizes).

**Strengths And Weaknesses:**

**Strengths**:

- The proposed approach is original. The algorithm can have the benefit of having no hyperparameter to tune a priori since it amounts to a block coordinate descent.
- The proofs are thorough. The authors present convergence results through multiple angles and derive convergence rates.
- The experiments show some promising results on small datasets.

**Weaknesses**:


- The proposed algorithm is tailored for a very particular structure of objective. Deep learning and machine learning strive by the flexibility to choose any model. I list here the current limitations:
  - *Single* hidden layer networks: while applications exist for those, neural networks are generally deeper.
  - Specific regularization: the authors consider a data-weighted $\ell_2$ regularization, that is non-standard. The algorithm could probably be run with a standard $\ell_2$ regularization on the weights though, but it should be discussed.
  - Deterministic regime: the algorithm and proofs are tailored for full-batch operations. They scale poorly with the number of samples m. Stochastic variants of the algorithm could be more relevant.
  - Regression problem with least-squared losses: numerous applications consider classification tasks with a logistic, a.k.a. cross-entropy, loss.
  - ReLU activation function: It is unclear how other activation functions such as a SiLU or a GeLU could be used.
- The complete computational complexity of each iteration of the algorithm is not fully discussed and compared to the complexity of a simple stochastic gradient descent and variants thereof. To compare fairly with the baselines, the experiments should be run for an equal amount of computational ressource, such as plotting the performance of the algorithms in time.
- Assumptions appear limiting: in the Appendix, the authors consider subsampling MNIST images to 10x10 for the problem to satisfy assumption 1. Note by the way, that the algorithm should adapt to the problem not the reverse.

---

> ### Author Response · Authors · 2023-11-22
> **Response to Reviewer JRQp**
>
> We want to thank the reviewer for the positive feedback and answer all questions in the following.
>
> # Response to required changes
>
> _1. The paper could really benefit from being able to justify the relevance of the proposed objective. One possibility would be to compare the given model to some much simpler models like classical linear regression for example with least squares or non-parametric regression with kernels. This would strengthen the case for needing a new optimizer for the current objective._
>
> Thank you for your suggestion. While classical linear regression and non-parametric kernel regression are valuable for certain problems, they fall short in capturing complex, non-linear relationships inherent in many datasets. Neural networks excel in modeling these non-linearities but suffer from challenges related to optimization. Our work addresses these specific challenges, providing a theoretical foundation for the optimization of single hidden layer neural networks. Nevertheless, we included the suggested experiments in our **new Appendix I**.
>
> **Action:** We add new experiments (see our **new Appendix I**).
>
> _2. It is unclear whether the baseline and the proposed algorithm were run on the same objective given the specific choice of regularization chosen by the authors for their algorithms. The authors need to clarify that._
>
> We want to thank the reviewer for mentioning this and helping us describe our experiments more transparently. In Section 6.1, we merely mentioned that Adam uses an $\ell_2$ regularization. We have now added the **standard** to $\ell_2$ regularization to emphasize this. At this point, however, we also want to clarify that our regularization is merely a standard $\ell_2$ regularization with an additional weight matrix $B$ and, hence, can be seen as a standard regularizer in a different energy norm. As all norms are equivalent in finite dimensions, this also means that there exist positive constants $c$ and $C$ such that
> $$ c\lVert \theta \rVert_2 \le \lVert \theta \rVert_B \le C \lVert \theta \rVert_2. $$
> Furthermore, all performance results in Table 1 and Table 2 are reported using the mean squared error and are thus independent of the regularization term.
>
> **Action:** We added the above explanation to clarify that a standard regularization was chosen for Adam in **Section 6.1**.
>
> _3. Moreover the authors need to justify clearly their choice of regularization. If the authors could also explain how the algorithm could tackle standard regularization it would be great._
>
> Thank you. We have revised our manuscript and **now clearly justify our regularization** (see our revised **Section 3.1**). Our choice of regularization is important for our theoretical derivations as (i) it allows to derive a convenient DC structure of the objective function, and (ii) it allows for a closed-form solution of the sub-gradients needed in our DCA routine. We included these points in the problem statement in section 3.1. A standard $\ell_2$ regularization would require a different DC representation, hence, for now DCON does not support standard regularization.
>
> **Action:** We added an explanation justifying our choice of the regularization (see our new materials in **Section 3.1**).
>
> _4. The authors need to detail fully the computational complexity of the given algorithm and compare to the computational complexity of SGD. Experiments in time in the appendix would also help understand the relevance of the proposed algorithm in practice. Poor results in time would not make the paper rejected but they cannot be missing for the paper to be accepted._
>
> We appreciate your valuable suggestion regarding the need for a detailed analysis of computational complexity. However, after careful consideration and review of the available literature, we believe that providing an accurate and comprehensive comparison of the computational complexity to SGD presents significant challenges, as the overall complexity of SGD depends on the number of iterations required to converge, which can vary significantly depending on factors such as the learning rate, or the curvature of the objective function. Even analyses on how, for instance, the batch size affects the computational efficiency are based on empirical studies rather than theoretical derivations (also see Golmant et. al. (2018)). Nevertheless, we added the suggested runtime experiments in our new **Appendix H**. We find that DCON is indeed slower than Adam if the number of training examples gets large. However, we can still yield runtime improvements given that DCON does not rely on any hyperparameters for training.
>
> **Action:** We add new runtime experiments (see our **new Appendix H**).

---

> > ### Author Response · Authors · 2023-11-22
> > **Response to Reviewer JRQp (Part 2)**
> >
> > # Response to Suggestions/Questions
> >
> > _1. Given that the algorithm is not yet practical enough to tackle usual tasks in deep learning, the authors may consider changing the tone of a paper from "we have an algorithm outperforming baselines" to "we explore alternatives to baselines and their advantages". For example, on page 2, the authors say "DCON offers a key benefit for machine learning practice". It is unclear how "practical" the DCON algorithm truly is especially in terms of overall computational complexity. If DCON takes 10 times more time to run, a simple Adam could have been launched with just 10 hyper-parameters for example. In the numerical examples, the authors often affirm that DCON is superior. Adding a pinch of salt and explaining something like "in epochs, DCON appear superior" would help emphasize the exploratory perspective taken by the authors_
> >
> > Thank you for this valuable suggestion. We have toned down our manuscript. For example, in the mentioned paragraph, we now write more carefully: *“Nevertheless, future work is needed to scale DCON to larger datasets in practice. We point to potential research directions in our discussion.”*.
> >
> > As a side remark, we also want to point to our new runtime experiments which show that DCON is indeed slower than Adam if the number of training examples gets larger, but can still yield large runtime improvements given that DCON does not rely on any hyperparameters for training. Again, we took great strides to navigate the pros/cons of the different methods more transparently and carefully.
> >
> > **Action:** We have carefully checked our manuscript and **toned down our language**. We also added a `pinch of salt’ in our explanations in Section 6.3..
> >
> > _2. For this paper to have larger audience and potential follow-ups, it would be great if the authors could suggest ways to extend the algorithm beyond single layer neural networks even if a proof of convergence would be lost._
> >
> > Thank you for your insightful comment. We agree that expanding the applicability of our algorithm to multi-layer neural networks could significantly broaden its audience and potential for follow-up studies. One idea of training a deeper neural network layer-by-layer has been proposed in previous literature. This approach, often referred to as greedy layer-wise training, has shown promise in certain contexts (Bengio et al. 2006). It involves the training of one layer at a time, freezing the parameters of trained layers, and then training subsequent layers. Using this method, DCON can be extended to train arbitrarily deep networks, simply by optimizing each layer individually before moving on to the next.
> >
> > **Action:** We **added a discussion of potential extensions** to multi-layer networks for future research (see our new materials in **Appendix K.3**).
> >
> > _3. Can the proofs take into account approximate solutions of the subproblems? Could the proofs be extended in a stochastic setting?_
> >
> > We want to thank the reviewer for this important question. For the first question, we refer to Zhang and Yamada (2023). Therein, the authors discuss different algorithms and approaches to tackle inexact DCA steps that might be useful in future works. We added discussion in our new **Appendix K.2**. The second question seems to refer to the ability of SGD to be trained in batches. The main part of DCON is to solve the quadratic program (QP). To address your second question, we outline how the quadratic program can also be solved using batches in **Appendix K.1**.
> >
> > **Action:** We **added a discussion** of approximate solution techniques for subproblems (see **Appendix K.2**). We further offer a discussion of a parallel solution approach using batches (see **Appendix K.1**).
> >
> > _4. Why did the authors consider a one vs all approach for the MNIST dataset while they could have considered a simple squared loss on one-hot labels?_
> >
> > Thank you for this question. The reason for our choice is that one-hot encoded labels would require vectorial outputs of the neural network while we consider only one-dimensional outputs.

---

> > > ### Author Response · Authors · 2023-11-22
> > > **Response to Reviewer JRQp (Part 3)**
> > >
> > > # Response to Other Details
> > >
> > > _Eq. 10, 11: unfortunate notation given that was already used as the targeted values for regression._
> > >
> > > Thank you for pointing this out. We adhere to standard notation in the DCA literature by using $y^\ast$ to ensure consistency with established conventions. The asterisk in $y^\ast$ serves to differentiate subgradients from target values.
> > >
> > > **Action:** We added a short clarification to introduce our notation (see our improvements to **Section 4.2.1**).
> > >
> > > _Page 17, test loss paragraph: a factor of 0.64 reads as if the algorithm was performing worse, maybe write_
> > >
> > > Thank you for highlighting this point. To clarify the performance metrics, we have **added detailed legends** to Tables 1 and 2. We hope that this helps in a more accurate interpretation of the results, especially regarding the factors in the context of test loss.
> > >
> > > _Page 19, paragraph before conclusion: "many datasets in practice and theory are of similar size": first what are datasets in theory? second, I would disagree that many datasets are of such size (MNIST, CIFAR10, or even numerous UCI datasets have much larger sizes)._
> > >
> > > Thank you. We have **carefully revised** the sentence to *“For comparison, Lee et. al (2013) use datasets with $m\approx100$, while datasets in Haugh & Kogan (2004) correspond to $m\approx4000$.”* to avoid any confusion.
> > >
> > > **References**
> > >
> > > Golmant, N., Vemuri, N., Yao, Z., Feinberg, V., Gholami, A., Rothauge, K., ... & Gonzalez, J. (2018). On the computational inefficiency of large batch sizes for stochastic gradient descent. arXiv preprint arXiv:1811.12941.
> > >
> > > Bengio, Y., Lamblin, P., Popovici, D., & Larochelle, H. (2006). Greedy Layer-Wise Training of Deep Networks. Advances in Neural Information Processing Systems.
> > >
> > > Zhang, Y., & Yamada, I. (2023). An Inexact Proximal Linearized DC Algorithm with Provably Terminating Inner Loop. arXiv preprint arXiv:2305.06520.

---

> ### Comment · Reviewer_JRQp · 2023-11-26
> **Thank you for your answers**
>
> **About required changes**
>
> The additional results fully answer my concerns. The current manuscript gives a full picture of the contributions made by the author on an experimental level. Even if the proposed algorithm may not be the new state of the art algorithm for deep learning, the current results offer clear perspectives on its potential benefits.
>
> **About suggestions**
>
> Thank you for taking the suggestions into account. I believe the manuscript offers an interesting perspective on the subject, even if it remains somewhat theoretical.
>
> **About other details**
>
> Thanks also for taking those into account.

---

### Review · Reviewer_bQDE · 2023-11-19

**Summary Of Contributions:**

The paper proposed a method called DCON, tailored for training 1-hidden-layer neural networks. Convergence analysis is provided. This is a purely theoretical paper.

**Audience:**

Yes

**Claims And Evidence:**

Yes

**Requested Changes:**

1. The motivation of the proposed method is to find the global optimal solution, in lieu of local ones. However, the theoretical guarantee of finding a global solution is rather weak: the global convergence only holds if the initial solution is close enough to the global min. This is not consistent with the motivation of the paper. In this sense, the advantage over the gradient-based method is unclear.  Please discuss.



2. For me, this is a purely theoretical paper that helps us understand the training of the most simple neural nets, i.e.,  1-hidden-layer networks. I would not expect the proposed method could be easily scaled up to more complicated network structures and large real datasets. However, it is still reasonable to ask the practical computational complexity of the proposed method. DCON needs to solve certain QP sub-problems, which could be expensive.    Please provide the wall-clock running time comparison with Sgd or Adam on your current experiments. Larger-scaled experiments with deep networks will also be appreciated.


3. To prove the convergence of DCON, What is the minimum number of hidden units required in theory?


4. Missing reference： the problem considered in the script is related to that in [1]. In [1], they consider finding a near-global solution of one-hidden-layer networks via projected gradient-based methods. Since the projected gradient-based method is cheap to implement, the proposed method in [1] seems more practical than DCON. Please discuss.

[1] when expressivity meets trainability: fewer than n neurons can work, Zhang et al, NeurIPS 2021.

**Strengths And Weaknesses:**

**Strength:**  The writing is mostly clear and logical. The analysis seems reasonable and believable.  I apologize that I did not have time to go through the whole proof.


**Drawbacks:** See below.

---

> ### Author Response · Authors · 2023-11-22
> **Response to Reviewer bQDE**
>
> We want to thank the reviewer for the positive feedback. We incorporated the requested changes into our revised paper. Below, we also provide a point-by-point response.
>
> _1. the global convergence only holds if the initial solution is close enough to the global min. This is not consistent with the motivation of the paper. In this sense, the advantage over the gradient-based method is unclear. Please discuss._
>
> Thank you for asking this important question. We want to clarify that **global convergence** does **not** mean **convergence to global minima**; see also Lanckriet and Sriperumbudur (2009) and the discussion about global convergence therein. Global convergence is the convergence independent of starting points and, hence, in our case, weight initialization. Lanckriet and Sriperumbudur (2009) describe it as *“The property of global convergence expresses, in a sense, the certainty that the algorithm works. It is very important to stress the fact that it does not imply (contrary to what the term might suggest) convergence to a global optimum for all initial points $x_0$.”*.
>
> **Action:** We **clarified the meaning of “global convergence”** in our paper, and, to this end, spell out the definition carefully (see our revised **Sections 1 and 5**).
>
> _2. For me, this is a purely theoretical paper that helps us understand the training of the most simple neural nets, i.e., 1-hidden-layer networks. I would not expect the proposed method could be easily scaled up to more complicated network structures and large real datasets. However, it is still reasonable to ask the practical computational complexity of the proposed method. DCON needs to solve certain QP sub-problems, which could be expensive. Please provide the wall-clock running time comparison with Sgd or Adam on your current experiments._
>
> Thank you for this comment. We followed your suggestions, as well as the related ones from Reviewers JRQp and QEZY, and added **new runtime experiments** (see our **new Appendix H**). We find that DCON is indeed slower than Adam if the number of training examples gets large. However, we can still yield runtime improvements given that DCON does not rely on any hyperparameters for training. Furthermore, our main contribution at this point is of theoretical nature and, while potential approaches to scale DCON to larger datasets and deeper neural network structures are discussed in the Appendix, we leave them for future work.
>
> **Action:** We add new runtime experiments (see our **new Appendix H**).
>
> _3. To prove the convergence of DCON, What is the minimum number of hidden units required in theory?_
>
> Thank you. An integral property of our DCON algorithm is its ability to achieve convergence independent of the number of hidden units. This characteristic is a fundamental aspect of our theoretical framework, and we believe that this is one of our key contributions to the field of neural network parameter optimization. Similar convergence analyses of algorithms for neural network parameter optimization usually rely on some sort of over-parameterization, i.e., a minimal number of hidden units, and, hence, do not offer general convergence results as our work.
>
> _4. Missing reference: the problem considered in the script is related to that in [1]. In [1], they consider finding a near-global solution of one-hidden-layer networks via projected gradient-based methods. Since the projected gradient-based method is cheap to implement, the proposed method in [1] seems more practical than DCON. Please discuss._
>
> Thank you for pointing us to Zhang et. al. (2021). We **included the reference** in the paragraph of our introduction where we discuss the expressiveness of SLFNs. Nevertheless, there are important differences between Zhang et. al. (2021) and our work. Their paper does _not_ yield any theoretical convergence result, and has rather to be seen as an empirical analysis. In contrast, our focus are theoretical guarantees.
>
> **Action**: We added the suggested reference.
>
>
> **References**
>
> Lanckriet, G., & Sriperumbudur, B. K. (2009). On the convergence of the concave-convex procedure. Advances in Neural Information Processing Systems (NeurIPS), 22.
>
> Zhang, J., Zhang, Y., Hong, M., Sun, R., & Luo, Z. Q. (2021). When Expressivity Meets Trainability: Fewer than $n$ Neurons Can Work. Advances in Neural Information Processing Systems (NeurIPS), 34, 9167-9180.

---

> > ### Comment · Reviewer_bQDE · 2023-11-23
> > **Thanks for your response**
> >
> > Thanks for the detailed response. I am still confused about the term "global convergence": if this term means that "the algorithm can converge to stationary points for any initialization" as interpreted by the authors, then isn't  "global convergence" also hold for gradient-based methods like sgd? Then what is the theoretical benefit of the proposed method?

---

> > > ### Author Response · Authors · 2023-11-23
> > > **Response to Reviewer bQDE**
> > >
> > > Thank you for your fast response. In general, there is **no** theoretical guarantee for global convergence of SGD. To establish convergence results for SGD, one usually relies on convexity or smoothness assumptions (or over-parameterization as discussed above). However, ReLU-neural-networks, as we are considering in our manuscript, are **not** differentiable and the objective function (see (PNN)) is highly **non-convex**.
> > >
> > > To the best of our knowledge, there are **no** convergence results more general than the ones provided in Davis et. al. (2020). Nevertheless, even here the authors can merely guarantee subsequence convergence which, in general, can result in arbitrarily bad cyclical behaviors. See also Figure 1 (b) in Zeng et. al. (2019), where the authors demonstrate how SGD fails to train a neural network, while alternative approaches based on block coordinate descent can achieve good results after a few epochs.
> > >
> > > We hope this clarifies your question. Please do not hesitate to reach out in case you have more questions.
> > >
> > > **References**
> > >
> > > Davis, D., Drusvyatskiy, D., Kakade, S., & Lee, J. D. (2020). Stochastic subgradient method converges on tame functions. Foundations of computational mathematics, 20(1), 119-154.
> > >
> > > Zeng, J., Lau, T. T. K., Lin, S., & Yao, Y. (2019, May). Global convergence of block coordinate descent in deep learning. In International conference on machine learning (pp. 7313-7323). PMLR.

---

> > > > ### Comment · Reviewer_bQDE · 2023-11-28
> > > > **Thanks for the clarification**
> > > >
> > > > Thanks for the clarification. Under the definition of  "global convergence" by authors, the setting of the paper is related to reference [1]. In [1], they also consider a 1-hidden-layer network with regularizers. They prove that under mild assumptions on width, certain GD-variants can converge to stationary points (actually near-optimal points). Please discuss the relation and differences.
> > > >
> > > >
> > > > [1] Liang S, Sun R, Srikant R. Achieving small test error in mildly overparameterized neural networks. arXiv preprint arXiv:2104.11895. 2021 Apr 24.

---

> > > > > ### Author Response · Authors · 2023-12-03
> > > > > **Response to Reviewer bQDE**
> > > > >
> > > > > Thank you for your comment. We are more than happy to discuss the differences:
> > > > > - First, we want to point out that the mentioned paper [1] considers a **binary classification** task, whereas our manuscript is based on a **regression task**.
> > > > > - Second, the paper attempts to provide theoretical guarantees under mild overparameterization assumptions. However, it is still a **restrictive** assumption that our work does **not** rely on. At this point, we want to emphasize again that our convergence analysis does **not** depend on the number of hidden neurons.
> > > > >
> > > > > Let us demonstrate the second point based on a numerical example. Let us assume that we aim to perform a convergence analysis based on an over-parameterization assumption as in [1]. In our experiments, the neural networks as per [1] would need to have a minimum of 100-1000 hidden neurons (depending on the actual size of the dataset).  In contrast, we are considering much smaller networks with only 10, 20, or 30 hidden neurons. Here, the convergence analysis from [1] would be **not** applicable. Even more strikingly, for the MNIST dataset, the convergence analysis of [1] would be **not** realistic. Here, the over-parameterization assumption as in [1] would even require **at least** 60,000 (!) hidden neurons.
> > > > >
> > > > > **Action:** We have **added the reference** by Liang et al [1] to our related work. We further **spell out clearly the difference** between [1] and our work. Furthermore, we also added a footnote explaining why the over-parameterization assumption is often not feasible in practice.

---

### Author Response · Authors · 2023-11-22
**General Response**

Thank you very much for seeing value in our paper and for your helpful suggestions! We addressed all of your comments point-by-point below. We **uploaded a revised version of our paper**, where we highlight key changes colored in **red**.

Our **main improvements** are the following:

1. We have carefully revised our paper to spell out clearly that our main contribution is of _theoretical nature_. We offer a _conceptual algorithm_ that shows how neural networks may be trained with theoretical guarantees. We also acknowledge that our paper makes theoretical contributions, but the actual runtime of our algorithm is slower than highly-tuned state-of-the-art solvers.
2. We **added new runtime experiments** (see our **new Appendix H**). Therein, we compare the runtime of Adam to reach the same mean squared error as DCON. We find that overall our DCON algorithm outperforms Adam in terms of runtime due to the fact that DCON does not need to perform hyperparameter tuning. However, for individual training loops, i.e., for a fixed set of hyperparameters, Adam is indeed faster as the number of training examples increases.
3.  We added **new visualizations** to explain the alpha subproblem (see our **Appendix B.1**).
4. We **added several clarifications** along the lines suggested by the reviewers (see revised **Sections 1 and 5**). In particular, we clarified that _global convergence_ does **not** mean _convergence to global minima_. Rather, global convergence is the convergence independent of starting points (here: independent of weight initialization) (see Lanckriet and Sriperumbudur, 2009).
5. We added a more elaborate discussion of **potential extensions for future research** (see our revised **Appendix K**). Therein, we discuss, for example, extensions for multi-layer networks, approximate solution techniques, and a parallel approach to solve (QP) in batches.

Given these improvements, we are confident that our paper will be a valuable contribution to the theory of neural network parameter optimization and a good fit for TMLR.

---

### Decision · Action_Editor_CRjp · 2023-12-14

**Recommendation:** Accept with minor revision

**Comment:**

This paper proposes a block-wise algorithm together with theoretical analysis to optimize a one-hidden-layer ReLU network. The algorithm runs as follows: (1) Randomly choose one neuron and then optimize only the incoming weights to that layer. (2) Repeat step 1. (3) Optimize the readout weight, which is a quadratic problem. Reviewers believe that the idea is novel and original, the analysis is thorough, and the small-scale empirical results show some promising results. Several issues have been raised by reviewers (e.g., confusion about "global convergence," fair comparison to Adam, the potential shortcomings of the method, etc.), most of which have been addressed properly during the rebuttal. Finally, all reviewers recommend acceptance of the paper, which I agree with.


Two minor points.
(1.) The paper's 20-page length might be daunting for a general audience. Consider significantly shortening it by moving some technical details to an appendix. This would enhance its accessibility and clarity for a wider readership.


(2.) The definition of "global convergence" in the paper is non-conventional (pointed out by several reviewers.) Please consider alternative terminology, or at least **highlight** how it differs from the standard definition.

**Audience:**

The paper proposes a new method to optimize one-hidden layer Relu Networks. The method in the paper might be interesting to researchers working on theory / optimization of neural networks.

**Claims And Evidence:**

The current manuscript is thorough in its theoretical analysis. Several experiments illustrate the performance of the algorithm on small scales. Most importantly, "the revision offers a clear understanding of the potential shortcomings of the method" per reviewer.

---

> ### Author Response · Authors · 2024-01-09
> **Response to Action Editor**
>
> Thank you for the positive response. We have now uploaded our camera-ready version. Therein, we also followed your suggestions closely:
>
> 1. We have significantly shortened the paper by moving some technical details to the appendix. This should enhance the accessibility and clarity for a wider readership.
>
> 2. We added and highlighted an explicit definition of "global convergence" to our introduction.
>
> Thank you for handling the review processes in such a timely and helpful manner.